# Secondary Organic Aerosol (SOA) yields from NO$_3$ radical + isoprene based on nighttime aircraft power plant plume transects

Juliane L. Fry[1], Steven S. Brown[2,5], Ann M. Middlebrook[2], Peter M. Edwards[2,3,4], Pedro Campuzano-Jost[3,5], Douglas A. Day[3,5], José L. Jimenez[3,5], Hannah M. Allen[6], Thomas B. Ryerson[2], Ilana Pollack[2,3,a], Martin Graus[3,b], Carsten Warneke[2,3], Joost A. deGouw[3,5], Charles A. Brock[2], Jessica Gilman[2,3], Brian M. Lerner[2,3,c], William P. Dubé[2,3], Jin Liao[2,3,d], André Welti[2,3,e]

[1]Chemistry Department, Reed College, Portland, OR, USA
[2]Chemical Sciences Division, Earth System Research Laboratory, National Oceanic and Atmospheric Administration, Boulder, CO, USA
[3]Cooperative Institute for Research in Environmental Sciences, University of Colorado, Boulder, CO, USA
[4]Wolfson Atmospheric Chemistry Laboratories, Department of Chemistry, University of York, York, UK
[5]Department of Chemistry, University of Colorado, Boulder, CO, USA
[6]Division of Chemistry and Chemical Engineering, California Institute of Technology, Pasadena, CA, USA
[a]now at Department of Atmospheric Science, Colorado State University, Fort Collins, CO, USA
[b]now at Department of Atmospheric and Cryospheric Sciences, University of Innsbruck, Austria.
[c]now at Aerodyne Research, Inc., Billerica, MA, USA
[d]now at Universities Space Research Association, Columbia, MD, USA and NASA Goddard Space Flight Center, Atmospheric Chemistry and Dynamic Laboratory, Greenbelt, MD, USA
[e]now at Leibniz Institute for Tropospheric Research, Department of Physics, Leipzig, Germany

## Abstract

Nighttime reaction of nitrate radicals (NO$_3$) with biogenic volatile organic compounds (BVOC) has been proposed as a potentially important but also highly uncertain source of secondary organic aerosol (SOA). The southeast United States has both high BVOC and nitrogen oxide (NO$_x$) emissions, resulting in a large model-predicted NO$_3$-BVOC source of SOA. Coal-fired power plants in this region constitute substantial NO$_x$ emissions point sources into a nighttime atmosphere characterized by high regionally widespread concentrations of isoprene. In this paper, we exploit nighttime aircraft observations of these power plant plumes, in which NO$_3$ radicals rapidly remove isoprene, to obtain field-based estimates of the secondary organic aerosol yield from NO$_3$ + isoprene. Observed in-plume increases in nitrate aerosol are consistent with organic nitrate aerosol production from NO$_3$ + isoprene, and these are used to determine molar SOA yields, for which the average over 9 plumes is 9% (+/- 5%). Corresponding mass yields depend on the assumed molecular formula for isoprene-NO$_3$-SOA, but the average over 9 plumes is 27% (+/- 14%), on average larger than those previously measured in chamber studies (12 – 14% mass yield as ΔOA/ΔVOC after oxidation of both double bonds). Yields are larger for longer plume ages. This suggests that ambient aging processes lead more effectively to condensable material than typical chamber conditions allow. We discuss potential mechanistic explanations for this difference, including longer ambient peroxy radical lifetimes and heterogeneous reactions of NO$_3$-isoprene gas phase products.

More in-depth studies are needed to better understand the aerosol yield and oxidation
mechanism of $NO_3$ radical + isoprene, a coupled anthropogenic – biogenic source of SOA that
may be regionally significant.
**1 Introduction**
Organic aerosol (OA) is increasingly recognized as a globally important component of the fine
particulate matter that exerts a large but uncertain negative radiative forcing on Earth's climate
(Myhre et al., 2013) and adversely affects human health around the world (Lelieveld et al.,
2015).This global importance is complicated by large regional differences in OA concentrations
relative to other sources of aerosol such as black carbon, sulfate, nitrate and sea salt. OA
comprises 20 – 50% of total fine aerosol mass at continental mid-latitudes, but more in urban
environments and biomass burning plumes, and up to 90% over tropical forests (Kanakidou et
al., 2005;Zhang et al., 2007). Outside of urban centers and fresh biomass burning plumes, the
majority of this OA is secondary organic aerosol (SOA) (Jimenez et al., 2009), produced by
oxidation of directly emitted volatile organic compounds followed by partitioning into the aerosol
phase. Forests are strong biogenic VOC emitters, in the form of isoprene ($C_5H_8$), monoterpenes
($C_{10}H_{16}$), and sesquiterpenes ($C_{15}H_{24}$), all of which are readily oxidized by the three major
atmospheric oxidants, OH, $NO_3$, and $O_3$. The total global source of biogenic SOA from such
reactions remains highly uncertain, with a eview estimating it at 90 +/- 90 Tg C $yr^{-1}$ (Hallquist et
al., 2009), a large fraction of which may be anthropogenically controlled (Spracklen et al.,
2011;Goldstein et al., 2009;Carlton et al., 2010;Hoyle et al., 2011). As most $NO_3$ arises from
anthropogenic emissions, OA production from $NO_3$ + isoprene is one mechanism that could
allow for the anthropogenic control of biogenic SOA mass loading.
Isoprene constitutes nearly half of all global VOC emissions to the atmosphere, with a flux of
~600 Tg $yr^{-1}$ (Guenther et al., 2006). As a result, accurate global biogenic SOA budgets depend
strongly on yields from isoprene oxidation. Recent global modeling efforts find that isoprene
SOA is produced at rates from 14 (Henze and Seinfeld, 2006;Hoyle et al., 2007) to 19 TgC $yr^{-1}$
(Heald et al., 2008), which implies that it could constitute 27% (Hoyle et al., 2007) to 48%
(Henze and Seinfeld, 2006) to 78% (Heald et al., 2008) of total SOA (based also on varying
estimates of total SOA burden in each study). More recent observational constraints on SOA
yield from isoprene find complex temperature-dependent mechanisms that could affect vertical
distributions (Worton et al., 2013) and suggest that isoprene SOA constitutes from 17% (Hu et
al., 2015) to 40% (Kim et al., 2015) up to 48% (Marais et al., 2016) of total OA in the
southeastern United States. This large significance comes despite isoprene's low SOA mass
yields – two recent observational studies estimated the total isoprene SOA mass yield to be
~3% (Marais et al., 2016;Kim et al., 2015), and modeling studies typically estimate isoprene
SOA yields to be 4 to 10%, depending on the oxidant, in contrast to monoterpenes' yields of 10
to 20% and sesquiterpenes' yields of >40% (Pye et al., 2010). Furthermore, laboratory studies
of SOA mass yields may have a tendency to underestimate these yields, if they cannot access
the longer timescales of later-generation chemistry, or are otherwise run under conditions that
limit oxidative aging of first-generation products (Carlton et al., 2009).

Laboratory chamber studies of SOA mass yield at OA loadings of ~ 10 µg m$^{-3}$ from isoprene
have typically found low yields from O$_3$ (1% (Kleindienst et al., 2007)) and OH (2% at low NO$_x$ to
5% at high NO$_x$ (Kroll et al., 2006;Dommen et al., 2009); 1.3% at low NO$_x$ and neutral seed
aerosol pH but rising to 29% in the presence of acidic sulfate seed aerosol due to reactive
uptake of epoxydiols of isoprene (IEPOX) (Surratt et al., 2010)). One recent chamber study on
OH-initiated isoprene SOA formation focused on the fate of second-generation RO$_2$ radical
found significantly higher yields, up to 15% at low NO$_x$ (Liu et al., 2016), suggesting that omitting
later-generation oxidation chemistry could be an important limitation of early chamber
determinations of isoprene SOA yields. Another found an increase in SOA formed with
increasing HO$_2$ to RO$_2$ ratios, suggesting that RO$_2$ fate could also play a role in the variability of
previously reported SOA yields (D'Ambro et al., 2017).
For NO$_3$ oxidation of isoprene, early chamber experiments already pointed to higher yields (e.g.,
12% (Ng et al., 2008)) than for OH oxidation. Ng et al. (Ng et al., 2008) also observed chemical
regime differences: SOA yields were approximately two times larger when chamber conditions
were tuned such that first-generation peroxy radical fate was RO$_2$+RO$_2$ dominated than when it
was RO$_2$+NO$_3$ dominated. In addition, Rollins et al. (Rollins et al., 2009) observed a significantly
higher SOA yield (14%) from second-generation NO$_3$ oxidation than that when only one double
bond was oxidized (0.7%). This points to the possibility that later-generation, RO$_2$+RO$_2$
dominated isoprene + NO$_3$ chemistry may be an even more substantial source of SOA than
what current chamber studies have captured. Schwantes et al. (Schwantes et al., 2015)
investigated the gas-phase products of NO$_3$ + isoprene in the RO$_2$+HO$_2$ dominated regime and
found the major product to be isoprene nitrooxy hydroperoxide (INP, 75-78% molar yield), which
can photochemically convert to isoprene nitrooxy hydroxyepoxide (INHE), a molecule that might
contribute to SOA formation via heterogeneous uptake similar to IEPOX. Here again, multiple
generations of chemistry are required to produce products that may contribute to SOA.
Because the SOA yield appears to be highest for NO$_3$ radical oxidation, and isoprene is such an
abundantly emitted BVOC, oxidation of isoprene by NO$_3$ may be an important source of OA in
areas with regional NO$_x$ pollution. Since the SOA yield with neutral aerosol seed appears to be
an order of magnitude larger than that from other oxidants, even if only 10% of isoprene is
oxidized by NO$_3$, it will produce comparable SOA to daytime photo-oxidation. For example,
Brown et al. (Brown et al., 2009) concluded that NO$_3$ contributed more SOA from isoprene than
OH over New England, where > 20% of isoprene emitted during the previous day was available
at sunset to undergo dark oxidation by either NO$_3$ or O$_3$. The corresponding contribution to total
SOA mass loading was 1 – 17% based on laboratory yields (Ng et al., 2017). Rollins et al.
(Rollins et al., 2012) concluded that multi-generational NO$_3$ oxidation of biogenic precursors was
responsible for one-third of nighttime organic aerosol increases during the CalNex-2010
experiment in Bakersfield, CA. In an aircraft study near Houston, TX, Brown et al. (Brown et al.,
2013) observed elevated organic aerosol in the nighttime boundary layer, and correlated vertical
profiles of organic and nitrate aerosol in regions with rapid surface level NO$_3$ radical production
and BVOC emissions. From these observations, the authors estimated an SOA source from
NO$_3$ + BVOCs within the nocturnal boundary layer of 0.05 – 1 µg m$^{-3}$ h$^{-1}$. Carlton et al. (Carlton
et al., 2009) note the large scatter in chamber-measured SOA yields from isoprene
photooxidation and point throughout their review of SOA formation from isoprene to the likely
importance of poorly understood later generations of chemistry in explaining field observations.
We suggest that similar differences in multi-generational chemistry could explain the variation
among the (sparse) chamber and field observations of $NO_3$ + isoprene yields described in the
previous paragraph, and summarized in a recent review of $NO_3$ + BVOC oxidation mechanisms
and SOA formation (Ng et al., 2017).
The initial products of $NO_3$ + isoprene include organic nitrates, some of which will partially
partition to the aerosol phase. Organic nitrates in the particle phase ($pRONO_2$) are challenging
to quantify with online methods, due to both interferences and their often overall low
concentrations in ambient aerosol. Hence, field datasets to constrain modeled $pRONO_2$ are
sparse (Ng et al., 2017;Fisher et al., 2016). One of the most used methods in recent studies,
used also here, is quantification with the Aerodyne Aerosol Mass Spectrometer (AMS). Organic
nitrates thermally decompose in the AMS vaporizer and different approaches have been used to
apportion the organic fraction contributing to the total nitrate signal. Allan et al. (Allan et al.,
2004a) first proposed the use of nitrate peaks at *m/z* 30 and 46 to distinguish various nitrate
species with the AMS. Marcolli et al. (Marcolli et al., 2006), in the first reported tentative
assignment of aerosol organic nitrate using AMS data, used cluster analysis to analyze data
from the 2002 New England Air Quality Study. In that study, cluster analysis identified two
categories with high *m/z* 30 contributions. One of these peaked in the morning when $NO_x$ was
abundant and was more prevalent in plumes with lowest photochemical ages, potentially from
isoprene oxidation products. The second was observed throughout the diurnal cycle in both
fresh and aged plumes, and contained substantial *m/z* 44 contribution (highly oxidized OA). A
subsequent AMS laboratory and field study discussed and further developed methods for
separate quantification of organic nitrate (in contrast to inorganic nitrate) (Farmer et al., 2010). A
refined version of one of these separation methods, based on the differing $NO_2^+/NO^+$
fragmentation ratio for organic vs. inorganic nitrate, was later employed to quantify organic
nitrate aerosol at two forested rural field sites where strong biogenic VOC emissions and
relatively low NOx combined to make substantial organic nitrate aerosol concentrations ((Fry et
al., 2013;Ayres et al., 2015)). Most recently, Kiendler-Scharr et al. (Kiendler-Scharr et al., 2016)
used a variant of this method to conclude that across Europe, organic nitrates comprise ~40%
of submicron organic aerosol. Modeling analysis concluded that a substantial fraction of this
organic nitrate aerosol is produced via $NO_3$ radical initiated chemistry. Chamber studies have
employed this fragmentation ratio method to quantify organic nitrates (Boyd et al., 2015;Bruns et
al., 2010;Fry et al., 2009;Fry et al., 2011;Rollins et al., 2009), providing the beginnings of a
database of typical organonitrate fragmentation ratios from various BVOC precursors.
Measurements conducted at the SOAS ground site in Centreville, Alabama in 2013 found
evidence of significant organonitrate contribution to SOA mass loading. Xu et al. (Xu et al.,
2015) reported that organic nitrates constituted 5 to 12% of total organic aerosol mass from
AMS data applying a variant of the $NO_2^+/NO^+$ ratio method. They identify a nighttime-peaking
"LO-OOA" AMS factor which they attribute to mostly $NO_3$ oxidation of BVOC (in addition to $O_3$ +
BVOC). They estimated that the $NO_3$ radical oxidizes 17% of isoprene, 20% of α-pinene, and
38% of β-pinene in the nocturnal boundary layer at this site. However, applying laboratory-
based SOA yields to model the predicted increase in OA, Xu et al. predict only 0.7 µg m$^{-3}$ of
SOA would be produced, substantially lower than the measured nighttime LO-OOA production
of 1.7 µg m$^{-3}$.  The more recent analysis of Zhang et al. (Zhang et al., 2018) found a strong
correlation of monoterpene SOA with the fraction of monoterpene oxidation attributed to $NO_3$,
even for non-nitrate containing aerosol, suggesting an influence of $NO_3$ even in pathways that
ultimately eliminate the nitrate functionality from the SOA, such as hydrolysis or $NO_2$
regeneration. Ayres et al. (Ayres et al., 2015) used a correlation of overnight organonitrate
aerosol buildup with calculated net $NO_3$ + monoterpene and isoprene reactions to estimate an
overall $NO_3$ + monoterpene SOA mass yield of 40 – 80%. The factor of two range in this
analysis was based on two different measurements of aerosol-phase organic nitrates. These
authors used similar correlations to identify specific CIMS-derived molecular formulae that are
likely to be $NO_3$ radical chemistry products of isoprene and monoterpenes, and found minimal
contribution of identified first-generation $NO_3$ + isoprene products to the aerosol phase (as
expected based on their volatility). Lee et al. (Lee et al., 2016) detected abundant highly
functionalized particle-phase organic nitrates at the same site, with apparent origin both from
isoprene and monoterpenes, and both daytime and nighttime oxidation, and estimated their
average contribution to submicron organic aerosol mass to be between 3 – 8 %. For the same
ground campaign, Romer et al. (Romer et al., 2016) found evidence of rapid conversion from
alkyl nitrates to $HNO_3$, with total alkyl nitrates having an average daytime lifetime of 1.7 hours.
Xie et al. (Xie et al., 2013) used a model constrained by observed alkyl nitrate correlations with
$O_3$ from the INTEX-NA/ICARTT 2004 field campaign to determine a range of isoprene nitrate
lifetimes between 4 and 6 hours, with 40-50% of isoprene nitrates formed by $NO_3$ + isoprene
reactions. Laboratory studies show that not all organic nitrates hydrolyze to $HNO_3$ equally
rapidly: primary and secondary organic nitrates were found to be less prone to aqueous
hydrolysis than tertiary organic nitrates (Darer et al., 2011;Hu et al., 2011;Fisher et al.,
2016;Boyd et al., 2015). This suggests that field-based estimates of the contribution of organic
nitrates to SOA formation could be a lower limit, if they are based on measurement of those
aerosol-phase nitrates. This is because if hydrolysis is rapid, releasing $HNO_3$ but leaving behind
the organic fraction in the aerosol phase, then that organic mass would not be accurately
accounted for as arising from nitrate chemistry. This was addressed in a recent modeling study
of SOAS (Pye et al., 2015) in which modeled hydrolysis products of particulate organic nitrates
of up to 0.8 µg m$^{-3}$ additional aerosol mass loading in the southeast U.S. were included in the
estimate of change in OA due to changes in $NO_x$. Another recent GEOS-Chem modeling study
using of gas- and particle-phase organic nitrates observed during the SEAC[4]RS and SOAS
campaigns similarly finds $RONO_2$ to be a major sink of $NO_x$ across the SEUS region (Fisher et
al., 2016;Lee et al., 2016).
Complementing these SOAS ground site measurements, the NOAA-led SENEX (Southeast
Nexus) aircraft campaign conducted 18 research flights focused in part on studying the
interactions between biogenic and anthropogenic emissions that form secondary pollutants
between 3 June and 10 July 2013 (Warneke et al., 2016). Flight instrumentation focused on
measurement of aerosol precursors and composition enable the present investigation of SOA
yields using this aircraft data set. Edwards et al. (Edwards et al., 2017) used data from the
SENEX night flights to evaluate the nighttime oxidation of BVOC, observing high nighttime
isoprene mixing ratios in the residual layer that can undergo rapid $NO_3$ oxidation when sufficient
$NO_x$ is present. These authors suggest that past $NO_x$ reductions may have been uncoupled
from OA trends due to $NO_x$ not having been the limiting chemical species for OA production, but
that future reductions in $NO_x$ may decrease OA if $NO_3$ oxidation of BVOC is a substantial
regional SOA source. Because isoprene is ubiquitous in the nighttime residual layer over the
southeastern United States and the $NO_3$ + isoprene reaction is rapid, $NO_3$ reaction will be
dominant relative to $O_3$ in places with anthropogenic inputs of $NO_x$ (Edwards et al. (Edwards et
al., 2017) concludes that when $NO_2$/BVOC > 0.5, $NO_3$ oxidation will be dominant).  Hence, a
modest $NO_3$ + isoprene SOA yield may constitute a regionally important OA source.
Several modeling studies have investigated the effects of changing $NO_x$ on global and SEUS
SOA. Hoyle et al. (Hoyle et al., 2007) found an increase in global SOA production from 35 Tg yr$^{-1}$
$^{1}$ to 53 Tg yr$^{-1}$ since preindustrial times, resulting in an increase in global annual mean SOA
mass loading of 51%, attributable in part to changing $NO_x$ emissions. Zheng et al. (Zheng et al.,
2015) found only moderate SOA reductions from a 50% reduction in NO emissions: 0.9 – 5.6 %
for global $NO_x$ or 6.4 – 12.0% for southeast US $NO_x$, which they attributed to buffering by
alternate chemical pathways and offsetting tendencies in the biogenic vs. anthropogenic SOA
components. In contrast, Pye et al. (Pye et al., 2015) find a 9% reduction in total organic aerosol
in Centreville, AL for only 25% reduction in $NO_x$ emissions. A simple limiting-reagent analysis of
$NO_3$ + monoterpene SOA from power plant plumes across the United States found that between
2008 and 2011, based on EPA-reported $NO_x$ emissions inventories, some American power
plants shifted to the $NO_x$-limited regime (from 3.5% to 11% of the power plants), and showed
that these newly $NO_x$-limited power plants were primarily in the southeastern United States (Fry
et al., 2015). The effect of changing $NO_x$ on SOA burden is clearly still in need of further study.
Here, we present aircraft transects of spatially discrete $NO_x$ plumes from electric generating
units (EGU), or power plants (PP), as a method to specifically isolate the influence of $NO_3$
oxidation.  These plumes are concentrated and highly enriched in $NO_x$ over a scale of only a
few km (Brown et al., 2012), and have nitrate radical production rates ($P$($NO_3$)) 10 – 100 times
greater than those of background air. The rapid shift in $P$($NO_3$) allows direct comparison of air
masses with slow and rapid oxidation rates attributable to the nitrate radical, effectively isolating
the influence of this single chemical pathway in producing SOA and other oxidation products.
Changes in organic nitrate aerosol (pRON$O_2$) concentration and accompanying isoprene
titration enable a direct field determination of the SOA yield from $NO_3$ + isoprene.

## 2 Field campaign and experimental and modeling methods

The Southeast Nexus (SENEX: http://esrl.noaa.gov/csd/projects/senex/) campaign took place 3
June through 10 July 2013 as the NOAA WP-3D aircraft contribution to the larger Southeast
Atmospheric Study (SAS: http://www.eol.ucar.edu/field_projects/sas/), a large, coordinated
research effort focused on understanding natural and anthropogenic emissions, oxidation
chemistry and production of aerosol in the summertime atmosphere in the southeastern United
States. The NOAA WP-3D aircraft operated 18 research flights out of Smyrna, Tennessee,
carrying an instrument payload oriented towards elucidating emissions inventories and reactions
of atmospheric trace gases, and aerosol composition and optical properties (Warneke et al.,
2016). One of the major goals of the larger SAS study is to quantify the fraction of organic
aerosol that is anthropogenically controlled, with a particular focus on understanding how OA
may change in the future in response to changing anthropogenic emissions.
The subset of aircraft instrumentation employed for the present analysis of nighttime $NO_3$ +
isoprene initiated SOA production includes measurements used to determine $NO_3$ radical
production rate ($P(NO_3) = k_{NO2+O3}(T) [NO_2] [O_3]$), isoprene and monoterpene concentrations,
other trace gases for plume screening and identification, aerosol size distributions, and aerosol
composition. The details on the individual measurements and the overall aircraft deployment
goals and strategy are described in Warneke et al. (Warneke et al., 2016). Briefly, $NO_2$ was
measured by UV photolysis and gas-phase chemiluminescence (P-CL) and by cavity ringdown
spectroscopy, (CRDS), which agreed within 6%. $O_3$ was also measured by both gas-phase
chemiluminescence and CRDS and agreed within 8%, within the combined measurement
uncertainties of the instruments. Various volatile organic compounds were measured with
several techniques, including for the isoprene and monoterpenes of interest here, proton
reaction transfer mass spectrometry (PTR-MS) and canister whole air samples and post-flight
GC-MS analysis (iWAS/GCMS). A comparison of PTR-MS and iWAS/GCMS measurements of
isoprene during SENEX has high scatter due to imperfect time alignment and isoprene's high
variability in the boundary layer, but the slope of the intercomparison is 1.04 ((Warneke et al.,
2016); for more details on the VOC intercomparisons, see also Lerner et al., (Lerner et al.,
2017)). Acetonitrile from the PTRMS was used to screen for the influence of biomass burning.
Sulfur dioxide ($SO_2$) was used to identify emissions from coal-fired power plants.  All gas-phase
instruments used dedicated inlets, described in detail in the supplemental information for
Warneke et al. (Warneke et al., 2016).
Aerosol particles were sampled downstream of a low turbulence inlet (Wilson et al., 2004), after
which they were dried by ram heating, size-selected by an impactor with 1 µm aerodynamic
diameter size cut-off, and measured by various aerosol instruments (Warneke et al., 2016). An
ultra-high-sensitivity aerosol sizing spectrometer (UHSAS, Particle Metrics, Inc., Boulder, CO
(Brock et al., 2011;Cai et al., 2008)) was used to measure the dry submicron aerosol size
distribution down to about 70 nm. Data for the UHSAS are reported at 1 Hz whereas AMS data
were recorded roughly every 10 seconds. The ambient (wet) surface areas were calculated
according to the procedures described in Brock et al., 2016 (Brock et al., 2016). A pressure-
controlled inlet (Bahreini et al., 2008) was employed to ensure that a constant mass flow rate
was sampled by a compact time-of-flight aerosol mass spectrometer (C-ToF-AMS) which
measured the non-refractory aerosol composition (Drewnick et al., 2005). The aerosol volume
transmitted into the AMS was calculated by applying the measured AMS lens transmission
curve (Bahreini et al., 2008) to the measured particle volume distributions from the UHSAS. For
the entire SENEX study, the mean, calculated fraction of aerosol volume behind the 1 micron
impactor that was transmitted through the lens into the AMS instrument was 97% (with ±4%
standard deviation), indicating that most of the submicron aerosol volume measured by the
sizing instruments was sampled by the AMS.

After applying calibrations and the composition-dependent collection efficiency following
Middlebrook et al. (Middlebrook et al., 2012), the limits of detection for the flight analyzed here
were 0.05 µg m$^{-3}$ for nitrate, 0.26 µg m$^{-3}$ for organic mass, 0.21 µg m$^{-3}$ for ammonium, and 0.05
µg m$^{-3}$ for sulfate, determined as three times the standard deviation of 10-second filtered air
measurements obtained for 10 minutes during preflight and 10 minutes during postflight (110
datapoints). Note that the relative ionization efficiency for ammonium was 3.91 and 3.87 for the
two bracketing calibrations and an average value of 3.9 was used for the flight analyzed here.
An orthogonal distance regression (ODR-2) of the volume from composition data (AMS mass
plus refractory black carbon) using a mass weighted density as described by Bahreini et al.
(Bahreini et al., 2009) versus the volume based on the sizing instruments (after correcting for
AMS lens transmission as above) had a slope of 1.06 for the entire SENEX study and 72% of
the data points were within the measurements' combined uncertainties of ±45% (Bahreini et al.,
2008). For the flight analyzed here, however, the same regression slope was 1.58, which is
slightly higher than the combined uncertainties. It is unclear why the two types of volume
measurements disagree more for this flight. This does not change the conclusions of this work
because this has been incorporated into the error in aerosol organic nitrate, which still show
positive enhancements in pRONO2 for these plumes (see Figure 4 below). These complete
error estimates are also used in Figure 5 to clearly show the uncertainties in the yields. The
volume comparison is discussed further in the Supplemental Information and shown for the
plumes of interest in Fig. S1.

The C-ToF-AMS is a unit mass resolution (UMR) instrument and the mass spectral signals that
are characteristic of aerosol nitrate at $m/z$ 30 and 46 (NO$^+$ and NO$_2^+$) often contain interferences
from organic species such as CH$_2$O$^+$ and CH$_2$O$_2^+$, respectively. Here, the $m/z$ 30 and 46 signals
have been corrected for these interferences by using correlated organic signals at $m/z$ 29, 42,
43, and 45 that were derived from high-resolution AMS measurements during the NASA
SEAC$^4$RS campaign that took place in the same regions of the SE US shortly after SENEX (see
Supplemental Information and Fig. S2). The corrections were applied to the individual flight
analyzed here from July 2. All of the corrections were well correlated with each other for the
SEAC$^4$RS dataset and we used the organic peak at $m/z$ 29 (from CHO$^+$) and the peak at $m/z$ 45
(from CHO$_2^+$), respectively, since those corrections were from peaks closest (in $m/z$) to those
being corrected. Once corrected, the nitrate mass concentrations in the final data archive for
this flight were reduced by 0-0.24 µg sm$^{-3}$, an average reduction of 0.11 µg sm$^{-3}$ or 32% from
the initial nitrate mass concentrations. The organic interferences removed from the $m/z$ 30 and
$m/z$ 46 signals are linearly correlated with the total organic mass concentrations, corresponding
to an average 1.3% increase in the total organic mass.

The ratio of the corrected NO$_2^+$/NO$^+$ signals was then used to calculate the fraction of aerosol
nitrate that was organic (pRONO$_2$) or inorganic (ammonium nitrate) based on the method
described first in (Fry et al., 2013). Here we used an organic NO$_2^+$/NO$^+$ ratio that was equal to
the ammonium nitrate NO$_2^+$/NO$^+$ ratio from our calibrations divided by 2.8. This factor was
determined from multiple datasets (see discussion in Supplemental Information). The
ammonium nitrate NO$_2^+$/NO$^+$ ratio was obtained from the two calibrations on 30 June and 7 July
that bracketed the flight on 2 July, which is analyzed here. It was 0.514 and 0.488, respectively,
and for all of the data from both calibrations it averaged 0.490. Hence, the organic nitrate
$NO_2^+/NO^+$ ratio was estimated to be 0.175. This is the first time, to our knowledge, that UMR
measurements of aerosol nitrate have been corrected with HR correlations and used to
apportion the corrected nitrate into inorganic or organic nitrate species.
The time since emission of intercepted power plant plumes was estimated from the slope of a
plot of $O_3$ against $NO_2$.  For nighttime emitted $NO_x$ plumes that consist primarily of NO (Peischl
et al., 2010), $O_3$ is negatively correlated with $NO_2$ due to the rapid reaction of NO with $O_3$ that
produces $NO_2$ in a 1:1 ratio:
$NO + O_3 \rightarrow NO_2 + O_2$ (R1)
Reaction R1 goes rapidly (NO pseudo first order loss rate coefficient of 0.03 $s^{-1}$ at 60 ppb $O_3$) to
completion, so that all $NO_x$ is present as $NO_2$, as long as the plume NO does not exceed
background $O_3$ after initial mixing of the plume into background air.  Subsequent oxidation of
$NO_2$ via reaction (R2) leads to an increasingly negative slope of $O_3$ vs $NO_2$:
$NO_2 + O_3 \rightarrow NO_3 + O_2$ (R2)
Equation (1) then gives plume age subsequent to the completion of (R1) in terms of the
observed slope, $m$, of $O_3$ vs $NO_2$ (Brown et al., 2006).
$$t_{plume} = \frac{ln[1-S(m+1)]}{Sk_1\overline{O_3}}$$ (1)
Here $S$ is a stoichiometric factor that is chosen for this analysis to be 1 based on agreement of
plume age with elapsed time in a box model run initialized with SENEX flight conditions (see
below); $k_1$ is the temperature dependent bimolecular rate constant for $NO_2 + O_3$ (R2) and $\overline{O_3}$ is
the average $O_3$ within the plume.
We calculate plume ages using both a stoichiometric factor of 1 (loss of $NO_3$ and $N_2O_5$
dominated by $NO_3$ reactions) and 2 (loss dominated by $N_2O_5$ reactions), although we note that
the chemical regime for $NO_3+N_2O_5$ loss may change over the lifetime of the plume, progressing
from 1 to 2 as the BVOC is consumed. We use $S$=1 values in the analysis that follows. Because
the more aged plumes are more likely to have $S$ approach 2, this means that some of the older
plumes may have overestimated ages. Fig. S3 in the Supplemental Information shows the
plume age calculated by Eq. 1 using modeled $NO_x$, $NO_y$ and $O_3$ concentrations for $S$=1 and
$S$=2, from nighttime simulations of plume evolution using an observationally constrained box
model. This confirms that for nighttime plumes, $S$=1 plume ages match modeled elapsed time
well. The model used for this calculation, and those used to assess peroxy radical lifetimes and
fates in Section 4.3, was the Dynamically Simple Model of Atmospheric Chemical Complexity
(DSMACC (Emmerson and Evans, 2009)) containing the Master Chemical Mechanism v3.3.1
chemistry scheme (Jenkin et al., 2015). More details on the model approach are provided in the
SI.

## 3 Nighttime flight selection

There were three nighttime flights (takeoffs on the evenings of 19 June, 2 July, and 3 July, 2013, local time) conducted during SENEX, of which one (2 July) surveyed regions surrounding Birmingham, Alabama, including multiple urban and power plant plume transects. As described in the introduction, these plume transects are the focus of the current analysis since they correspond to injections of concentrated NO (and subsequently high $P(NO_3)$) into the regionally widespread residual layer isoprene. The nighttime flight on 3 July, over Missouri, Tennessee and Arkansas sampled air more heavily influenced by biomass burning than biogenic emissions. The 19 June night flight sampled earlier in the evening, in the few hours immediately after sunset, and sampled more diffuse urban plume transects that had less contrast with background air. Therefore, this paper uses data exclusively from the 2 July flight, in which 9 transects of well-defined $NO_x$ plumes from power plants emitted during darkness can be analyzed to obtain independent yields measurements.

A map of the 2 July flight track is shown in Fig. 1a. After takeoff at 8:08 pm local Central Daylight Time on 2 July, 2013 (1:08 am UTC 3 July, 2016), the flight proceeded towards the southwest until due west of Montgomery, AL, after which it conducted a series of east-west running tracks while working successively north toward Birmingham, AL. Toward the east of Birmingham, the aircraft executed overlapping north-south tracks at six elevations to sample the E. C. Gaston power plant. During the course of the flight, concentrated $NO_x$ plumes from the Gaston, Gorgas, Miller and Greene City power plants were sampled. Around 1:30 and 2:30 AM Central Daylight Time (5:30 and 6:30 am UTC), two transects of the Birmingham, AL urban plume were measured prior to returning to the Smyrna, TN airport base.

The flight track is shown colored by the nitrate radical production rate, $P(NO_3)$, to show the points of urban and/or power plant plume influence:

$$P(NO_3) = k_2(T) \, [NO_2][O_3] \tag{2}$$

Here, $k_2$ is again the temperature-dependent rate coefficient for reaction of $NO_2 + O_3$ (Atkinson et al., 2004), and the square brackets indicate concentrations. Fig. 1b further illustrates the selection of power plants plumes: sharp peaks in $P(NO_3)$ are indicative of power plant plume transects, during which isoprene mixing ratios also are observed to drop from the typical regional residual layer background values of ~ 1 ppb, indicative of loss by $NO_3$ oxidation (an individual transect is shown in more detail below in Fig. 2). Also shown in Fig. 1b are measured concentrations of isoprene and monoterpenes throughout the flight, showing substantial residual layer isoprene and supporting the assumption that effectively all $NO_3$ reactivity is via isoprene (see calculation in next section). Residual layer concentrations of other VOCs that could produce SOA (e.g., aromatics) are always below 100 pptv, and their reaction rates with $NO_3$ are slow. Edwards *et al*. (Edwards et al., 2017) have shown that $NO_3$ and isoprene mixing ratios for this and other SENEX night flights exhibit a strong and characteristic anticorrelation that is consistent with nighttime residual layer oxidation chemistry.

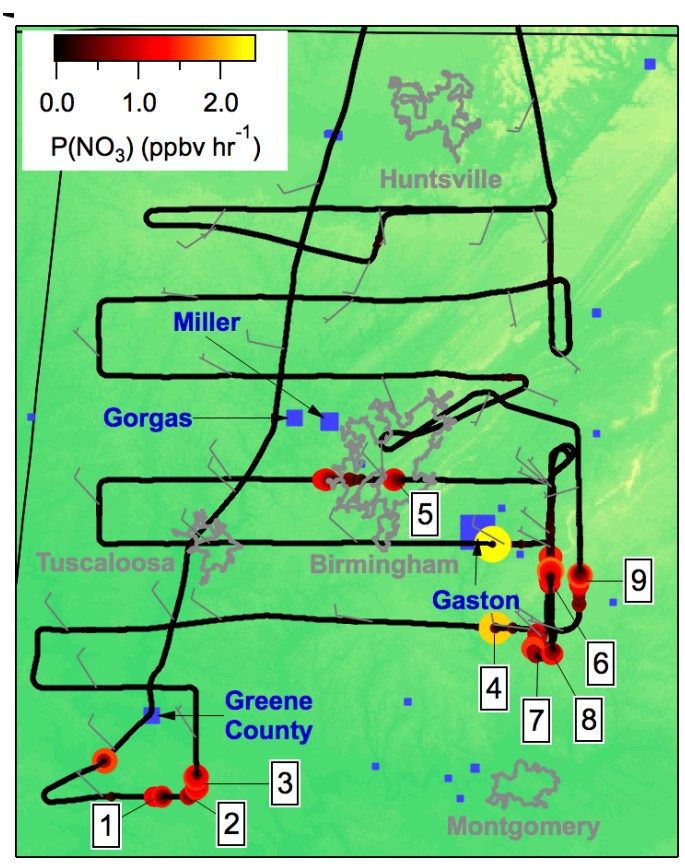

1a

1b

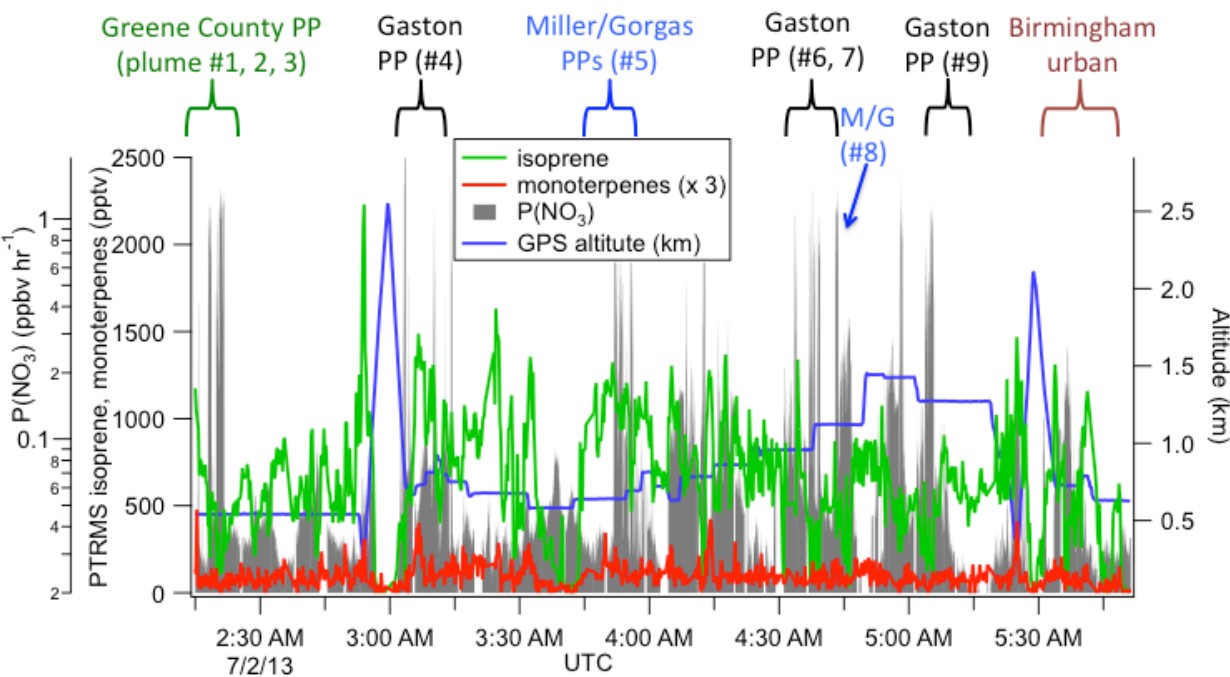

**Figure 1a.** Map of northern Alabama, showing the location of the flight track of the 2 July 2013 night flight used in the present analysis, with plume numbers labeled and wind direction shown. Although the wind direction changed throughout the night, these measurements enable us to

attribute each plume to a power plant source (see labels in Figure 1b and Table 2). Color scale
shows $P(NO_3)$ based on aircraft-measured $[NO_2]$ and $[O_3]$, while power plants discussed in the
text are indicated in blue squares with marker size scaled to annual $NO_x$ emissions for 2013
(scale not shown). Isoprene emissions are widespread in the region (Edwards et al., 2017).
**Figure 1b** shows time series data from the same flight, with plume origins and numbers labeled,
showing aircraft-measured isoprene and monoterpene concentrations, altitude, and $P(NO_3)$
determined according to Eq. 2 (log scale), showing that the isoprene was uniformly distributed
(mixing ratios often in excess of 1 ppbv), while the more reactive monoterpenes were present at
mixing ratios below 100 ppt except at the lowest few hundred meters above ground in the
vertical profiles (not used in the present analysis). Figure 1b also shows that sharp peaks in
nitrate radical production rate occur both at the lowest points of these vertical profiles, when the
aircraft approached the surface, but also frequently during periods of level flight in the residual
layer, which correspond to the power plant plume transects analyzed in this paper.

## 453    4 Results

### 454    4.1 Selection of plumes

Figure 2 shows a subset of the July 2 flight time series data, illustrating three $NO_x$ plumes used
for analysis. The large $NO_3$ source and isoprene loss was accompanied by an increase in
organic nitrate aerosol mass, which we attribute to the $NO_3$ + isoprene reaction based on prior
arguments. We observed each plume as a rapid and brief perturbation to background
conditions, of order 10 – 50 sec., or 1 – 5 km in spatial scale.  Each plume's perturbed
conditions can correspond to different plume ages, depending on how far downwind of the
power plant the plume transect occurred.

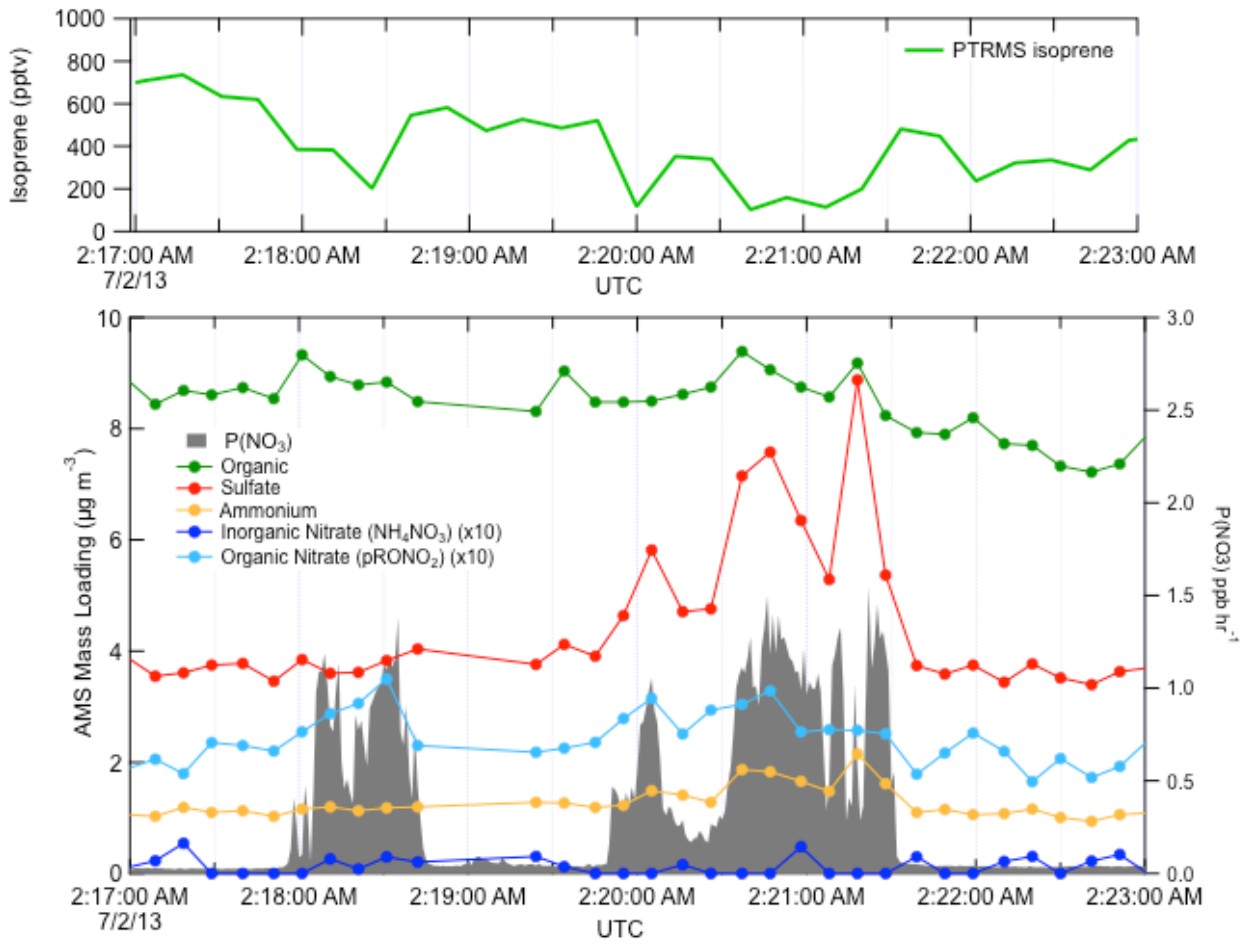

**Figure 2.** Three representative plume transect observations from the 2 July 2013 flight (plumes
are identified by the peaks in $P(NO_3)$, listed in Table 1 at times 02:18, 02:20, and 02:21 UTC).
Note the difference in sulfate enhancement in the three plumes, which is largest in the third
plume, and is accompanied by increases in ammonium. In all three cases, the isoprene
concentration drops in the plumes, accompanied by a clear increase in organic nitrate, no
changes in the inorganic nitrate, and a modest changes in organic aerosol mass concentrations.
Candidate plumes were initially identified by scanning the time series flight data for any period
where the production rate of nitrate radical ($P(NO_3)$) rose above 0.5 ppbv hr$^{-1}$. This threshold
was chosen to be above background noise and large enough to isolate only true plumes (see
Fig. 1a). The value is thus subjectively chosen, but was consistently applied across the dataset.
For each such period, a first screening removed any of these candidate plumes that occurred
during missed approaches or other periods where radar altitude above ground level (AGL) was
changing, because in the stratified nighttime boundary layer structure, variations in altitude may
result in sampling different air-masses, rendering the adjacent out of plume background not
necessarily comparable to in-plume conditions. A second criterion for rejection of a plume was
missing isoprene or AMS data during brief plume intercepts. No selected plumes on July 2
showed enhanced acetonitrile or refractory black carbon, indicating no significant biomass
burning influence.  Finally, two plumes downwind of the Gaston power plant (at 03:10 and
03:14) were removed from the present analysis, because (03:10) the background isoprene was
changing rapidly, preventing a good baseline measurement, and (03:14) there was no observed
decrease in isoprene concentration in-plume (as well as no increase in nitrate aerosol). The
03:14 plume was apparently too recently emitted to have undergone significant nighttime
reaction; its $O_3/NO_2$ slope was unity to within the combined measurement error of $O_3$ and $NO_2$
(Eq. 1). After this filtering, there are 9 individual plume observations for determination of $NO_3$ +
isoprene SOA yields (see Table 1). The rapid increases in $P(NO_3)$ appeared simultaneously with
significant decreases in isoprene and increases in aerosol nitrate. The aerosol and isoprene
measurements (taken at data acquisition rates < 1 Hz) were not exactly coincident in time which
leads to some uncertainty in the yield analysis below.
Derivation of SOA yields from observed changes in isoprene and aerosol mass in plumes
depends on two conditions, and has several caveats that will be discussed in the text that
follows (see Table 3 below for a summary of these caveats). The two conditions are: (1) that the
majority of VOC mass consumed by $NO_3$ in plumes is isoprene (rather than monoterpenes or
other VOC), and then either or both (2a) that the change in aerosol organic mass concentration
during these plumes is due to $NO_3$ + isoprene reactions, and/or (2b) that the change in aerosol
nitrate mass concentration is due to $NO_3$ + isoprene reactions. There are separate
considerations for each of these conditions.
For the first condition, we note that the isoprene to monoterpenes ratio just outside each plume
transect was always high (a factor of 10 to 70, on average 26). With the 298 K $NO_3$ rate
constants of ~ 5 × $10^{-12}$ $cm^3$ $molec^{-1}$ $s^{-1}$ for monoterpenes and 6.5 × $10^{-13}$ $cm^3$ $molec^{-1}$ $s^{-1}$ for
isoprene (Calvert et al., 2000), isoprene  (~ 2 ppb) will always react faster with nitrate than
monoterpenes (~ 0.04 ppbv). At these relative concentrations, even if all of the monoterpene is
oxidized, the production rate of oxidation products will be much larger for isoprene. Contribution
to aerosol by $N_2O_5$ uptake is also not important in these plumes. Edwards et al. (Edwards et al.,
2017) calculated the sum of $NO_3$ and $N_2O_5$ loss throughout this flight and showed that it is
consistently $NO_3$+BVOC dominated (Fig. S4 of that paper). As isoprene depletes, $N_2O_5$ uptake
will increasingly contribute to $NO_3$ loss, but as shown below, we are able to rule out a
substantial source of inorganic nitrate for most plumes. We also know that despite increased
OH production in-plume, the isoprene loss is still overwhelming dominated by $NO_3$ (Fig. S5 in
Edwards, et al. (Edwards et al., 2017)) .
The second condition requires that we can find an aerosol signal that is attributable exclusively
to $NO_3$ + isoprene reaction products, whether it be organic aerosol (OA) or organic nitrate
aerosol (pRONO$_2$) mass loading, or both. We note that the ratio of in-plume aerosol organic
mass increase to pRONO$_2$ mass increase is noisy (see discussion below at Fig. 6), but indicates
an average in-plume ΔOA to ΔpRONO$_2$ ratio of about 5. The large variability is primarily due to
the fact that the variability in organic aerosol mass between successive 10-second data points
for the entire flight is quite large (of order 0.75 µg m$^{-3}$) and comparable to many of the individual
plume ΔOA increases, far exceeding the expected organonitrate driven increases in OA, which
are roughly twice the pRONO$_2$ mass increases. It is also possible that in these plumes, where
total aerosol mass is elevated, semivolatile organic compounds may re-partition to the aerosol
phase, contributing a non-pRONO$_2$ driven variability in ΔOA. For example, if some gas phase
IEPOX is present in the residual layer, it may be taken up into the highly acidic aerosol from the
power plants. Alternatively, very polar gas-phase compounds could partition further into the
higher liquid water associated with the sulfate in the plume. Therefore, in-plume organic aerosol
increases cannot be attributed clearly to NO$_3$ + isoprene SOA production, so we do not use
them in the SOA yield calculations.
This leaves consideration 2b, whether all increase in nitrate mass is due to NO$_3$ + isoprene
reactions. Here we must evaluate the possibility of inorganic nitrate aerosol production in these
high-NO$_x$ plumes. Fine-mode aerosol inorganic nitrate can be formed by the (reversible)
dissolution of HNO$_{3(g)}$ into aqueous aerosol. In dry aerosol samples, inorganic nitrate is typically
in the form of ammonium nitrate (NH$_4$NO$_3$), when excess ammonium is available after
neutralization of sulfate as (NH$_4$)$_2$SO$_4$ and NH$_4$(HSO$_4$). Because of the greater stability of
ammonium sulfate salt relative to ammonium nitrate, in high-sulfate plumes with limited
ammonium, inorganic nitrate aerosol will typically evaporate as HNO$_{3(g)}$ (Guo et al., 2015)
(reaction R3):
$2NH_4NO_{3(aq)} + H_2SO_{4(aq)} \rightleftarrows (NH_4)_2SO_{4(aq)} + 2HNO_{3(g)}$                     (R3)
Inorganic nitrate can also form when crustal dust (e.g. CaCO$_3$) or seasalt (NaCl) are available.
Uptake of HNO$_3$ is rendered favorable by the higher stability of nitrate mineral salts, evaporating
CO$_2$ or HCl. Inorganic nitrate can also be produced by the heterogeneous uptake of N$_2$O$_5$ onto
aqueous aerosol; Edwards et al. (2017) demonstrated that this process is negligible relative to
NO$_3$ + BVOC for the July 2 SENEX night flight considered here.
There are several lines of evidence that the observed nitrate aerosol is organic and not
inorganic. First, examination of the NO$_2^+$/NO$^+$ (interference-corrected *m/z* 46:*m/z* 30) ratio
measured by the aircraft AMS (Fig. 3) shows a ratio throughout the July 2 flight, including the
selected plumes, that is substantially lower than that from the bracketing ammonium nitrate
calibrations. This lower AMS measured NO$_2^+$/NO$^+$ ratio has been observed for organic nitrates
(Farmer et al., 2010), and some mineral nitrates (e.g. Ca(NO$_3$)$_2$ and NaNO$_3$, (Hayes et al.,
2013)), which are not important in this case because aerosol was dominantly submicron. As
described above, we can separate the observed AMS nitrate signal into pRONO$_2$ and inorganic
nitrate contributions. These mass loadings are also shown in Fig. 3, indicating dominance of
pRONO$_2$ throughout the flight.

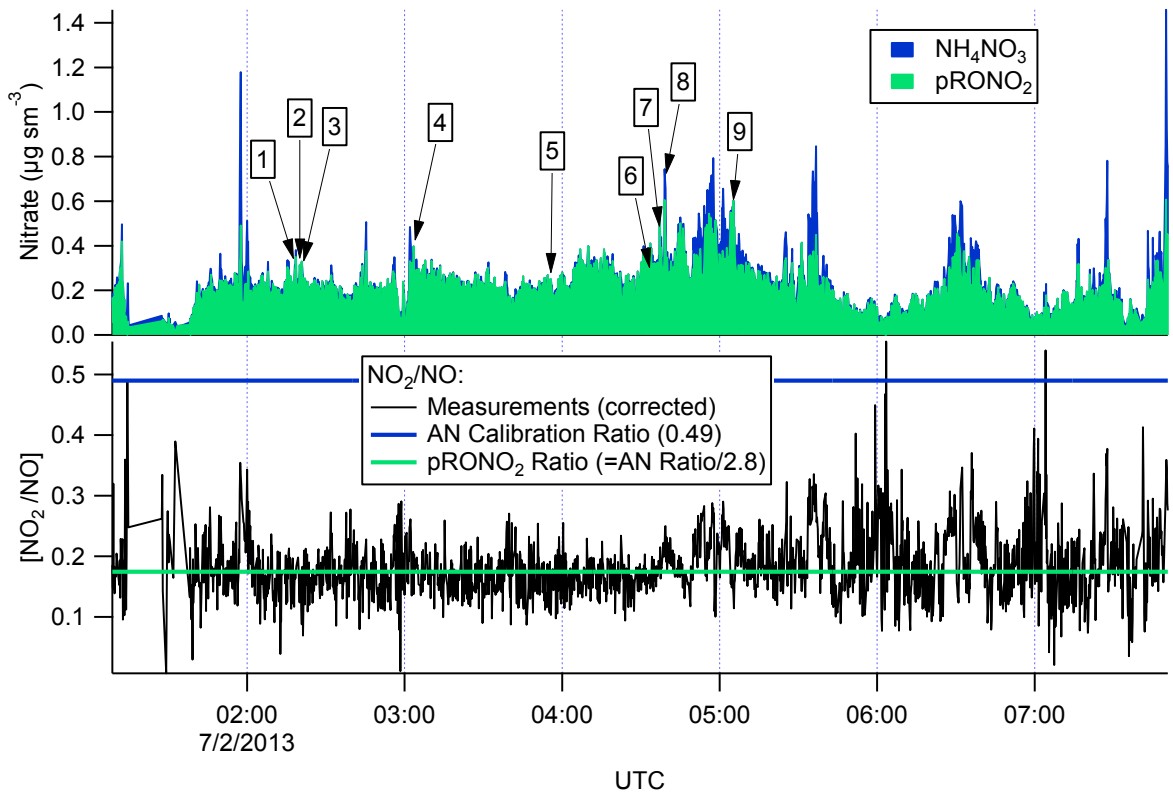

**Figure 3.** For the flight under consideration, the estimated relative contributions of ammonium
and organic nitrate to the total corrected nitrate signal (top panel) was calculated from the ratios
of the corrected peaks at *m/z* 30 and 46 (lower panel). Each of the plumes is identified here by
plume number. The ratios of $NO_2^+/NO^+$ (black data in the lower panel) from the corrected peaks
at *m/z* 46 and 30, respectively, are compared to the ratios expected for ammonium nitrate (AN
Calibration Ratio, blue horizontal line at 0.49) or organic nitrate (pRONO$_2$ Ratio, green
horizontal line at 0.175 which is estimated from the AN calibration ratio using multiple data sets
(see discussion in Supplemental Information). The measured ratio for most of the flight is more
characteristic of organic nitrate than ammonium nitrate.
We can also employ the comparison of other AMS-measured aerosol components during the
individual plumes to assess the possibility of an inorganic nitrate contribution to total measured
nitrate. Fig. S5a shows that the in-plume increases in sulfate are correlated with increases in
ammonium with an $R^2$ of 0.4. The observed slope of 5.4 is characteristic of primarily $(NH_4)HSO_4$,
which indicates that the sulfate mass is not fully neutralized by ammonium. We note, however,
that if the largest observed aerosol nitrate increase is due solely to ammonium nitrate, the
ammonium increase would be only 0.11 µg m$^{-3}$, which would be difficult to discern from the $NH_4$
variability of order 0.11 µg m$^{-3}$. However, the slope is consistent with incomplete neutralization
of the sulfate by ammonium, which would make $HNO_{3(g)}$ the more thermodynamically favorable
form of inorganic nitrate. The ion balance for the ammonium nitrate calibration particles and the
plume enhancements are shown in Fig. S5b. Complete neutralization of the calibration aerosols
is nearly always within the gray 10% uncertainty band for the relative ionization efficiency of
ammonium (Bahreini et al., 2009). In contrast, many of the plume enhancements are near the
1:2 line (as primarily ammonium bisulfate) within the combined 10% ammonium and 15%
sulfate uncertainty error bars or without ammonium (sulfuric acid). Thus, $NH_4NO_3$ is unlikely to
be stable in the aerosol phase under the conditions of these plumes, consistent with the AMS
observations.
A plot of the calculated plume enhancements from the derived apportionment into organic
($pRONO_2$) and inorganic (ammonium) nitrate is shown in Fig. 4. The increases in aerosol nitrate
for nearly all of the plumes appear to be mostly due to enhancements in pRONO2. Based on
these considerations, we conclude that in-plume $pRONO_2$ mass increases are a consequence
(and thus a robust measure) of organic nitrate aerosol produced from $NO_3$ + isoprene. Since
each isoprene molecule condensing will have one nitrate group, the ratio of these increases to
isoprene loss is a direct measure of the molar organic aerosol yield from $NO_3$-isoprene
oxidation.

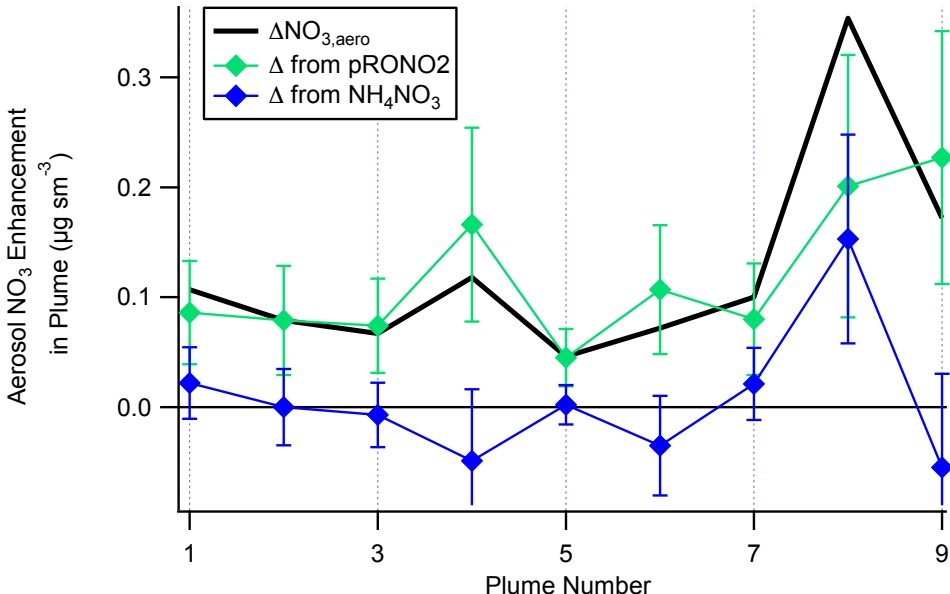



**Figure 4.** The contribution of each species to the nitrate enhancements in each of the plumes,
showing that the enhancements in most of the plumes are mainly due to enhancements in
organic nitrate, with the exception of Plume 8 which had enhancements in both organic and
ammonium nitrate. Error bars are estimated from the measurement variability, the UMR
corrections to the nitrate signals, apportionment between organic and inorganic nitrate, and the
total nitrate uncertainty (see Supplemental Information).

Table 1 shows the selected plumes to be used for yield analysis. Wherever possible, multiple
points have been averaged for in-plume and background isoprene and nitrate aerosol
concentrations; in each case the number of points used is indicated and the corresponding
standard deviations are reported. In two cases (2:20 and 3:03 plumes), the plumes were so
narrow that only a single point was measured in-plume at the 10 s time resolution of the PTR-
MS and AMS; for these "single-point" plumes it is not possible to calculate error bars. Error bars
were determined using the standard deviations calculated for in-plume and background
isoprene and nitrate aerosol concentrations, accounting also for the additional uncertainty in the
AMS measurement described in the caption to Figure 4, and propagated through the yield
formula detailed in the following section.

**Table 1.** List of plumes used in this $NO_3$ + isoprene SOA yield analysis. For each plume, the
delta-values listed indicate the difference between in-plume and outside-plume background in
average observed concentration, and the standard deviations (SD) are the propagated error
from this subtraction. (For $\Delta NO_3$ from $pRONO_2$, the standard deviations also include error
propagated as described in the caption for Figure 4) After each plume number, the numbers of
points averaged for isoprene (10 s resolution) and AMS (10 s resolution), respectively, are
listed. Because the isoprene data were reported at a lower frequency, these numbers are
typically lower to cover the same period of time. Plume numbers annotated with * indicate brief
plumes for which only single-point measurements of in-plume aerosol composition were
possible. Additional AMS and auxiliary data from each plume is included in the Supplemental
Information, Table S3.

| plume number [#isop/#AMS] | 7/2/13 plume time (UTC) | P($NO_3$) (ppbv hr$^{-1}$) | ΔISOP (ppt) [± SD] | $\Delta NO_{3,aero}$ (µg m$^{-3}$) [± SD] | $\Delta NO_3$ from $pRONO_2$ (µg m$^{-3}$) [± SD] | $\Delta NO_3$ from $NH_4NO_3$ (µg m$^{-3}$) [± SD] |
|---|---|---|---|---|---|---|
| Typical variability (µg m$^{-3}$): | | | | 0.05 | 0.05 | 0.05 |
| 1 [2/3] | 2:18 | 0.9 | -335 [128] | 0.107 [0.039] | 0.086 [0.047] | 0.022 [0.012] |
| 2 [*] | 2:20 | 0.8 | -404 | 0.079 | 0.079 [0.049] | 0 |
| 3 [4/5] | 2:21 | 1.2 | -228 [121] | 0.067 [0.039] | 0.074 [0.043] | -0.007 [0.027] |
| 4 [*] | 3:03 | 1.4 | -453 | 0.118 | 0.166 [0.088] | -0.049 |
| 5** [3/4] | 3:55 | 1.0 | -255 [251] | 0.046 [0.019] | 0.045 [0.026] | 0.002 [0.015] |
| 6 [2/2] | 4:34 | 0.6 | -713 [219] | 0.072 [0.031] | 0.107 [0.059] | -0.035 [0.029] |
| 7 [5/6] | 4:37 | 0.8 | -298 [197] | 0.100 [0.082] | 0.080 [0.051] | 0.021 [0.034] |
| 8*** [2/3] | 4:39 | 0.9 | -443 [75] | 0.354 [0.058] | 0.201 [0.120] | 0.153 [0.057] |
| 9 [7/8] | 5:04 | 0.6 | -293 [131] | 0.172 [0.048] | 0.227 [0.115] | -0.055 [0.042] |

**Plume 5 has the smallest $\Delta NO_{3,aero}$ and may be affected by background $pRONO_2$ variability.
***Plume 8 has a measurable increase in inorganic nitrate as well as organic.

## 4.2 SOA yield analysis

A **molar** SOA yield refers to the number of molecules of aerosol organic nitrate produced per
molecule of isoprene consumed. In order to determine molar SOA yields from the data
presented in Table 1, we convert the aerosol organic nitrate mass loading differences to mixing
ratio differences (ppt) using the $NO_3$ molecular weight of 62 g mol$^{-1}$ (the AMS organic nitrate
mass is the mass only of the –ONO$_2$ portion of the organonitrate aerosol). At standard
conditions of 273 K and 1 atm (all aerosol data are reported with this STP definition), 1000 ppt
NO$_3$ = 2.77 μg m$^{-3}$, so each $\Delta M_{pRONO2}$ is multiplied by 361 ppt (μg m$^{-3}$)$^{-1}$ to determine this molar
yield:
$$Y_{SOA,molar} = \frac{(pRONO2_{plume} \pm SD_{pRONO2plume}) - (pRONO2_{bkg} \pm SD_{pRONO2bkg})}{-[(isop_{plume} \pm SD_{isopplume}) - (isop_{bkg} \pm SD_{isopbkg})]} \times \frac{361 \, ppt \, NO_3}{\mu g \, m^{-3}} \tag{3}$$
The SOA molar yields resulting from this calculation are shown in Table 2, spanning a range of
5-28%, with uncertainties indicated based on the SDs in measured AMS and isoprene
concentrations. In addition to this uncertainty based on measurement precision and ambient
variability, there is an uncertainty of 50% in the AMS derived-organic nitrate mass loadings (see
SI) and 25% in the PTR-MS isoprene concentrations (Warneke et al., 2016). The average molar
pRONO$_2$ yield across all plumes, with each point weighed by the inverse of its standard
deviation, is 9%. (As noted below, the yield appears to increase with plume age, so this average
obscures that trend.) An alternate graphical analysis of molar SOA yield from all nine plumes
plus one 'null' plume (03:14, in which no isoprene had yet reacted and thus not included in
Tables 1 and 2) obtains the same average molar yield of 9% (Fig. 5). Here, the molar yield is
the slope of a plot of plume change in pRONO$_2$ vs plume change in isoprene. The slope is
determined by a linear fit with points weighted by the square root of the number of AMS data
points used to determine in-plume pRONO$_2$ in each case. This slope error gives a rather narrow
uncertainty range for the slope (0.0930 +/- 0.0011); to obtain an upper limit in the uncertainty of
this molar yield we apply the combined instrumental uncertainties, based on adding in
quadrature the PTR-MS uncertainty of 5% and the AMS uncertainty of 50%. This gives an
overall uncertainty of 50.2%, resulting in upper and lower limit slopes of 0.140 and 0.046,
respectively; we use this maximum uncertainty estimate to report the average molar yield as 9%
(+/- 5%). We have not corrected the calculated yields for the possibility of NO$_3$ heterogeneous
uptake, which could add a nitrate functionality to existing aerosol. Such a process could be rapid
if the uptake coefficient for NO$_3$ were 0.1, a value characteristics of unsaturated substrates (Ng
et al., 2017), but would not contribute measurably at more conventional NO$_3$ uptake coefficients
of 0.001 (Brown and Stutz, 2012).

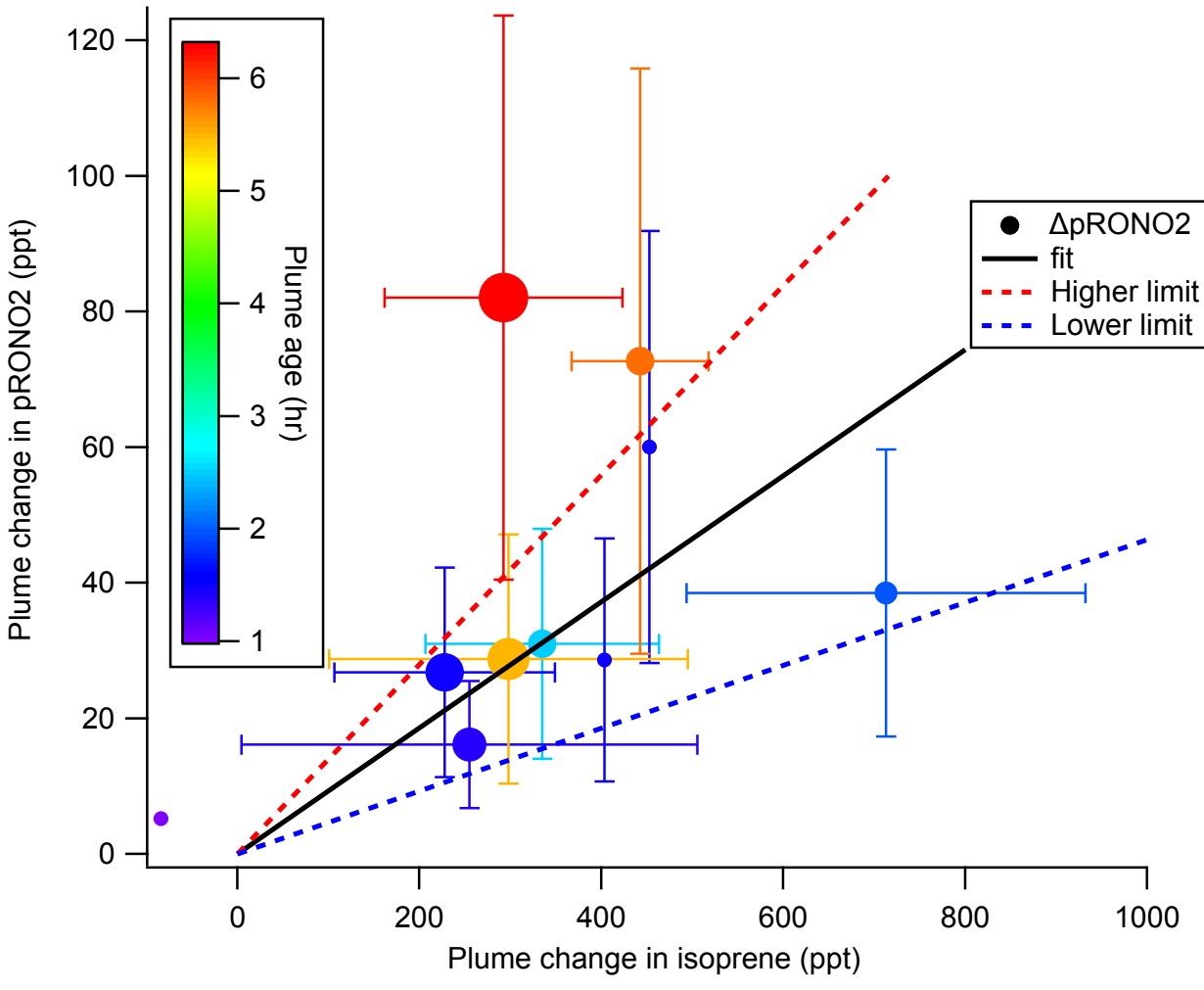

**Figure 5.** SOA molar yield can be determined as the slope of $\Delta pRONO_2$ vs. $\Delta$isoprene, both in mixing ratio units. The linear fit is weighted by square root of number of points used to determine each in-plume $pRONO_2$, with intercept held at zero. The slope coefficient ± one standard deviation is 0.0930 ± 0.0011. Larger "outside" high and low limits of the slope (shown as dashed red and blue lines) are obtained by adding and subtracting from this slope the combined instrumental uncertainties, based on adding in quadrature the PTR-MS uncertainty of 5% and the AMS uncertainty of 50%. This gives an overall uncertainty of 50.2%, resulting in upper and lower limit slopes of 0.140 and 0.046, respectively. Points are colored by plume age, and size scaled by square root of number of points (the point weight used in linear fit). This plot and fit includes the nine plumes listed in Tables 1 and 2, as well as the 03:14 "unreacted" plume (at $\Delta$isoprene = -84 ppt). Error bars on isoprene are the propagated standard deviations of the (in plume - out plume) differences, for plumes in which multi-point averages were possible. Error bars on pRONO2 are the same as in Figure 4, converted to ppt. The points without error bars are single-point plumes.

To estimate SOA **mass** yields, we need to make some assumption about the mass of the
organic molecules containing the nitrate groups that lead to the observed nitrate aerosol mass
increase. The observed changes in organic aerosol are too variable to be simply interpreted as
the organic portion of the aerosol organic nitrate molecules. We conservatively assume the
organic mass to be approximately double the nitrate mass (62 g mol$^{-1}$), based on an "average"
molecular structure of an isoprene nitrate with 3 additional oxygens: e.g. a tri-hydroxynitrate
(with organic portion of formula $C_5H_{11}O_3$, 119 g mol$^{-1}$), consistent with 2$^{nd}$-generation oxidation
product structures suggested in Schwantes, et al. (Schwantes et al., 2015). Based on this
assumed organic to nitrate ratio, all plumes' expected organic mass increases would be less
than the typical variability in organic of 0.75 μg m$^{-3}$. This assumed structure is consistent with
oxidation of both double bonds, which appears to be necessary for substantial condensation of
isoprene products, and which structures would have calculated vapor pressures sufficiently low
to partition to the aerosol phase (Rollins et al., 2009). Another possible route to low vapor
pressure products is intramolecular H rearragement reactions, discussed below in Section 4.3,
which would not require oxidant reactions at both double bonds. In the case of oxidant reactions
at both double bonds, it is difficult to understand how the second double bond would be oxidized
unless by another nitrate radical, which would halve these assumed organic to nitrate ratios
(assuming the nitrate is retained in the molecules). On the other hand, any organic nitrate
aerosol may lose $NO_3$ moieties, increasing the organic to nitrate ratio. Given these uncertainties
in both directions, we use the assumed "average" structure above to guess an associated
organic mass of double the nitrate mass. Thus, to estimate SOA mass yield, we multiply the
increase in organic nitrate aerosol mass concentration by three (i.e., $2 \times \Delta M_{pRONO2} + \Delta M_{pRONO2}$),
and divide by the observed decrease in isoprene, converted to μg m$^{-3}$ by multiplying by 329 ppt
(μg m$^{-3}$)$^{-1}$, the conversion factor based on isoprene's molecular weight of 68.12 g mol$^{-1}$.

$$Y_{SOA,mass} = \frac{(pRONO2_{plume} \pm SD_{pRONO2plume}) - (pRONO2_{bkg} \pm SD_{pRONO2bkg})}{-[(isop_{plume} \pm SD_{isopplume}) - (isop_{bkg} \pm SD_{isopbkg})]} \times 3 \times \frac{329ppt}{\mu g \; m^{-3}} \qquad (4)$$

Note that the SOA mass yield reported here is based on the (assumed) mass of organic aerosol
plus the (organo)nitrate aerosol formed in each plume. If instead the yield were calculated using
only the assumed increase in *organic* mass (i.e., 2x $\Delta M_{pRONO2}$ instead of 3x $\Delta M_{pRONO2}$), which
would be consistent with the method used in Rollins, et al. (Rollins et al., 2009) and Brown et al.
(Brown et al., 2009), the mass yields would be 2/3 the values reported here. However, since
SOA mass yield is typically defined based on the total increase in aerosol mass, we use the
definition with the sum of the organic and nitrate mass here. This results in an average SOA
mass yield of 27%, with propagated instrumental errors (see caption to Fig. 5) giving a range of
723     27% +/- 14%.

We note also that correlation of in-plume increases in OA with pRONO$_2$ (Fig. 6) point to a
substantially larger 5:1 organic-to-nitrate ratio; if this were interpreted as indicating that the
average molecular formula of the condensing organic nitrate has 5 times the organic mass as
nitrate, this would increase the SOA mass yields reported here. However, due to the
aforementioned possibility of additional sources of co-condensing organic aerosol, which led us
to avoid using ΔOA in determining SOA yields, we do not consider this to be a direct indication
of the molecular formula of the condensing organic nitrate. Including OA in the SOA yield
determination, based on this 5:1 slope rather than the assumed 2:1 OA:pRONO$_2$, would give
2.5 times larger SOA mass yields than reported here.

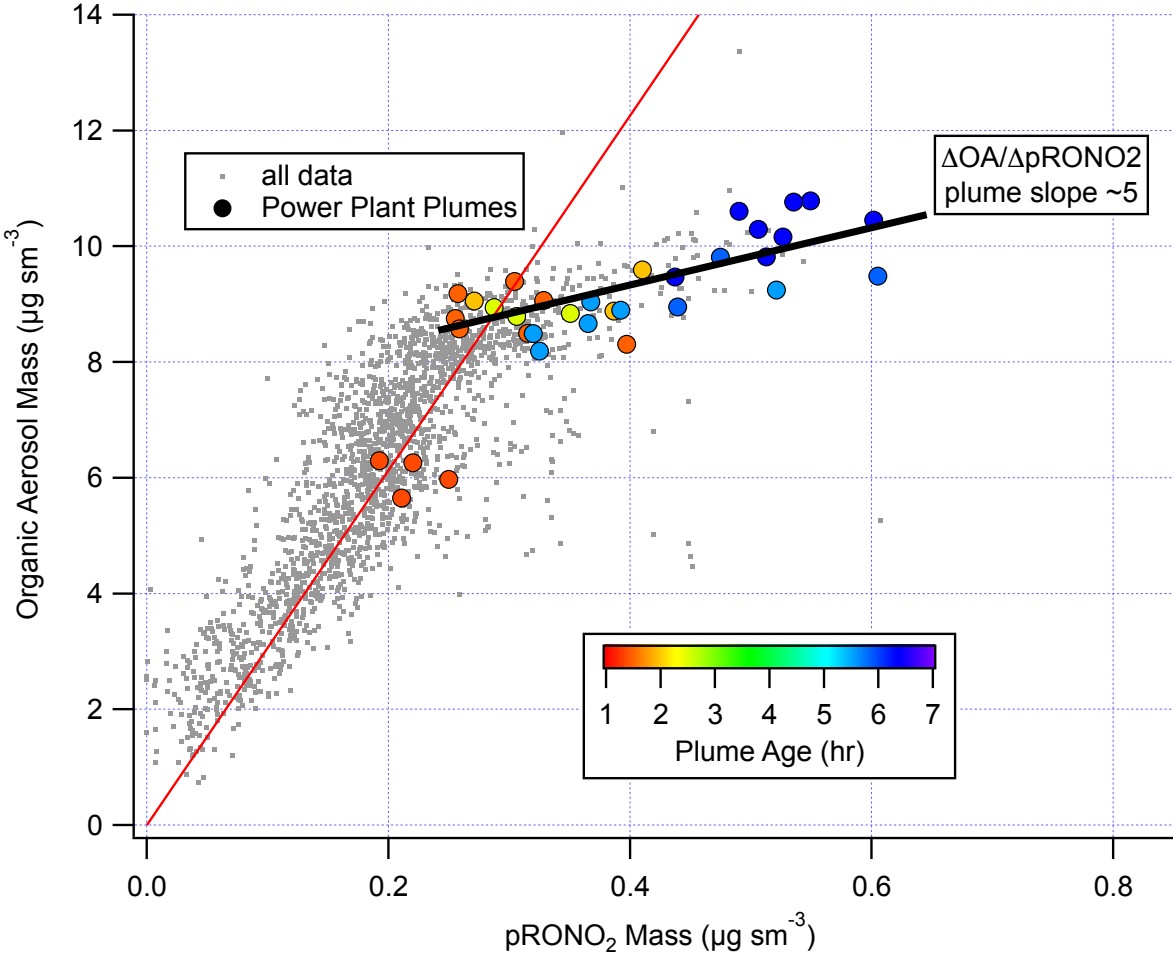

**Figure 6**. Correlation of organic aerosol mass concentration with pRONO$_2$ mass concentration
for the full 2 July flight (grey points and red fit line, fitted slope and thus average OA/pRONO$_2$
mass ratio of ~30) and for the points during the selected plumes (colored points, colored by
plume age, average OA/pRONO$_2$ mass ratio of ~ 5).

**Table 2.** SOA Yields for each plume observation, estimated plume age, and likely origin. See
text for description of uncertainty estimates. For the mass yields, the calculated SOA mass
increase includes both the organic and (organo)nitrate aerosol mass; the measurements for OA
increases shown in Figure 6 do not include the nitrate mass.

| plume number | plume time (UTC) | SOA molar yield (fraction) [± SD] | SOA mass yield (fraction) [± SD] | plume age from $O_3$/ $NO_2$ clock assuming S=1 (hours) | Likely NOx origin & altitude (m) |
|---|---|---|---|---|---|
| 1 | 7/2/13 2:18 | 0.09 [0.06] | 0.25 [0.17] | 2.5 | Greene County @ 540 m |
| 2 | 7/2/13 2:20 | 0.07 | 0.21 | 1.5 | *ibid* |
| 3 | 7/2/13 2:21 | 0.12 [0.10] | 0.32 [0.25] | 1.5 | *ibid* |
| 4 | 7/2/13 3:03 | 0.13 | 0.36 | 1.5 | Gaston @ 720 m |
| 5 | 7/2/13 3:55 | 0.06 [0.07] | 0.17 [0.20] | 1.4 | Miller / Gorgas @ 690 m |
| 6 | 7/2/13 4:34 | 0.05 [0.03] | 0.15 [0.09] | 2 | *ibid* |
| 7 | 7/2/13 4:37 | 0.10 [0.09] | 0.26 [0.24] | 5.5 | *ibid* |
| 8 | 7/2/13 4:39 | 0.16 [0.10] | 0.45 [0.28] | 5.8 | Miller / Gorgas @ 1120 m |
| 9 | 7/2/13 5:04 | 0.28 [0.19] | 0.77 [0.52] | 6.3 | Gaston @ 1280 m |


**Table 3.** Several caveats to the present SOA yields analysis are listed below, alongside the
expected direction each would adjust the estimated yields. Because we do not know whether or
how much each process may have occurred in the studied plumes, we cannot quantitatively
assess the resulting uncertainties, so we simply list them here. See text above for more detailed
discussion.

| Process | Effect on determined SOA yield |
|---|---|
| Organic nitrate aerosol loses $NO_3$ functional group | Larger, because the non-nitrate OA would not be counted in this analysis |
| Both double bonds in isoprene are oxidized by $NO_3$: two nitrates per condensing molecule | Smaller, because the assumed organic to nitrate mass ratio assumes one nitrate per molecule |
| $NO_3$ oxidizes daytime isoprene oxidation products (e.g. ISOPOOH) to make new aerosol | Smaller, because this would produce organic nitrate aerosol without corresponding decrease in isoprene, so that some of existing SOA production is mis-attributed to isoprene + $NO_3$ |
| Assumed organic to nitrate mass ratio is incorrect | Unknown direction of effect, depends on whether assumed ratio is high or low |
| Daytime-produced IEPOX uptake onto acidic particles | No effect (only changes ΔOA, not nitrate) |
| Suppression of $O_3$ + monoterpene or $O_3$ + isoprene SOA in plumes | No effect (only changes ΔOA, not nitrate) |

Finally, the large range in observed yields can be interpreted by examining the relationship to
estimated plume age. Using the slope of $O_3$ to $NO_2$ (Eq. 1) to estimate plume age as described
above, a weak positive correlation is observed (Table 2, Fig. S4), suggesting that as the plume
ages, later-generation chemistry results in greater partitioning to the condensed phase of $NO_3$ +
isoprene organonitrate aerosol products. This is consistent with the observation by Rollins et al.
(Rollins et al., 2009) that 2nd-generation oxidation produced substantially higher SOA yields
than the oxidation of the first double bond alone, but we note that these mass yields (averaging
27%, would be 18% using the organic mass only) are higher than even the largest yield found in
that chamber study (14%, used organic mass only).
We observe increasing SOA yield, from a molar yield of around 10% at 1.5 hours up to 30% at 6
hours of aging. The lowest yields observed are found in the most recently emitted plumes,
suggesting the interpretation of the higher yields as a consequence of longer aging timescales
in the atmosphere.
## 4.3 Mechanistic considerations
These larger SOA mass yields from field determinations (average 27%) relative to chamber
work (12 – 14%, see introduction) may arise for several reasons. We first assess the volatility of
assumed first- and second-generation products using group contribution theory in order to
predict partitioning. After a single oxidation step, with a representative product assumed to be a
$C_5$ hydroperoxynitrate, the saturation vapor pressure estimated by group contribution theory
(Pankow and Asher, 2008) at 283 K would be $2.10 \times 10^{-3}$ Torr ($C^* = 1.7 \times 10^4$ µg m$^{-3}$ for MW =
147 g mol$^{-1}$), while a double-oxidized isoprene molecule (assuming a $C_5$ dihydroxy dinitrate) has
an estimated vapor pressure of $7.95 \times 10^{-8}$ Torr ($C^* = 1.01$ µg m$^{-3}$ for MW = 226 g mol$^{-1}$). This
supports the conclusion that while the first oxidation step produces compounds too volatile to
contribute appreciably to aerosol formation, oxidizing both double bonds of the isoprene
molecule is sufficient to produce substantial partitioning, consistent with Rollins et al. (Rollins et
al., 2009). This is also true if the second double bond is not oxidized by nitrate (group
contribution estimate $P_{vap}$ for a $C_5$ tri-hydroxy nitrate is $7.7 \times 10^{-8}$ Torr, $C^* = 0.79$ µg m$^{-3}$ for MW =
181 g mol$^{-1}$). These $C^*$ saturation concentration values suggest that no dimer formation or
oligomerization is *required* to produce low-enough volatility products to condense to the aerosol
phase; however, such oligomerization would result in more efficient condensation. The fact that
Rollins et al. (Rollins et al., 2009) did not observe larger mass yields may indicate that it takes
longer than a typical chamber experiment timescale to reach equilibrium, or that this absorptive
partitioning model did not accurately capture those experiments, or that substantial loss of
semivolatiles to the chamber walls (e.g. (Krechmer et al., 2016)) suppressed apparent yields.

Determination of yields from ambient atmospheric data differs from chamber determinations in
several additional respects. First, ambient measurements do not suffer from wall loss effects,
such that no corrections are necessary for loss of aerosol or semi-volatile gases (Matsunaga
and Ziemann, 2010;Krechmer et al., 2016).  Second, ambient measurements take place on the
aging time scale of the atmosphere rather than a time scale imposed by the characteristics of
the chamber or the choice of oxidant addition. Third, the typical lifetime of the initially produced
nitrooxy-isoprene-$RO_2$ radical is more representative of the ambient atmosphere rather than a
chamber. The unique conditions of a high $NO_x$ power plant plume affect lifetime and fates of
peroxy radicals, as described below.

To help interpret these in-plume peroxy radical lifetimes, a box model calculation using the
MCM v3.3.1 chemistry scheme was run (see details in Supplemental Information). This box
model shows substantially longer peroxy radical lifetimes during nighttime than daytime,
initializing with identical plume-observed conditions. These long peroxy radical lifetimes may
have consequences for comparison to chamber experiments: for example, in Schwantes'
(Schwantes et al., 2015) chamber experiment on the $NO_3$ + isoprene reaction mechanism, the
$HO_2$-limited nitrooxy-$RO_2$ lifetime was at maximum 30 s. In the plumes investigated in this study,
peroxy radical lifetimes are predicted to be substantially longer (>200 s early in the night, see
Fig. 7), allowing for the possibility of different bimolecular fates, or of unimolecular
transformations of the peroxy radicals that may result in lower-volatility products (e.g., auto-
oxidation to form highly oxidized molecules (Ehn et al., 2014)).


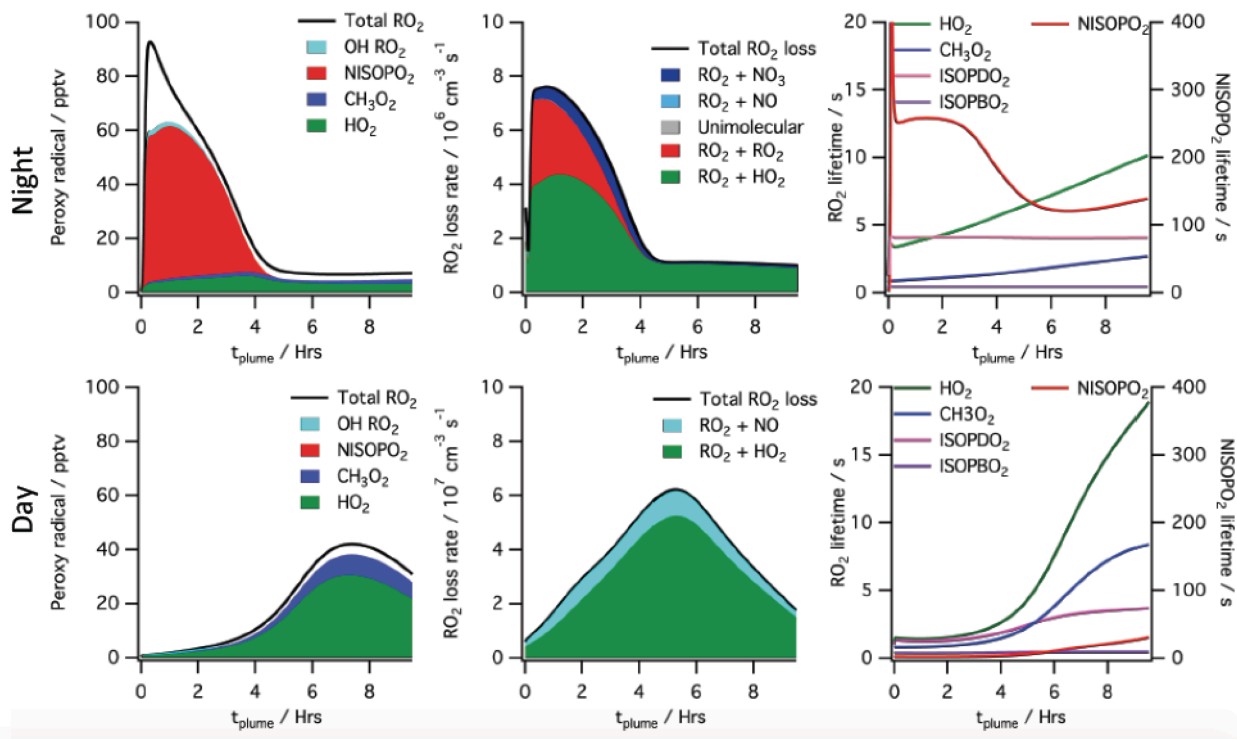

**Figure 7.** Simulated peroxy radical concentration (left), loss rates (middle), and lifetime (right), using the MCM v3.3.1 chemical mechanism, for conditions typical of a nighttime intercepted power plant plume (top) and the same plume initial conditions run for daytime simulation (bottom, local noon occurs at 5 hrs). Included are total peroxy radical concentration and losses, as well as the highlighted subclasses $HO_2$, $CH_3O_2$, total nitrooxy-isoprene-$RO_2$, and the total hydroxy-isoprene-$RO_2$ produced from OH oxidation. The righthand panels show $HO_2$, $CH_3O_2$ and the dominant hydroxy-isoprene-$RO_2$ ISOPBO$_2$ and ISOPDO$_2$ ($\beta$-hydroxy-peroxy radicals from OH attack at carbons 1 and 4 respectively) lifetime on the left axis and nitrooxy-isoprene-$RO_2$ on the right axis, showing nighttime lifetimes an order of magnitude longer than daytime for this $NO_3$ + isoprene derived $RO_2$ radical (NISOPO$_2$).

The typically assumed major fate of nighttime $RO_2$ in the atmosphere is reaction with $HO_2$ to yield a hydroperoxide, $NO_3$-ROOH. This is shown in the model output above as the green reaction, and is responsible for half of early $RO_2$ losses in the MCM modeled plume. Schwantes *et al.* (Schwantes et al., 2015) proposed reaction of these nighttime derived hydroperoxides with OH during the following day as a route to epoxides, which in turn can form SOA via reaction with acidic aerosol. Reaction of hydroperoxides with nighttime generated OH may similarly provide a route to SOA through epoxides, albeit more slowly than that due to photochemically generated OH.

The predicted longer nighttime peroxy radical lifetimes may enable unique chemistry. For example, if nitrooxy-isoprene-$RO_2$ self-reactions are substantially faster than assumed in the MCM, as suggested by Schwantes *et al.* (Schwantes et al., 2015), $RO_2$+$RO_2$ reactions may compete with the $HO_2$ reaction even more than shown in Fig. 7, and dimer formation may be favored at night, yielding lower volatility products. The 5:1 AMS Organic:Nitrate ratio observed in

840 the SOA formed in Rollins et al. (Rollins et al., 2009) , and consistent with aggregated
841 observations reported here, may suggest that in some isoprene units the nitrate is re-released
842 as $NO_2$ in such oligomerization reactions. We note that this larger organic to nitrate ratio would
843 mean higher SOA mass yields than estimated in Table 2.
844
845 Alternatively, longer nighttime peroxy radical lifetimes may allow sufficient time for
846 intramolecular reactions to produce condensable products. This unimolecular isomerization
847 (auto-oxidation) of initially formed peroxy radicals is a potentially efficient route to low-volatility,
848 highly functionalized products that could result in high aerosol yields. For OH-initiated oxidation
849 of isoprene, laboratory relative rate experiments found the fastest 1,6-H-shift isomerization
850 reaction to occur for the hydroxy-isoprene-$RO_2$ radical at a rate of 0.002 $s^{-1}$ (Crounse et al.,
851 2011), meaning that peroxy radicals must have an ambient lifetime of >500 s for this process to
852 be dominant. As shown in Fig. 7, the simulated power plant plume peroxy radical lifetimes are
853 long (>200 s), so an isomerization reaction at this rate may play a significant role. However, a
854 recent study has demonstrated that OH-initiated and $NO_3$-initiated $RO_2$ radicals from the same
855 precursor VOC can have very different unimolecular reactive fates due to highly structurally
856 sensitive varying rates of reactions of different product channels (Kurtén et al., 2017). A similar
857 theoretical study on the rate of unimolecular autooxidation reactions of nitrooxy-isoprene-$RO_2$
858 radicals would be valuable to help determine under what conditions such reactions might occur,
859 and this knowledge could be applied to comparing chamber and field SOA yields.

860 **4.4 Atmospheric implications and needs for future work**
861 Because this paper proposes higher SOA yield for the $NO_3$ + isoprene reaction than measured
862 in chamber studies, we conclude with some discussion of the implications for regional aerosol
863 burdens, and further needs for investigation in the $NO_3$ + isoprene system.
864
865 Using an isoprene + $NO_3$ yield parameterization that gave a 12% SOA mass yield at 10 µg $m^{-3}$,
866 Pye et al. (2010) found that adding the $NO_3$ + isoprene oxidation pathway increased isoprene
867 SOA mass concentrations in the southeastern United States by about 30%, increases of 0.4 to
868 0.6 µg $m^{-3}$. The larger $NO_3$ + isoprene SOA mass yields suggested in this paper, with average
869 value of 30%, could double this expected $NO_3$ radical enhancement of SOA production.
870 Edwards et al. (2017) concluded that the southeast U.S. is currently in transition between $NO_x$-
871 independent and $NO_x$-controlled nighttime BVOC oxidation regime. If $NO_3$-isoprene oxidation is
872 a larger aerosol source than currently understood, and if future $NO_x$ reductions lead to a
873 stronger sensitivity in nighttime BVOC oxidation rates, regional SOA loadings could decrease by
874 a substantial fraction from the typical regional summertime OA loadings of 5 +/- 3 µg $m^{-3}$ (Saha
875 et al., 2017).
876
877 Analysis of the degree of oxidation and chemical composition of $NO_3$ + isoprene SOA would
878 help to elucidate mechanistic reasons for the different field and lab SOA yields. For example,
879 the potential contribution of the uptake of morning-after OH + NISOPOOH produced epoxides,
880 discussed above in section 4.3, onto existing (acidic) aerosol could be quantified by
881 measurement of these intermediates or their products in the aerosol phase. Assessment of
882 degree of oxidation could help determine whether auto-oxidation mechanisms are active. Future

similar field studies would benefit from the co-deployment of the complementary tool of a
Chemical Ionization Mass Spectrometer (CIMS) to detect $NO_3$ + isoprene products such as
organic nitrates (Slade et al., 2017;Lee et al., 2016). Because of the potentially large effect on
predicted SOA loading in regions of high isoprene emissions, a better mechanistic
understanding of these observed yields is crucial.

**Acknowledgements**
JLF gratefully acknowledges funding from the EPA STAR Program (no. RD-83539901) and from
the Fulbright U.S. Scholars Program in the Netherlands. PCJ, DAD, and JLJ were partially
supported by EPA STAR 83587701-0 and DOE (BER/ASR) DE-SC0016559. This paper has not
been formally reviewed by EPA. The views expressed in this document are solely those of the
authors, and do not necessarily reflect those of EPA.  EPA does not endorse any products or
commercial services mentioned in this publication.

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

## Supplemental Information

In the main text, we noted a discrepancy between overall average aerosol volume estimates based on size measurements vs. AMS for the flight analyzed here (see Figure S1). We checked to see if this bias was also present in the individual plumes studied here by calculating the volume changes from the sizing instruments and the derived volume changes from the AMS+rBC mass. There is quite a bit of scatter in the volume enhancements, with with some of the points falling along the same line as the data for this flight and some falling significantly below the line. It is unclear why the two types of volume measurements disagree more for this flight. Therefore, the bias in volume changes introduces additional uncertainty in the magnitude of the plume enhancements, which is not included in the uncertainty propopagation.

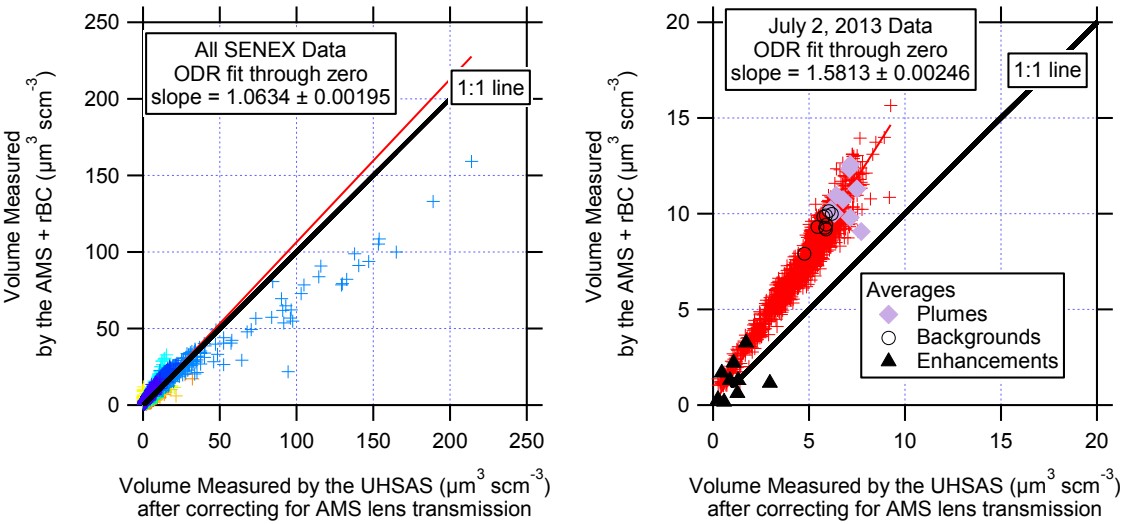

**Figure S1.** Aerosol volume measured using the total aerosol mass from the AMS plus refractory black carbon (rBC) and mass-weighted densities versus the aerosol volume measured by optical size with the UHSAS after correcting for AMS lens transmission. The procedure for calculating the mass-weighted density is described by Bahreini et al. (2009). On average, the measured aerosol volume from composition is roughly equal to the measured aerosol volume from size for the entire SENEX study (left hand panel) and is higher than one for the flight analyzed here (July 2, 2013, right hand panel).

**Corrections for AMS UMR nitrate data and applicability to pRONO$_2$ estimation**

Nitrate in the AMS is quantified in unit mass resolution mode (UMR) as the sum of the estimated NO$^+$ at $m/z$ 30 and NO$_2^+$ at $m/z$ 46, with a correction factor to account for the smaller ions (N$^+$ and HNO$_3^+$, mostly) produced from nitrate (Allan et al., 2004b). The default AMS UMR quantification algorithm (documented in the AMS "fragmentation table") estimates NO$^+$ as the total signal at $m/z$ 30 minus a small (2.2% of OA at $m/z$ 29, "Org29" in AMS parlance) subtraction to account for organic interferences and an isotopic correction for naturally-occurring $^{15}$N$_2$ from nitrogen in air. The default UMR fragmentation table was developed for mixed ambient aerosols, in particular in urban studies, and it is the responsibility of each AMS user to correct it as needed for each study. In environments with high biogenic contributions to total OA, and/or

low total nitrate concentrations, the contribution of the $CH_2O^+$ ion can be much larger than the
default subtraction at $m/z$ 30. Similarly, the $CH_2O_2^+$ ion at $m/z$ 46 becomes non-negligible, and
hence nitrate reported from AMS data with UMR resolution will frequently be overestimated in
these situations. The poor performance of the default AMS correction is likely due to the initial
focus on urban OA with high nitrate fractions when deriving those corrections (Allan et al.,
2004b;Zhang et al., 2004).
Here we derive a set of corrections based on an aircraft high-resolution (HR) dataset acquired
with the University of Colorado HR-AMS (Dunlea et al., 2009) on the NASA DC-8 during the
SEAC[4]RS campaign (Toon et al., 2016). SEAC[4]RS took place with a strong emphasis on the
SEUS 6 weeks after the SENEX flight analyzed in this manuscript. Based on an initial screening
of the correlations of the $CH_2O^+$ and $CH_2O_2^+$ ions with UMR signals, 10 potential UMR $m/z$
between $m/z$ 29 and $m/z$ 53 were selected as viable for deriving suitable corrections. Further
analysis using three specific SEAC[4]RS flights (RF11 on 30 Aug 30[th], 2013, RF16 on Sep 11[th],
2013 and RF18 on Sep 16[th], 2013) that covered a wide range of OA composition with both
strong biogenic contributions and fresh and aged biomass plumes showed that only four $m/z$
(29, 42, 43 and 45) had good enough S/N and robust enough correlations to be used as
corrections. Table S1 summarizes the correction coefficients obtained in this analysis, and
Figure S2 shows the ability of matching the actual $NO^+$ and $NO_2^+$ signals (as obtained from
high-resolution analysis of these flights) with the corrected UMR procedure. These corrections
are applied as:
1394                 UMR NO =  Signal($m/z$30) – $a_i$*Signal(Variable$_i$)
1395                 UMR NO$_2$ = Signal($m/z$ 46) – $b_i$*Signal(Variable$_i$)
with the coefficients $a_i$ and $b_i$ as reported in Table S1. It should be noted that in all cases the
contributions of $C^{18}O^+$ to $m/z$ 30 need to be subtracted first before applying the correction (which
is constrained to the organic $CO_2^+$ signal, measured at $m/z$ 44, by the naturally-occurring
isotopic ratio and assuming that OA produces $CO^+ = CO_2^+$ (Zhang et al., 2005;Takegawa et al.,
2007). Likewise, the contribution of $^{13}CO^+$ to Org29 needs to be subtracted first. It is hence very
important for this analysis that the corrections to the AMS frag table to suitably estimate the
contribution of gas phase $CO_2^+$ to total $UMR$ $m/z$ 44 as well as the baseline correction for $m/z$ 29
be properly applied first (Allan et al., 2004b). Finally, also note that the corrections using $m/z$ 29
and 43 are rather based on Org29 and Org43, which are standard AMS products that take the
OA relative ionization efficiency (RIE) into account.
For the SEAC[4]RS dataset, the corrections amounted to on average subtracting 55% from UMR
$m/z$ 30 and 33% from UMR $m/z$ 46. Despite this large subtraction, the corrected data correlates
very well with the HR AMS results, with less than 5% deviation in the regression slope between
the two datasets.
Although all of the corrections in Table S1 were valid for the SEAC[4]RS data set, for the flight
analyzed here we chose Org29 to correct $m/z$ 30 and $mz$ 45 correction to correct $m/z$ 46
because they were the closest organic signals to the UMR nitrate peaks with organic
interferences and may be more valid for other field studies where different types of OA are
sampled. After these UMR signals were corrected and the appropriate RIEs and CE were
applied, the nitrate mass concentrations in the final data archive for the flight analyzed here
were reduced by 0-0.24 µg sm$^{-3}$, averaging 0.11 µg sm$^{-3}$ or 32%. The corresponding increase in
OA due to the organic interferences in the UMR nitrate had linear dependence on the reported
OA mass concentrations ($r^2$ = 0.89) with a slope of 1.3%.
To estimate the fraction of nitrate that is organic nitrate (pRONO$_2$) the use of the NO$_2^+$/NO$^+$ ratio
with an empirically determined pRONO$_2$ calibration ratio has been successfully used previously
with HR-AMS data (Farmer et al., 2010;Fry et al., 2013;Ayres et al., 2015;Fisher et al.,
2016;Day et al., 2017;Palm et al., 2017;Lee et al., 2016). Figure S2 summarizes how well the
ratio of the corrected UMR *m/z* 30 and 46 signals correlate with the NO$_2^+$ and NO$^+$ (and ratios)
determined using HR data. As expected, there is considerable scatter at very low nitrate
concentrations (which is a considerable part of the dataset, as the time series shows, since the
free troposphere was sampled extensively). However, for the predicted pRONO$_2$ (which is
mass-weighted), most of this scatter disappears, and for concentrations above 0.1 µg sm$^{-3}$ of
nitrate there is good agreement between the HR results and the UMR-corrected pRONO$_2$,
regardless of the correction chosen. For lower concentrations the scatter is considerable larger,
with the Org29 correction providing the best overall agreement. Based on the variability in this
dataset for this correction (Org29), we estimate the uncertainty in pRONO$_2$ fraction
apportionment using UMR to be about 30%, in addition to an estimated uncertainty for the
apportionment method using HR of 20%. From the comparison of UMR-corrected total nitrate
to HR nitrate (not shown), we estimate an additional error of 5% for total nitrate error using
these corrections.
As mentioned in the main text, the empirically determined pRONO$_2$ calibration ratio used for the
flight data analyzed here was the ratio of NO$_2^+$/NO$^+$ from the ammonium nitrate calibration
aerosols divided by 2.8. This factor was determined as the average of several literature studies
(Fry et al., 2009;Farmer et al., 2010;Rollins et al., 2009;Sato et al., 2010;Fry et al., 2011;Boyd et
al., 2015) and applied according to the "ratio of ratios" method (Fry et al., 2013). The ammonium
nitrate NO$_2^+$/NO$^+$ ratio was obtained from the two calibrations on 30 June and 7 July that
bracketed the flight on 2 July, as described above. This ratio averaged 0.490. Hence, the
organic nitrate NO$_2^+$/NO$^+$ ratio was estimated to be 0.175. The ratio of NO$_2^+$/NO$^+$ from the flight
data was then used with the pRONO$_2$ and ammonium nitrate NO$_2^+$/NO$^+$ calibration ratios to
estimate the fraction of the total corrected nitrate mass concentrations that was organic
(pRONO$_2$) or inorganic (nitrate associated with ammonium or NH$_4$NO$_3$). Propagating the 30%
UMR vs HR uncertainty and 20% apportionment (see above) error on top of the 34% AMS total
nitrate measurement uncertainty results in ±50% uncertainties in the derived organic nitrate
mass concentrations (and similar for NH$_4$NO$_3$; however it will depend on the relative
contributions of pRONO$_2$ and NH$_4$NO$_3$ to total nitrate since the absolute concentration errors
associated with pRONO$_2$ - NH$_4$NO$_3$ apportionment should be similar [64]).

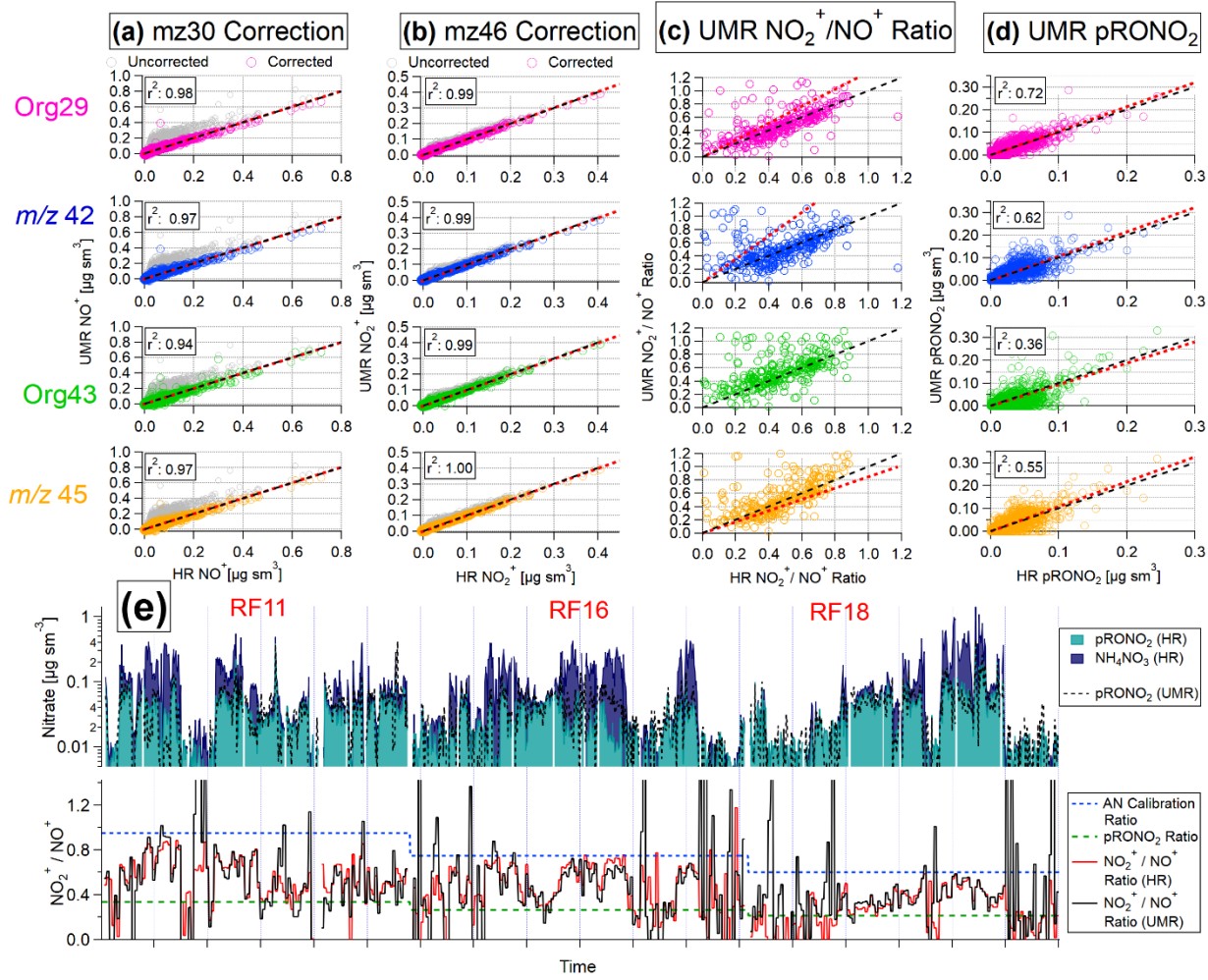


**Figure S2.** (a and b) Comparison of $m/z$ 30 and 46 with the $NO^+$ and $NO_2^+$ signals from the high
resolution analysis of the AMS data before and after applying the four different corrections listed
in Table S1. The Pearson $r^2$ for the corrected dataset is shown as well. (c) Comparison of the
$NO_2^+/NO^+$ ratio obtained from HR analysis with the ratios of the corrected UMR NO and $NO_2$
variables (d) Comparison of the $pRONO_2$ concentrations derived using the HR and UMR $NO_2^+/$
$NO^+$ ratios. (e) Time series of the total and speciated nitrate as reported from HR analysis of the
SEAC[4]RS data (DOI: 10.5067/Aircraft/SEAC4RS/Aerosol-TraceGas-Cloud) compared to the
speciation using the Org29 correction (note the logarithmic scale). The bottom time series
shows the $NO_2^+/NO^+$ ratio that the speciation is based on, again for the HR and corrected UMR
case.
**Table S1.** Coefficients used to correct $m/z$ 30 and 46 to estimate total nitrate.

| AMS Variable | Correction coefficient for $m/z$ 30 ($a_i$) | Correction coefficient for $m/z$ 46 ($b_i$) |
|---|---|---|
| Org29 | 0.215 | 0.037 |
| $m/z$ 42 | 0.51 | 0.092 |
| Org43 | 0.215 | 0.037 |
| $m/z$ 45 | 0.72 | 0.127 |


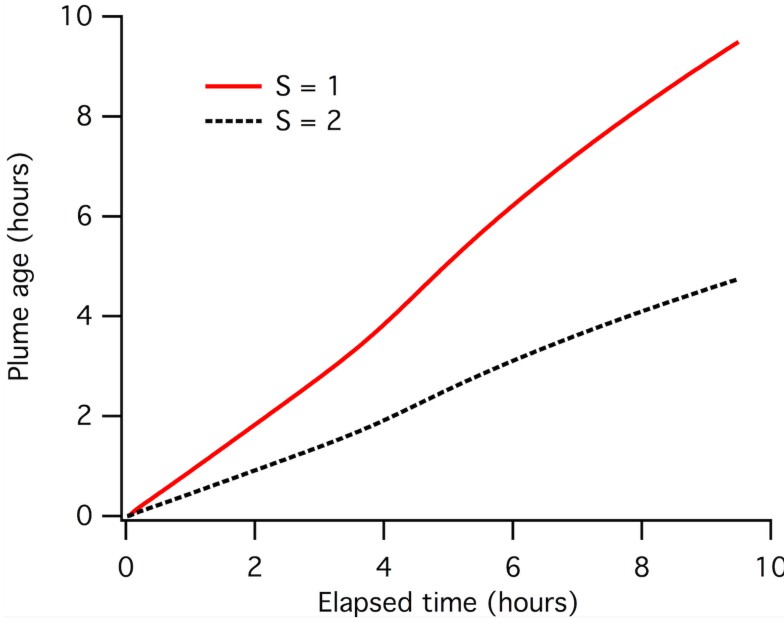

**Figure S3.** Calculated plume age vs. elapsed time in a box model run for a single
representative night. Plume ages on the y-axis are calculated based on Equation 1 in the main
text but using model $NO_2$ and $O_3$ data. Time since sunset on the x-axis is the model elapsed
time (i.e., run time of the model during darkness).

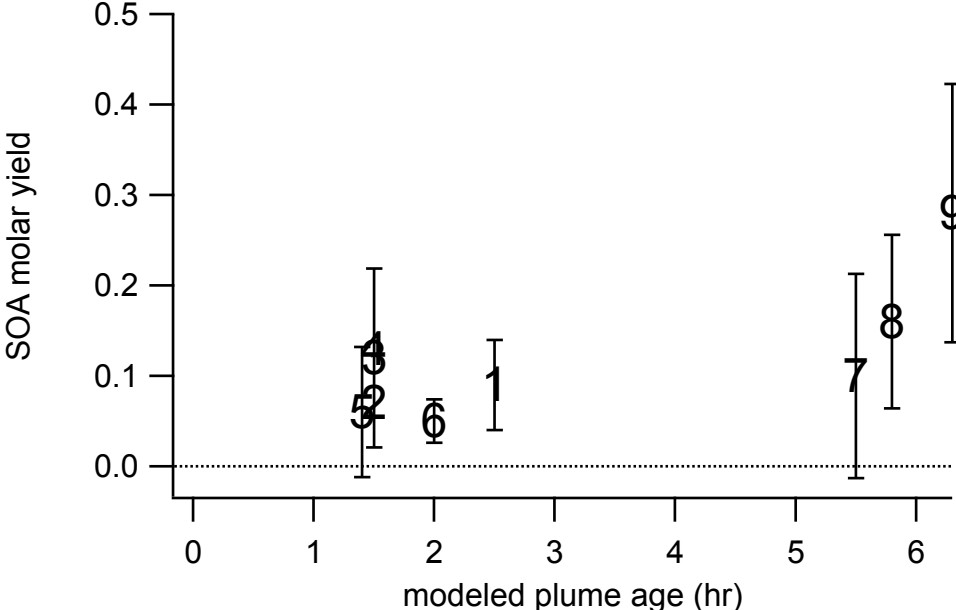

**Figure S4.** SOA molar yield is positively correlated with estimated plume age. This SOA molar
yield is based on Eq. 3, with error bars determined by propagation of observed variability in
$pRONO_2$ and isoprene, where multiple point averaging was possible. Markers correspond to
plume numbers. ). Based on the box model described in more detail below, the first-generation
isoprene products peak at a approximately 4 hours plume age and then begin to decay.
**Table S2**. Peak ambient (wet) aerosol surface area during each plume used in the yield
analysis (plume numbers 1 – 9), and for the two longer urban plumes transected at the end of
the flight.

| plume number | 7/2/13 plume time (UTC) | Peak aerosol surface area ($\mu m^2\ cm^{-3}$) |
|---|---|---|
| 1 | 2:18 | 280 |
| 2 | 2:20 | 370 |
| 3 | 2:21 | 470 |
| 4 | 3:03 | 340 |
| 5 | 3:55 | 800 |
| 6 | 4:34 | 470 |
| 7 | 4:37 | 370 |
| 8 | 4:39 | 420 |
| 9 | 5:04 | 490 |
| Urban plume | 5:36 | 340 |
| Urban plume | 6:37 | 300 |


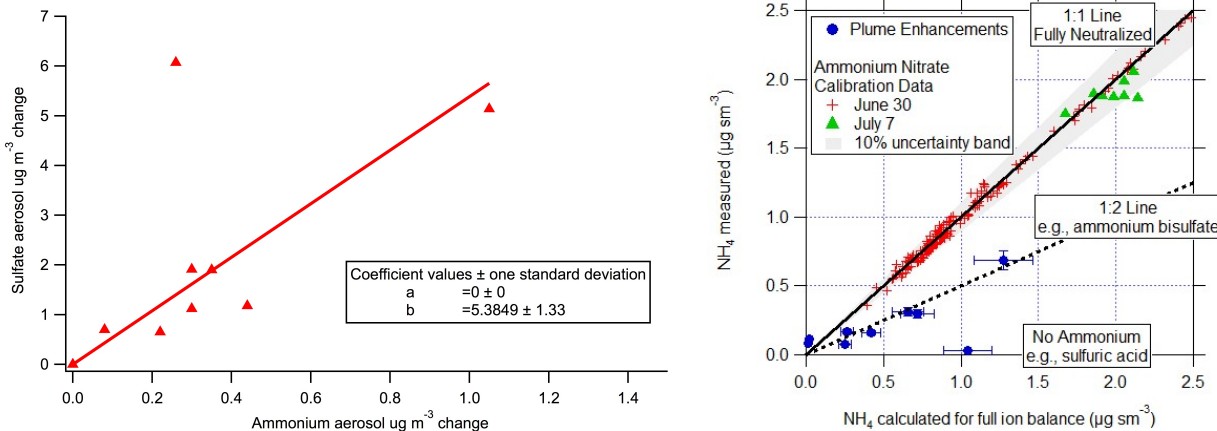


**Figure S5.** (a) In-plume change in sulfate mass concentration vs. change in ammonium aerosol
mass concentration is generally well correlated, with a slope of 5.4. The masses of the cations
and anions would give an ion balance for pure $(NH_4)_2SO_4$ of $MW(SO_4)/(2 \times MW(NH_4)) = 2.7$, and
for $(NH_4)HSO_4$ of $MW(SO_4)/(MW(NH_4)) = 5.4$. Hence, this slope provides support for a mix of
these two ammonium sulfate salts, with sometimes exclusively $(NH_4)HSO_4$. This is consistent
with incomplete neutralization of the sulfate mass by ammonium. The one clear outlier (sulfate
increase of 6 µg m$^{-3}$ for Plume #5) suggests excess sulfate, rendering ammonium or other
inorganic nitrate formation even less likely. Points with ammonium aerosol below 0.1 µg m$^{-3}$ are
within the variability of that measurement; their omission does not change the slope. (b)
Measured vs. calculated (ion balanced) $NH_4$ for calibration data and plume enhancements. This
also shows that plumes are acidic than ammonium sulfate, ruling out the possibility of inorganic
nitrate formation.

 **Additional AMS and auxiliary data from plumes**


**Table S3.** Additional information for the list of plumes used in this $NO_3$ + isoprene SOA yield
analysis, for which key yield-related data is presented in Table 1. For each plume, the delta-
values listed indicate the difference between in-plume and outside-plume background in
average observed concentration. After each plume number, the numbers of points averaged for
isoprene and AMS, respectively, are listed. Plume numbers annotated with * indicate brief
plumes for which only single-point measurements of in-plume aerosol composition were
possible. Also shown are the plume changes in isoprene used in the present analysis (Δisop,
the difference between in-plume and background isoprene concentration, reproduced from
Table 1), alongside for comparison the Δisop determined as the difference between in-plume
isoprene and the modeled sunset (initial) concentration of isoprene present at that location
outside of the plume, determined using an iterative box model (Edwards et al., 2017). The
similarity between these two values for most points suggests that the isoprene just outside of
each plume transect was largely unperturbed from the sunset initial value.

| plume number [#isop/#AMS] | 7/2/13 plume time (UTC) | $\Delta ORG_{aero}$ (µg m$^{-3}$) | $\Delta NH_{4,aero}$ (µg m$^{-3}$) | $\Delta SO_{4,aero}$ (µg m$^{-3}$) | Temp (C) | %RH | Δisop (pptv) | Δisop from model (pptv) | Isop:MT Mole Rat |
|---|---|---|---|---|---|---|---|---|---|
| Typical variability (µg m$^{-3}$): | | 0.75 | 0.1 | 0.5 | | | | | |
| 1 [2/3] | 2:18 | 0.35 | 0 | 0 | 23.6 | 66.5 | -335 | -327 | 36.5 |
| 2 [*] | 2:20 | 0.89 | 0.3 | 1.91 | 23.6 | 65 | -404 | -453 | 71.4 |
| 3 [4/5] | 2:21 | 1.25 | 1.05 | 5.14 | 23.6 | 65.2 | -228 | -337 | 16.6 |
| 4 [*] | 3:03 | 0.16 | 0.08 | 0.7 | 21.2 | 68.1 | -453 | -391 | 50.6 |
| 5 [3/4] | 3:55 | 0.32 | 0.26 | 6.07 | 21.9 | 65.5 | -255 | -376 | 34.2 |
| 6 [2/2] | 4:34 | 0.57 | 0.3 | 1.12 | 19.9 | 74.6 | -713 | -233 | 17.3 |
| 7 [5/6] | 4:37 | 1.05 | 0.22 | 0.65 | 19.7 | 76.2 | -298 | -221 | 14.2 |
| 8 [2/3] | 4:39 | 1.26 | 0.44 | 1.18 | 18.3 | 82.2 | -443 | -353 | 11.0 |
| 9 [7/8] | 5:04 | 1.45 | 0.35 | 1.9 | 17.2 | 84.8 | -293 | -434 | 17.8 |

**Box model calculations**
Box model simulations were performed using the Dynamically Simple Model of Atmospheric
Chemical Complexity (DSMACC, http://wiki.seas.harvard.edu/geos-
chem/index.php/DSMACC_chemical_box_model), containing the Master Chemical Mechanism
v3.3.1 chemistry scheme (http://mcm.leeds.ac.uk/MCM/). The model approach is similar to that
described in detail in Edwards et al. 2017, and the accompanying supplement, with the model
run over a 9.5 hour night to simulate the nocturnal residual layer. For the nocturnal simulation
used in this work (for both the plume lifetime calculation and the peroxy radical lifetime analysis
in Sect. 4.3) the model was initialized with concentrations of the constraining species
representative of the SENEX observations (Table S4). As the model is simulating power plant
plume evolution from point of emission, a starting NO mixing ratio of 10 ppb was used to
constrain $NO_x$, and the chemistry scheme was subsequently allowed to partition the reactive
nitrogen. The top panels in Figure S7 show the evolution of key species during this nocturnal
simulation.
**Table S4**: Species constrained (MCM v3.3.1 names) during model simulations and constraining
values. Constraint column indicates if species concentrations were held at the constrained value
throughout the simulation (Fixed) or allowed to vary after initialization (Initial).

| Species | Mixing ratio | Units | Constraint |
|---------|--------------|-------|------------|
| NO | 9.28 | ppb | Initial |
| O3 | 55.72 | ppb | Initial |
| CO | 134.00 | ppb | Fixed |
| CH4 | 1920.00 | ppb | Fixed |
| C5H8 | 2606.80 | ppt | Initial |
| APINENE | 38.87 | ppt | Initial |
| BPINENE | 195.50 | ppt | Initial |
| LIMONENE | 12.42 | ppt | Initial |
| MACR | 454.13 | ppt | Initial |
| MVK | 1006.00 | ppt | Initial |
| IC4H10 | 47.00 | ppt | Fixed |
| NC4H10 | 128.00 | ppt | Fixed |
| C2H6 | 1199.00 | ppt | Fixed |
| C2H4 | 117.00 | ppt | Fixed |
| C2H2 | 145.00 | ppt | Fixed |
| NC6H14 | 20.00 | ppt | Fixed |
| IC5H12 | 120.00 | ppt | Fixed |
| NC5H12 | 76.00 | ppt | Fixed |
| C3H8 | 344.00 | ppt | Fixed |
| C3H6 | 26.00 | ppt | Fixed |
| CH3COCH3 | 2556.00 | ppt | Fixed |
| BENZENE | 35.90 | ppt | Fixed |
| C2H5OH | 2239.00 | ppt | Fixed |
| MEK | 309.00 | ppt | Fixed |
| CH3OH | 5560.00 | ppt | Fixed |

The daytime simulation used for comparison in Sect. 4.3 of the main manuscript (lower panels
of Figure S7) uses the same initialization as the nocturnal simulation; with the only difference
being the model is run during the daytime. Photolysis rates are calculated using TUV
(https://www2.acom.ucar.edu/modeling/tropospheric-ultraviolet-and-visible-tuv-radiation-model).
The daytime simulation does not accurately simulate daytime mixing ratios of species such as
$O_3$ representative of SENEX observations. However, the intent of this simulation is to compare
model daytime peroxy radical fate and lifetime with the nocturnal simulation. The presence of

 intense convective mixing in the daytime planetary boundary layer of the Southeast US makes
accurately modeling these concentrations difficult with a zero dimensional model.

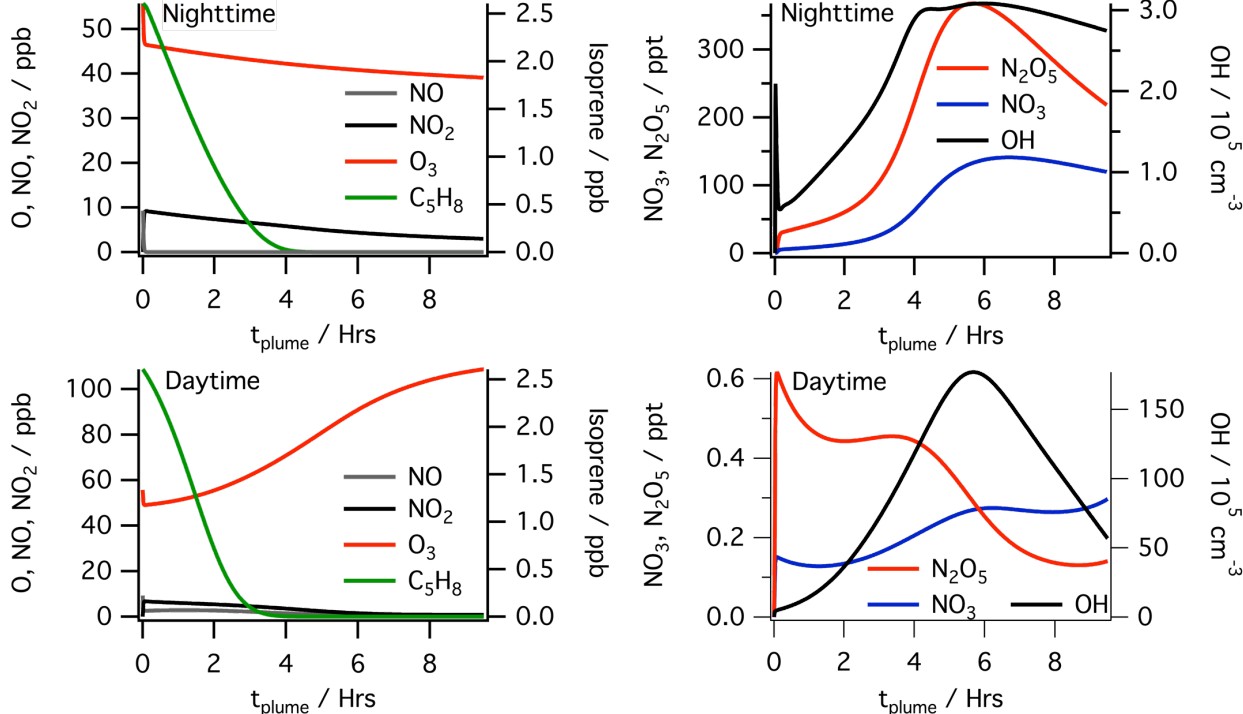

**Fig. S6**. Model calculated NO, NO$_2$, O$_3$, and isoprene (left) and NO$_3$, N$_2$O$_5$ and OH (right for the nocturnal (top) and daytime (bottom) simulations shown in Sect. 4.3.


### Additional considerations investigated via RO$_2$ fate box modeling


Based on the potentially larger than previously estimated contribution of RO$_2$+RO$_2$ reactions at night, we considered a related possible source of a high bias in the determined SOA yields. If NO$_3$ reaction with the major daytime isoprene oxidation products MVK and/or MACR produces RO$_2$ radicals that can cross-react with NO$_3$ + isoprene products to produce condensable products, this would be a mechanism of recruiting isoprene-derived organic mass into the aerosol, but that original isoprene oxidation would not be counted in the denominator of the yield calculation, since its interaction with NO$_3$ began as MACR or MVK. In the box model, substantial MVK and MACR are available in the plume at nighttime, but only MACR reacts with NO$_3$, and a maximum fraction of one-quarter of MVK+MACR losses go to reaction with NO$_3$ overnight (see Figure S8). In addition, in our power plant plume observations, MVK+MACR are not observed to be appreciably depleted by the large NO$_3$ injection, further suggesting that this chemistry is not a substantial additional source of SOA (see Figure S9).

1571

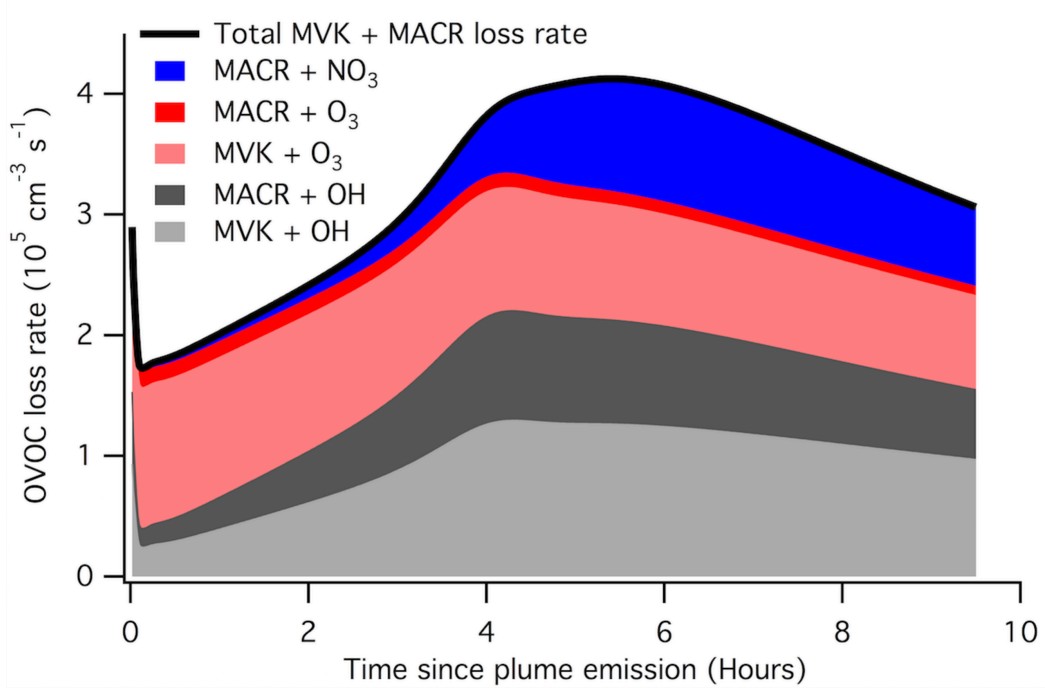

**Figure S7.** Calculated (via MCM) loss rate contributions for the daytime isoprene products methyl vinyl ketone (MVK) and methacrolein (MACR) in the simulated nighttime plume used in the text. Only MACR reacts with $NO_3$, and the contribution of this process to total losses (green stack) is relatively minor.

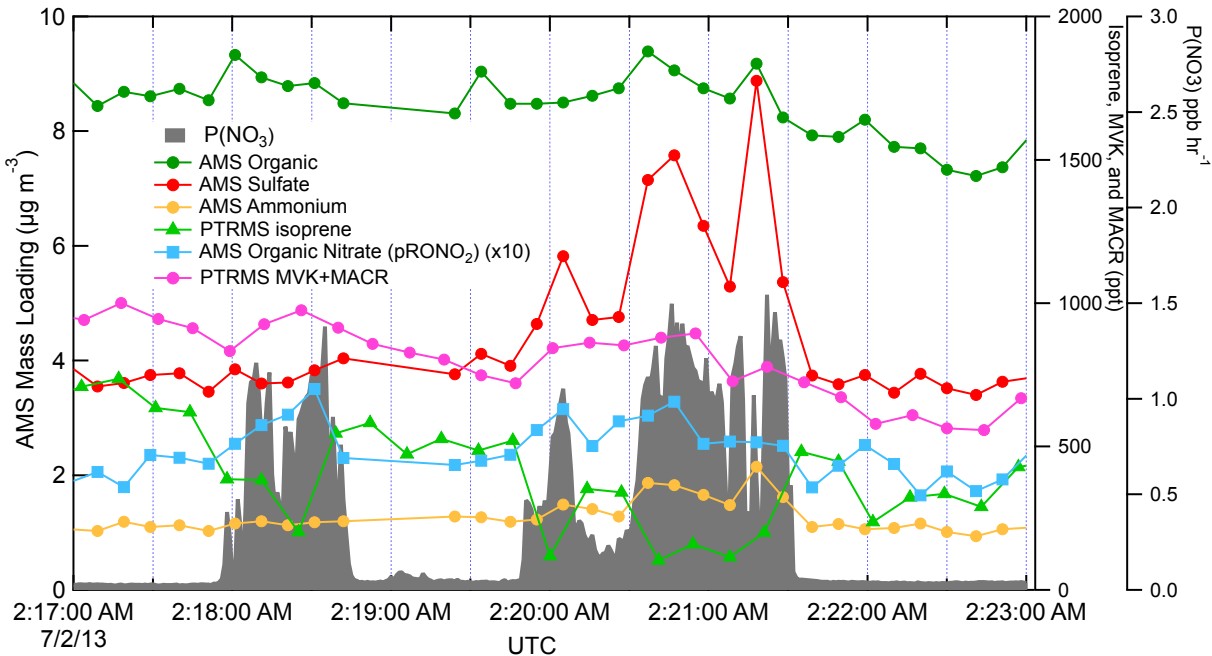

**Figure S8**. MVK and MACR are not titrated on the timescale of these yield estimates in power plant plumes.

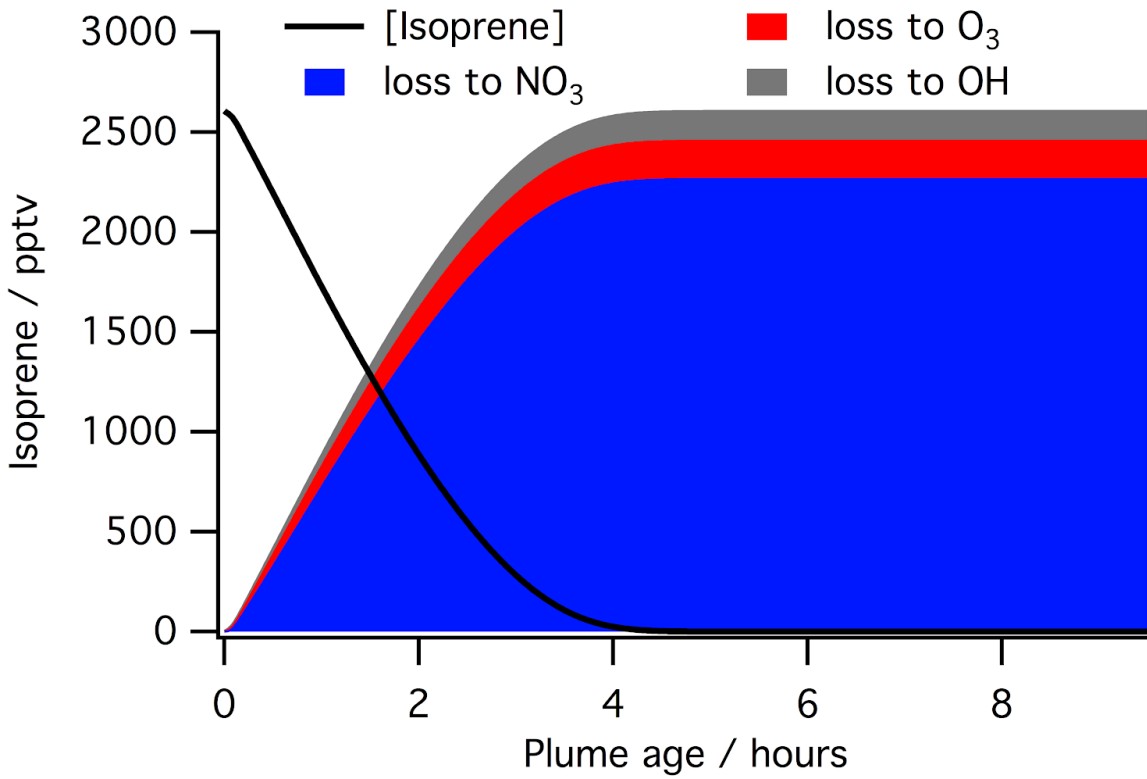

1581
1582

**Figure S9.** Model simulation of typical in-plume consumption of isoprene (black line), and stacked plot showing the contributions to this from the $NO_3$, $O_3$, and OH. Modeled plume was emitted at sunset, so this represents nocturnal processing under power plant plume conditions.

## Two urban plume case studies

In addition to the nine power plant plumes analyzed above to determine the $NO_3$ + isoprene SOA molar yield, towards the end of the July 2 flight, the Birmingham urban plume was intercepted twice (around 5:36 am and 6:37 am UTC, Fig. 8). These downwind urban plumes are among the most aged plumes (estimated at 5.2 and 5.8 hours, respectively), but are also substantially more diffuse than the narrow power plant plume intercepts and have lower peak $P(NO_3)$. Nevertheless, we note that these two plumes contain periods of apparent anti-correlation of isoprene and organic nitrate aerosol time series and high apparent SOA molar yields (23%, 19%) and mass yields (62%, 51%), if calculated by the same method as above and omitting the period of vertical profiling in the second plume. Potentially complicating these urban SOA yield determinations is the fact that the inorganic fraction of nitrate was much larger than in the power plant plumes (see Fig. 8). The background isoprene is also somewhat lower in these urban plumes, potentially shifting the $NO_3/N_2O_5$ fate to reactions other than $NO_3$ + isoprene (see Fig. S4 in Edwards et al. (Edwards et al., 2017)). The aerosol surface area is not noticeably higher in these urban plumes, which one might expect to lead to a larger contribution of $N_2O_5$ uptake and hydrolysis. In the more complex mix of gases characteristic of an urban plume, we hesitate to attribute these apparent yields exclusively to the $NO_3$ + isoprene reaction.

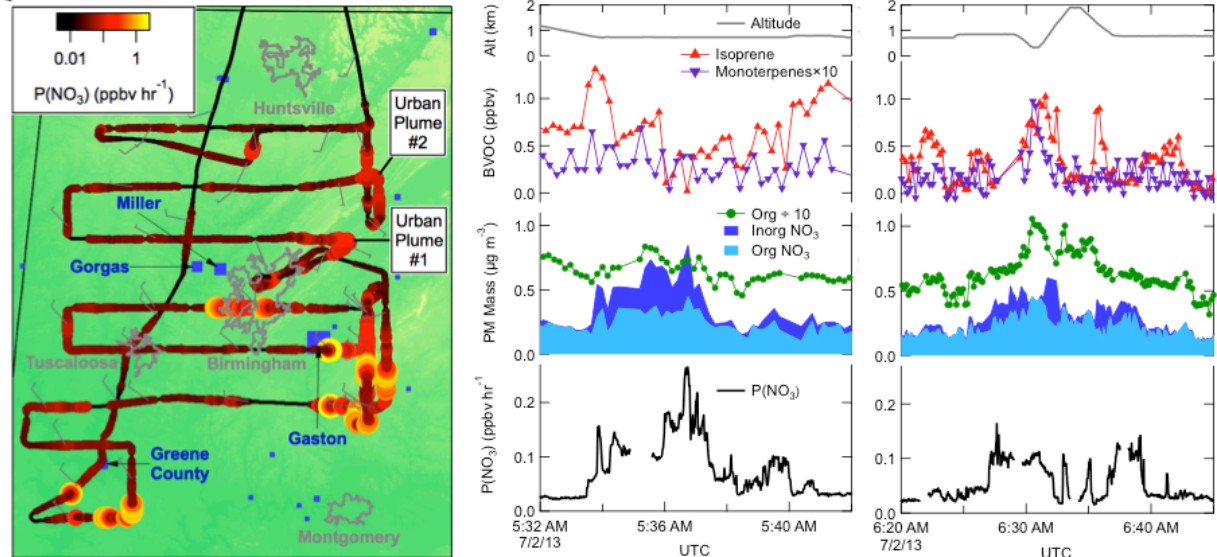

 **Figure S10.** Flight map and time series of two urban plume intercepts, showing anticorrelation
 of organic nitrate and isoprene. These more diffuse plumes, with lower $P(NO_3)$ and larger
 inorganic nitrate contribution, make yield determination more uncertain, so we do not include
 them in the overall yield determination. However, using the same methodology as for the power
 plant plumes would give similarly high yields for these very aged plumes.