# Peer review of "Secondary Organic Aerosol (SOA) yields from NO3 radical"

_Atmospheric Chemistry and Physics, 2018_

## Referee Comment (RC1) · Anonymous Referee #1 · 20 Apr 2018

The manuscript is original and very interesting to read. The authors tried to get the optimum out of the data, but still addressed openly the limitations of their approach.

From the viewpoint of raising interesting questions regarding the role of isoprene chemistry and isoprene NO3 chemistry for SOA formation and interesting approaches to address these questions, the paper could be published after some minor revisions (most of it of formal character, e.g. references in text and supplement, see below).

However the manuscript fails clearly short behind its title claim and from this point of view, I suggest to reject the manuscript, due to the major concerns following below. Since the authors have already done the best with their data in a positive sense, I

[Figure]

guess major revisions would not make sense.

A way out could be a reformulation of the title of the manuscript away from "providing yields" (reliable numbers) to a more procedural character of "addressing an important issue with interesting approaches and possibly important findings".

Major:

The authors convinced me that pRONO2 and thus organic nitrate in plumes is enhanced and that may indeed relate to enhanced NO3 concentrations (Figure 6). However, the paper does not really show that that increase of pRONO2 is related to isoprene oxidation alone (Figure 5). While the reasoning of a single -ONO2 group per organic nitrate molecules is an acceptable approach to derive molar yields, the scatter in Figure 5 casts doubts, if the increase ofpRONO2 is really related solely to isoprene oxidation. Herein the weak point is the limited number of data points. I don't say the authors are wrong, but one would need more observations to strengthen the case. I concede that the authors revealed an interesting phenomenon, interesting enough to pursue the ideas and go out and get more/better proof.

L: 651: In going from the molar yield to the mass yield the uncertainty - and speculations clearly indicated as such, though - become even larger. On one hand obviously two oxidation steps are needed to achieve condensable isoprene oxidation products, on the other hand NO3 seems to be the only available oxidant. Oxidation of both double bonds should thus lead to dinitrates. If pOrgNO3 would really isoprene dinitrates the estimated yield would drop from 27% to 18%, not so far away from the referenced value by Rollins of 14%. I can follow the authors that it is likely thatpRONO2 dinitrates could be hydrolysed, but why should hydrolysis stop after one group, why not hydrolysing every second -ONO2 group ore even both ONO2 groups? Moreover, as far as I understand, Rollin's value is based in parts on observations of several hours in a large chamber. So reaction time cannot be an issue?! I agree that wall losses could be an issue, though, but wall losses are also less important in large chambers. Therefor with the same right I could argue that Rollin's yield of 14% is correct and then ask where could the rest of organic nitrate come from. I follow the authors that inorganic nitrate can be excluded as source. Could it be that the organic nitrates arise from liquid phase or heterogeneous processes via NO2, NO3? Is anything known about such heterogeneous nitration processes? NOX and NO3 were by definition high in the plumes.

Actually if I really think about it the mass yield analysis adds not much beyond the molar yield considerations and an analogous plot would just reproduce Figure5 with slopes of 18% or 27%, depending only on the assumption if the isoprene nitrates bring in two or three times the molecular weight of isoprene itself.

L483/ Fig. 2: What also concerns me and this is again related to small number of cases: There are indeed correlations between PNO3 andpRONO2 and anti-correlation with isoprene, but pOrg NO3 sometimes increases by the same amount in the absence of plumes (2:17.30AM, 2:22.00AM) and some plumes do not create OrgNO3 despite lower isoprene (ca. 2:21.20AM).

Minor:

l315, Fig.S1: If I compare the SENEX data with actual data in the range of plumes and background the difference is more a factor of 2 than 1.6. Moreover, two of the plumes fall off line while all the background measurements correlate as all other data. Unfortunatly, the exception of AMS performance(?) or UHSAS performance(?) for "just that flight" in addition weakens the case.

I suggest listing also PNO3 in Table 1; that would help to link quickly oxidation strength and observed effect

Main text: references not in ACP format Replace "author et al. (author et al., year)" by "author et al. (year)"

Supplement: the literature is not assessable and given in bracketed format

l59: review

l172: Xu et al. (2015)

l339: Fry et al. [47]

l657: no "N" in the formula, it is not clear that refer only to organic rest of the tri-hydroxynitrate

l418: I suggest to replace "number densities" with "concentration" in context of gases

---

## Referee Comment (RC2) · Anonymous Referee #2 · 23 Apr 2018

Summary:

The manuscript by Fry et al. addresses, for the first time, the potential to measure in-situ secondary organic aerosol (SOA) yields from isoprene oxidation in a power plant plume by aircraft. This is a completely original and timely study that aims to assess SOA yields in the ambient environment without the competing effects of wall loss, which has hampered most laboratory (reaction chamber) studies in the past. In this view, the paper is highly suitable for Atmospheric Chemistry and Physics. The authors determine isoprene-derived SOA yields from NO3 oxidation in the plume based on measured enhancements in aerosol organic nitrate and isoprene loss in the plume

relative to aerosol organic nitrate and isoprene concentrations outside of the plume. The authors find that isoprene-derived molar SOA yields from reaction with NO3 is on the order of 9%, and mass-based SOA yields are 27%, larger than those measured previously in the laboratory (12-14%). The authors conclude that the relatively larger SOA mass yield is due to the longer plume age and processing (forming more nitrates) compared to apparently shorter processing time in chamber studies. While I thought the paper was creative, well written, and well supported by the literature, before I can fully support publication, I encourage the authors to address my points of concern in a revised manuscript as stated below.

Major comments:

1. Although I thought the authors did their due diligence by addressing several of the caveats in this study, I have a couple of additional concerns (but possible solutions) with the calculation of SOA yield that I encourage the authors to address in a revised manuscript.

First, the authors use isoprene measured outside of the plume as the initial (starting) concentration and from that derive the SOA yield based on the difference in isoprene concentrations measured inside and outside of the plume. Ideally, I think you would want to use isoprene measured from the point of plume emission as the starting concentration of isoprene, i.e., measure the isoprene concentration in the plume near the point source, and then measure isoprene in the plume at a distance further downwind of the point source, because then you know how much of the initial isoprene in the plume (same air mass) was consumed. My main concern with using isoprene outside of the plume as the starting concentration is that it does not necessarily represent the isoprene that has undergone processing in the plume. According to the isoprene time series shown in Fig. 2, in the span of 5 minutes, isoprene outside of the plume can be 700 ppt, 500 ppt, and 300 ppt, for example. Thus, the SOA yields reported in this work depend critically on the choice of concentration measured outside of the plume. While I am not suggesting the authors are wrong in their approach, it might be helpful

if the authors could identify a case where they sampled the same plume twice at different locations downwind of the point source and calculate the SOA yield based on the difference in isoprene/nitrate measured in the first transect and a later transect. This would at least strengthen/validate the approach. Alternatively, it may help to show that "background" isoprene measured outside of the plume does not vary significantly near and further downwind of the plume source.

Second, what is the impact of $O_3$ (and other oxidants) on isoprene loss in the plume? I thought there would be more discussion of this – while the reaction rate of $O_3$ with isoprene is several orders of magnitude less than $NO_3$, the concentration of $O_3$ can be several orders of magnitude greater than $NO_3$, and therefore may rival $NO_3$ in regards to isoprene consumption in the plume at night. In the Edwards, et al. [2017] study referenced by the authors, $O_3$ accounts for 45% of the BVOC consumption at night. In this study, the SOA yield is based on the premise that VOC consumption is controlled entirely by $NO_3$. If other reactants that consume isoprene (e.g., $O_3$ and OH) are present in sufficient quantities, the calculated yields might overestimate the contribution from $NO_3$. I encourage the authors to address this more explicitly, e.g., by calculating the relative loss rates of isoprene at night by $NO_3$, $O_3$, and OH.

2. The scatter and limited number of observations used to calculate the average yield as shown in Fig. 5 may be a point of concern. Uncertainty bars on the data would certainly help to convey how far off from the fit the measurements truly are. Often, SOA mass yields are expressed as a function of the change in particle mass ($\Delta M$); if the authors were to instead plot plume change in pRONO2 mass as a function of plume change in isoprene mass, could it be that the larger/smaller enhancements in aerosol organic nitrate mass simply result from a shift in equilibrium partitioning more/less to the particle phase owing to a larger/smaller $\Delta M$? I encourage the authors to show the effects of $\Delta M$ in some capacity, e.g., by normalizing each point in Fig. 5 by the measured $\Delta M$ (i.e., difference in M between inside and outside of plume) and/or making a separate figure to show mass yield as a function of $\Delta M$. Alternatively, instead of using

√n as the bubble size in Fig. 5, scale bubble size by ΔM.

Minor comments

1. In the SOA molar yield calculation, the authors first convert the aerosol nitrate from mass concentration units to equivalent ppt assuming the aerosol organic nitrate has a molar mass of 62 g mol-1. This seems far too small a molar mass expected for isoprene+NO3 oxidation products. Why not assume a molar mass consistent with the first generation carbonyl nitrate produced from isoprene+NO3 (MW=145 g mol-1) (Jenkin et al., 2015) or another suitable organic compound as done later with the SOA mass yield calculation?

2. Page 2, line 52: "review"

3. Page 3, lines 92-94: Please include reference.

4. Page 9, Eq. 1 (lines 367-371): Equation (1) has k1, whereas text states k2.

5. Page 14, lines 500-502: It's probably more correct to write the production rate of isoprene oxidation products by NO3 reaction is greater than for monoterpenes.

6. Figure 5: It would be helpful to the readers if in the legend, the symbol for ΔpRONO2 were black with a color scale next to the current legend (the red color of the symbol is confusing with some of the points being red). A separate legend for marker/bubble size would also be helpful.

References

Edwards PM, et al. (2017) Transition from high- to low-NOx control of night-time oxidation in the southeastern US. Nature Geoscience 10(7):490-+.

Jenkin ME, Young JC, & Rickard AR (2015) The MCM v3.3.1 degradation scheme for isoprene. Atmos. Chem. Phys. 15:11433-11459.

[Figure]

2018.

---

## Referee Comment (RC3) · Anonymous Referee #3 · 5 May 2018

Fry et al use airborne observations from the SENEX campaign to infer SOA yields for the reaction of isoprene with NO3 radicals. Specifically they show that night time transects through power plant plumes capture conditions in which the loss of NO3 is dominated by the reaction with isoprene. Comparisons of out of plume isoprene and particle phase nitrate measurements with values observed in the seconds to minutes long in-plume parts of the flight, are used to calculate SOA molar and mass yields. While the approach of using field data to evaluate SOA yields in "wall free" environments is interesting, the data analysis is based on highly speculative assumptions and the SOA yields can therefore not be taken as reliable real world reference. The paper needs major modifications before it can be published.

**Major Points**

The particulate organic nitrate mass concentration is evaluated according to an established method using AMS observed NO2+/NO+ ion ratios. While this method has been used before for high resolution data sets, the authors have to apply corrections for unknown organic interferences to their C-TOF-AMS dataset, subtracting 55% and 33% of the total measured signal on m/z 30 and 46, respectively. As shown in Figure S2 e (lower panel), the thus derived UMR corrected NO2+/NO+ ratio agrees relatively well with the HR ratio, except for periods in which the total nitrate signal is low. The authors should have a look into this feature and derive from it a threshold total nitrate mass concentration below which no reliable analysis of organic nitrate is possible. Note that the values for R ammonium nitrate and R organic nitrate indicated in Figure S2 do not match with the values of 0.49 and 0.175 reported in the paper and in Figure 3.

The use of a value R=0.175 of NO2+/NO+ for organic nitrates is justified with reference to Day et al 2017, a paper in preparation. As the R-value directly affects the calculated mass concentration of organic nitrates, basing its justification on unpublished work is not acceptable. In a more conservative approach the authors should instead use the organic nitrate R-value of 0.1, which will lead to a lower estimate of organic nitrate mass concentration. Implementing this value for the data set in Table 2 would lead to a reduction of organic nitrate mass concentration by $\sim$ 25%, directly reducing the SOA molar and mass yields by the same percentage. Noteworthy, the use of R=0.1 for organic nitrates would also increase slightly the mass concentration of ammonium nitrate. As for many plumes the authors calculate negative ammonium nitrate mass concentration, this negative bias for the ammonium nitrate would be overcome, further supporting the use of R=0.1 instead of R=0.175. As mentioned above, the use of R=0.1 would reduce organic nitrate mass concentration and therefore the SOA mass yield would be reduced to $\sim$20% instead of the current 27%. Accounting for the 2/3 organic mass the SOA mass yield presented here would translate into an organic mass yield of 13%, well comparable to the literature data cited by the authors.

**[ACPD](ACPD)**

Interactive
comment

The discussion on urban plumes, although acknowledging uncertainties, is far too speculative and should be removed from the manuscript.

In view of the uncertainties in organic nitrate mass determination, the SOA molar and mass yields have to be reconsidered, accounting for above approach.

Other points (in order of appearance in the manuscript)

Page 5, line 172, 174: "$0.7\mu$gm-3...a factor of three lower than .... $1.7\mu$gm-3" the numbers don't match up, check for consistency.

Page 13, line 469: the nitrate radical production rate that was used to identify in-plume parts of the flight needs justification

Page 16, line 567 and following: To justify the statement, the authors need to show calibration data for deriving RIE of NH4 and show the precision of ion balance in the calibration aerosol.

Although the authors cite, that NO3 loss is dominated by reaction with isoprene, they could use the calculated potential for inorganic nitrate formation from N2O5 uptake to support the interpretation of most in-plume particulate nitrate formation having organic sources.

---

## Author Comment (AC1) · 9 Jun 2018

**Response to reviewers for the paper "Secondary Organic Aerosol (SOA) yields from NO3 radical + isoprene based on nighttime aircraft power plant plume transects" by J.L. Fry et al.**

We thank the reviewers for their careful reading of and thoughtful comments on our paper. To guide the review process we have copied the reviewer comments in black text. Our responses are in regular blue font. We have responded to all the referee comments and made alterations to our paper (**in bold text**).

**Overall response to reviews:**

Taken together, these three reviews suggest that the referees struggled with many of the same issues that we did as we worked through the data analysis and wrote this paper. With the help of reviewer suggestions, we have attempted to further clarify how we have ruled out some potential confounding effects and how we can constrain the likely impact of others on our results, to ensure a clear discussion of the strengths and limitations of this yield analysis. We do remain convinced that despite the limits of this small dataset, its unique strength as a direct measurement of $NO_3$ + isoprene SOA yields under conditions of atmospherically relevant peroxy radical lifetime merits publication and will make a useful contribution to our field. We hope that with these responses the reviewers and editor will support our reporting these results as SOA yields, with clear discussion of the attendant uncertainties. We do understand the reviewers' concerns about the small number of measurements and large uncertainties in these yields, so we have proposed edits to the figures and discussion to emphasize this further. While the number of data are limited and the uncertainties substantial, it is the case that chamber-derived yields may also have large uncertainties due to instrument uncertainties, unaccounted for gas losses to Teflon chamber walls, $RO_2$ fate relevance, etc. -- and there are also quite limited chamber data available on this reaction. Thus, we feel that this paper adds a valuable contribution to the literature. We believe that as revised, this manuscript will not prematurely induce modelers to substitute uncertain yields into their models, but instead this report of higher yield values than those previously observed will pique other researchers' interest, and thus spur valuable follow-up studies.

**Anonymous Referee #1**

The manuscript is original and very interesting to read. The authors tried to get the optimum out of the data, but still addressed openly the limitations of their approach.

From the viewpoint of raising interesting questions regarding the role of isoprene chemistry and isoprene NO3 chemistry for SOA formation and interesting approaches to address these questions, the paper could be published after some minor revisions (most of it of formal character, e.g. references in text and supplement, see below).

However the manuscript fails clearly short behind its title claim and from this point of view, I

suggest to reject the manuscript, due to the major concerns following below. Since the authors have already done the best with their data in a positive sense, I guess major revisions would not make sense.

A way out could be a reformulation of the title of the manuscript away from "providing yields" (reliable numbers) to a more procedural character of "addressing an important issue with interesting approaches and possibly important findings".

The basic observation we report in this paper is a change in particulate nitrate for a change in isoprene ($\Delta$pRONO2/$\Delta$Isop). As long as the $\Delta$pRONO2 is attributable to organic nitrate (with uncertainties clearly acknowledged), and as long as the association is plausibly the result of the isoprene lost, then this number can only be called a yield. Therefore, we argue that the term yield should be retained in the title, but that, as described below, we provide clear accounting of the uncertainties.

We propose to update Figure 5 with error bars to more clearly emphasize the uncertainties (see response R2.2. below) and adjust wording (see abstract text change below) to ensure that the reliability of our derived yields is appropriately discussed. This way, the yield numbers will not be taken to be an update from previous chamber studies, but rather, this will spur valuable further work to better constrain these yields under atmospherically relevant conditions.

The last sentence of the abstract has been edited to emphasize this goal: "**More in-depth** studies are needed to better understand the **aerosol yield and** oxidation mechanism of $NO_3$ radical + isoprene, a coupled anthropogenic – biogenic source of SOA **that may be regionally significant.**"

Major:

R1.1. The authors convinced me that pRONO2 and thus organic nitrate in plumes is enhanced and that may indeed relate to enhanced NO3 concentrations (Figure 6). However, the paper does not really show that that increase of pRONO2 is related to isoprene oxidation alone (Figure 5). While the reasoning of a single -ONO2 group per organic nitrate molecules is an acceptable approach to derive molar yields, the scatter in Figure 5 casts doubts, if the increase of pRONO2 is really related solely to isoprene oxidation. Herein the weak point is the limited number of data points. I don't say the authors are wrong, but one would need more observations to strengthen the case. I concede that the authors revealed an interesting phenomenon, interesting enough to pursue the ideas and go out and get more/better proof.

We agree that there are not many data points in Figure 5, although the paper describes in detail how many power plant plume intercepts were available and how many were suitable for analysis. Thus the data set is by its nature unavoidably limited. Nevertheless, the increase in pRONO2 associated with each isoprene depletion is clear and repeatable, such that there is not another, more plausible explanation for the observation of pRONO2 enhancement caused by

rapid NO$_3$ oxidation of isoprene. When these points are displayed in the format of Figure 5, they produce considerable scatter, indicating that the same yield is not necessarily obtained for each plume, or that the uncertainty in the determination is large. One potential reason for scatter in the data is that the plumes are not all of the same age, as the color code indicates. To clearly show the uncertainty in yields, we have modified Figure 5 to show the error bars that are associated with the yield determination.

R1.2. L: 651: In going from the molar yield to the mass yield the uncertainty - and speculations clearly indicated as such, though - become even larger. On one hand obviously two oxidation steps are needed to achieve condensable isoprene oxidation products, on the other hand NO3 seems to be the only available oxidant. Oxidation of both double bonds should thus lead to dinitrates.

Two oxidation steps are needed only if auto-oxidation is an unimportant mechanism.

If pOrgNO3 would really isoprene dinitrates the estimated yield would drop from 27% to 18%, not so far away from the referenced value by Rollins of 14%. I can follow the authors that it is likely thatpRONO2 dinitrates could be hydrolysed, but why should hydrolysis stop after one group, why not hydrolysing every second -ONO2 group ore even both ONO2 groups? Moreover, as far as I understand, Rollin's value is based in parts on observations of several hours in a large chamber. So reaction time cannot be an issue?!

We agree with the reviewer's comment that the mass or molar yield would be different by a factor of two in the case of both double bonds oxidized by NO$_3$ and retention of both nitrate groups on the isoprene backbone.  We do not understand the statement that reaction time cannot then be an issue.  Especially in the case of a two step oxidation, with a slower rate constant anticipated for the second step, additional reaction time would be required, as stated in the manuscript.

I agree that wall losses could be an issue, though, but wall losses are also less important in large chambers.

Wall losses are not unimportant in large chambers.  Cited references show that partitioning of semivolatile organic compounds to walls is an important effect in any chamber study.

There for with the same right I could argue that Rollin's yield of 14% is correct and then ask where could the rest of organic nitrate come from. I follow the authors that inorganic nitrate can be excluded as source. Could it be that the organic nitrates arise from liquid phase or heterogeneous processes via NO2, NO3? Is anything known about such heterogeneous nitration processes? NOX and NO3 were by definition high in the plumes. Actually if I really think about it the mass yield analysis adds not much beyond the molar yield considerations and an analogous plot would just reproduce Figure 5 with slopes of 18% or 27%, depending only on the assumption if the isoprene nitrates bring in two or three times the molecular weight of isoprene itself.

We concur that the mass yield estimate is subject to substantial additional uncertainties beyond the molar yield, which we have endeavored to describe clearly here. Because it took the author team quite some time and discussions to come up with all of these considerations, we thought it could be helpful to readers to have them collected here in one place. Because of the noted uncertainties, however, we choose not to emphasize these mass yields with a figure, instead showing molar yields in Figure 5. This way a future reader with additional information about likely reaction mechanisms or SOA composition could do exactly the calculation this reviewer does to determine refined mass yield estimates based on that new information.

In response to the suggestion to consider heterogeneous uptake of $NO_3$ onto organic particles, we make an estimate of the rate of that process to determine whether it might contribute to observed organic nitrate aerosol. Based on available literature, the maximum $NO_3$ uptake coefficient would be 0.1, and this condition would only occur if there are a significant number of double bonds remaining in the newly formed organic aerosol [Ng review paper, p. 2114]. Given this uptake coefficient and the observed in-plume wet aerosol surface of on average 300 um$^2$ cm$^{-3}$ (=3x10$^{-4}$ m$^{-1}$), the kinetic molecular theory predicted uptake rate constant is k = {gamma}*v*SA/4 = 0.0024 s$^{-1}$. At average in-plume [$NO_3$] of 20 pptv, this would correspond to an uptake of 0.17 ppb $NO_3$ per hour. This means that a 5 to 6-hour old plume could have up to ~1 ppb of nitrate functional groups produced by this heterogeneous process if this high uptake coefficient is true. However, because the aerosol surface area is not exclusively alkene nor even exclusively organic aerosol (indeed, it is calculated to be partially aqueous), we expect that a much smaller uptake coefficient, on the order of 0.001, is more realistic, and thus heterogeneous $NO_3$ uptake is not likely to contribute significantly. (Brown & Stutz 2012).

We have added the following line to the revised manuscript at line 733: "**We have not corrected the calculated yields for the possibility of $NO_3$ heterogeneous uptake, which could add a nitrate functionality to existing aerosol. Such a process could be rapid if the uptake coefficient for $NO_3$ were 0.1, a value characteristics of unsaturated substrates (Ng et al, 3016), but would not contribute measurably at more conventional $NO_3$ uptake coefficients of 0.001 (Brown & Stutz 2012).**"

R1.3. L483/ Fig. 2: What also concerns me and this is again related to small number of cases: There are indeed correlations between PNO3 andpRONO2 and anti-correlation with isoprene, but pOrg NO3 sometimes increases by the same amount in the absence of plumes (2:17.30AM, 2:22.00AM) and some plumes do not create OrgNO3 despite lower isoprene (ca. 2:21.20AM).

We again concur that this study would be stronger with a larger number of observed plume transects, however we reassert that our screening methodology and error limits have all been stated in the manuscript. We have considered the variations in the background in making estimates of the organic nitrate increases in plumes. We average as many points as possible in each of multiple plumes.

Minor:

R1.4. l315, Fig.S1: If I compare the SENEX data with actual data in the range of plumes and background the difference is more a factor of 2 than 1.6. Moreover, two of the plumes fall off line while all the background measurements correlate as all other data. Unfortunately, the exception of AMS performance(?) or UHSAS performance(?) for "just that flight" in addition weakens the case.

The right-hand panel of Figure S1 is reproduced below with a 2:1 line shown, to illustrate that the slope of 1.6 is correct for this flight.

[Figure]

As far as the scatter in the plume enhancements and the contribution of this volume related uncertainty -- we now cite forward to Figure 4, where the enhancements are shown with complete propagated error on pRONO2, showing that while there are uncertainties due to (among other things) the uncertainty in the volume comparison, the enhancements are always positive. These same error bars are now also shown on Figure 5, so that the full uncertainties are available for the reader to evaluate.

We also noted in the process of updating this that the uncertainties in the yield tables had not incorporated these additional uncertainties -- these have all been updated to include the full error propagation. (Note: this did not change yields and increased errors on only some plumes)

Changed text to: "This does not change the conclusions of this work **because this has been incorporated into the error in aerosol organic nitrate, which still show positive enhancements in pRONO2 for these plumes (see Figure 4 below). These complete error estimates are also used in Figure 5 to clearly show the uncertainties in the yields**. The volume comparison is discussed further in the Supplemental Information and shown for the plumes of interest in Fig. S1

R1.5. I suggest listing also PNO3 in Table 1; that would help to link quickly oxidation strength and observed effect

This has been added to Table 1.

R1.6. Main text: references not in ACP format Replace "author et al. (author et al., year)" by "author et al. (year)"

Fixed.

R1.7. Supplement: the literature is not assessable and given in bracketed format

Fixed.

R1.8. l59: review

This appears to be an editor's note.

l172: Xu et al. (2015)

Fixed.

l339: Fry et al. [47]
Fixed.

l657: no "N" in the formula, it is not clear that refer only to organic rest of the trihydroxynitrate

Thank you, clarified: "e.g. a tri-hydroxynitrate (**with organic portion of formula** $C_5H_{11}O_3$, 119 g $mol^{-1}$)"

l418: I suggest to replace "number densities" with "concentration" in context of gases I

Done.

**Anonymous Referee #2**
Summary: The manuscript by Fry et al. addresses, for the first time, the potential to measure in-situ secondary organic aerosol (SOA) yields from isoprene oxidation in a power plant plume by aircraft. This is a completely original and timely study that aims to assess SOA yields in the ambient environment without the competing effects of wall loss, which has hampered most laboratory (reaction chamber) studies in the past. In this view, the paper is highly suitable for Atmospheric Chemistry and Physics. The authors determine isoprene-derived SOA yields from NO3 oxidation in the plume based on measured enhancements in aerosol organic nitrate and

isoprene loss in the plume relative to aerosol organic nitrate and isoprene concentrations outside of the plume. The authors find that isoprene-derived molar SOA yields from reaction with NO3 is on the order of 9%, and mass-based SOA yields are 27%, larger than those measured previously in the laboratory (12-14%). The authors conclude that the relatively larger SOA mass yield is due to the longer plume age and processing (forming more nitrates) compared to apparently shorter processing time in chamber studies. While I thought the paper was creative, well written, and well supported by the literature, before I can fully support publication, I encourage the authors to address my points of concern in a revised manuscript as stated below.

Major comments:

R2.1a. Although I thought the authors did their due diligence by addressing several of the caveats in this study, I have a couple of additional concerns (but possible solutions) with the calculation of SOA yield that I encourage the authors to address in a revised manuscript. First, the authors use isoprene measured outside of the plume as the initial (starting) concentration and from that derive the SOA yield based on the difference in isoprene concentrations measured inside and outside of the plume. Ideally, I think you would want to use isoprene measured from the point of plume emission as the starting concentration of isoprene, i.e., measure the isoprene concentration in the plume near the point source, and then measure isoprene in the plume at a distance further downwind of the point source, because then you know how much of the initial isoprene in the plume (same air mass) was consumed. My main concern with using isoprene outside of the plume as the starting concentration is that it does not necessarily represent the isoprene that has undergone processing in the plume. According to the isoprene time series shown in Fig. 2, in the span of 5 minutes, isoprene outside of the plume can be 700 ppt, 500 ppt, and 300 ppt, for example. Thus, the SOA yields reported in this work depend critically on the choice of concentration measured outside of the plume. While I am not suggesting the authors are wrong in their approach, it might be helpful if the authors could identify a case where they sampled the same plume twice at different locations downwind of the point source and calculate the SOA yield based on the difference in isoprene/nitrate measured in the first transect and a later transect. This would at least strengthen/validate the approach. Alternatively, it may help to show that "background" isoprene measured outside of the plume does not vary significantly near and further downwind of the plume source.

We thank the reviewer for these suggestions. For each plume point, we used an iterative box model to calculate the isoprene that would have been present at sunset at that location outside of the NOx plume. This enables an alternate Δisoprene calculation based on in-plume isoprene minus modeled sunset isoprene, for comparison to the calculation used in the yield calculations, based on in-plume minus background isoprene. The similarity between these two values for most points suggests that the isoprene just outside of each plume transect was largely unperturbed from the sunset initial value. We have added these values to Table S3, with explanatory text:

"**Also shown are the plume changes in isoprene used in the present analysis (Δisop, the**

**difference between in-plume and background isoprene concentration, reproduced from Table 1), alongside for comparison the Δisop determined as the difference between in-plume isoprene and the modeled sunset (initial) concentration of isoprene present at that location outside of the plume, determined using an iterative box model (ref). The similarity between these two values for most points suggests that the isoprene just outside of each plume transect was largely unperturbed from the sunset initial value."**

| plume number [#isop/#AMS] | 7/2/13 plume time (UTC) | $\Delta ORG_{aero}$ ($\mu g\ m^{-3}$) | $\Delta NH_{4,aero}$ ($\mu g\ m^{-3}$) | $\Delta SO_{4,aero}$ ($\mu g\ m^{-3}$) | Temp (C) | %RH | Δisop (pptv) | Δisop from model (pptv) | Isop:MT Mole Ratio |
|---|---|---|---|---|---|---|---|---|---|
| Typical variability ($\mu g\ m^{-3}$): | | 0.75 | 0.1 | 0.5 | | | | | |
| 1 [2/3] | 2:18 | 0.35 | 0 | 0 | 23.6 | 66.5 | -335 | -327 | 36.5 |
| 2 [*] | 2:20 | 0.89 | 0.3 | 1.91 | 23.6 | 65 | -404 | -453 | 71.4 |
| 3 [4/5] | 2:21 | 1.25 | 1.05 | 5.14 | 23.6 | 65.2 | -228 | -337 | 16.6 |
| 4 [*] | 3:03 | 0.16 | 0.08 | 0.7 | 21.2 | 68.1 | -453 | -391 | 50.6 |
| 5 [3/4] | 3:55 | 0.32 | 0.26 | 6.07 | 21.9 | 65.5 | -255 | -376 | 34.2 |
| 6 [2/2] | 4:34 | 0.57 | 0.3 | 1.12 | 19.9 | 74.6 | -713 | -233 | 17.3 |
| 7 [5/6] | 4:37 | 1.05 | 0.22 | 0.65 | 19.7 | 76.2 | -298 | -221 | 14.2 |
| 8 [2/3] | 4:39 | 1.26 | 0.44 | 1.18 | 18.3 | 82.2 | -443 | -353 | 11.0 |
| 9 [7/8] | 5:04 | 1.45 | 0.35 | 1.9 | 17.2 | 84.8 | -293 | -434 | 17.8 |

This will allow the reader to assess the general robustness of the isoprene background values. However, we don't believe that it would be appropriate to calculate yields based on these values, because we don't have analogous pre-plume values for pRONO2. Thus, for the yield calculations we think it's best to use in-plume and plume-adjacent backround isoprene values even though there is noise in the background. We do account for this noise in the standard deviation error bars on the Δisop values.

We appreciate the second suggestion to use multiple, successive downwind plume transects. While this approach has worked in previous analysis of nighttime power plant plumes (e.g., Brown et al. 2012), identifying such plumes is difficult. In the present case, there were not two easily identifiable successive Lagrangian plumes intercepts. Rather, we encountered a range of plumes, often at different altitudes, with different transport times but not in a successive manner.

R2.1b. Second, what is the impact of O3 (and other oxidants) on isoprene loss in the plume? I thought there would be more discussion of this – while the reaction rate of O3 with isoprene is several orders of magnitude less than NO3, the concentration of O3 can be several orders of magnitude greater than NO3, and therefore may rival NO3 in regards to isoprene consumption in the plume at night. In the Edwards, et al. [2017] study referenced by the authors, O3 accounts for 45% of the BVOC consumption at night. In this study, the SOA yield is based on the premise that VOC consumption is controlled entirely by NO3. If other reactants that consume isoprene (e.g., O3 and OH) are present in sufficient quantities, the calculated yields might overestimate the contribution from NO3. I encourage the authors to address this more explicitly, e.g., by calculating the relative loss rates of isoprene at night by NO3, O3, and OH.

We have added to the supplemental information the below plot (new Figure S9) showing isoprene loss in the plume model simulation (black) and stacked plot showing the contributions to this from the $NO_3$, $O_3$, and OH. As described in the model description, the modelled plume was emitted at sunset so these are all nocturnal processes. As is clear from this figure, in the power plant plumes, the isoprene loss is not entirely, but approximately 90% via $NO_3$ radical.

[Figure]

**Figure S9.** Model simulation of typical in-plume consumption of isoprene (black line), and stacked plot showing the contributions to this from the $NO_3$, $O_3$, and OH. Modeled plume was emitted at sunset, so this represents nocturnal processing under power plant plume conditions.

R2.2. The scatter and limited number of observations used to calculate the average yield as shown in Fig. 5 may be a point of concern. Uncertainty bars on the data would certainly help to convey how far off from the fit the measurements truly are. Often, SOA mass yields are expressed as a function of the change in particle mass ($\Delta M$); if the authors were to instead plot plume change in pRONO2 mass as a function of plume change in isoprene mass, could it be that the larger/smaller enhancements in aerosol organic nitrate mass simply result from a shift in equilibrium partitioning more/less to the particle phase owing to a larger/smaller $\Delta M$? I encourage the authors to show the effects of $\Delta M$ in some capacity, e.g., by normalizing each point in Fig. 5 by the measured $\Delta M$ (i.e., difference in M between inside and outside of plume) and/or making a separate figure to show mass yield as a function of $\Delta M$. Alternatively, instead of using $\sqrt{n}$ as the bubble size in Fig. 5, scale bubble size by $\Delta M$.

We thank the reviewer for these insightful suggestions. We tried re-plotting Figure 5 with points colored by $\Delta M$, and don't see a clear dependence that explains the high points (see figure below, not added to manuscript). Given this, we believe the colorbar in the manuscript shows a more likely contributing factor: plume age. The highest pRONO2 values occur for the longest plume ages, which would allow for several routes to more pRONO2: (1) more molecules in with

the second double bond has been oxidized, (2) more time for intramolecular H-rearrangement reactions, or (3) more time for contribution of (slower) heterogeneous uptake of NO3 on organic aerosol.

[Figure]

We have added uncertainty error bars, thank you for that nudge. The new Figure 5 and updated caption are:

[Figure]

**Figure 5.** SOA molar yield can be determined as the slope of $\Delta$pRONO$_2$ vs. $\Delta$isoprene, both in mixing ratio units. The linear fit is weighted by square root of number of points used to determine each in-plume pRONO$_2$, with intercept held at zero. The slope coefficient ± one standard deviation is 0.0930 ± 0.0011. Points are colored by plume age (red = longest), and size scaled by square root of number of points (the point weight used in linear fit). This plot and fit includes the nine plumes listed in Tables 1 and 2, as well as the 03:14 "unreacted" plume (at $\Delta$isoprene = -84 ppt). **Error bars on isoprene are the propagated standard deviations of the (in plume - out plume) differences, for plumes in which multi-point averages were possible. Error bars on pRONO2 are the same as in Figure 4. The points without error bars are single-point plumes.**

These errors bars are the propagated standard deviations of the {in plume - out plume} differences for plumes in which multi-point averages were possible (the points without error bars are single-point plumes). This responds also to Reviewer 1's concern about clearly demonstrating the uncertainties in the derived yields, and the x error bars respond to comment R2.1a. above about the variability of isoprene around these plumes.

Minor comments

R2.3. In the SOA molar yield calculation, the authors first convert the aerosol nitrate from mass concentration units to equivalent ppt assuming the aerosol organic nitrate has a molar mass of 62 g mol-1. This seems far too small a molar mass expected for isoprene+NO3 oxidation products. Why not assume a molar mass consistent with the first generation carbonyl nitrate produced from isoprene+NO3 (MW=145 g mol-1) (Jenkin et al., 2015) or another suitable organic compound as done later with the SOA mass yield calculation?

The nitrate measurement by the AMS is calibrated to be the mass of the nitrate (NO3) moiety alone, hence, 62 g mol$^{-1}$, and we use these masses, converted to mixing ratio, in order to determine molar yields. Using the nitrate component alone avoids needing to make any assumption about the molecular weight of the organic mass that accompanies the NO3 in the produced SOA. We thus can make these assumptions separately to estimate a mass yield (see equations 3 and 4 and discussion thereof). To ensure clarity in the text, added this text:

"we convert the aerosol organic nitrate mass loading differences to mixing ratio differences (ppt) using the NO$_3$ molecular weight of 62 g mol$^{-1}$ **(the AMS organic nitrate mass is the mass only of the –ONO$_2$ portion of the organonitrate aerosol)**."

R2.4. Page 2, line 52: "review"

This appears to be an editor's note.

R2.5. Page 3, lines 92-94: Please include reference.

Added reference to: D'Ambro, E. L., K. H. Møller, F. D. Lopez-Hilfiker, S. Schobesberger, J. Liu, J. E. Shilling, B. H. Lee, H. G. Kjaergaard and J. A. Thornton (2017). "Isomerization of Second-Generation Isoprene Peroxy Radicals: Epoxide Formation and Implications for Secondary Organic Aerosol Yields." Environmental Science & Technology **51**(9): 4978-4987.

R2.6. Page 9, Eq. 1 (lines 367-371): Equation (1) has k1, whereas text states k2.

Thank you! Corrected.

R2.5. Page 14, lines 500-502: It's probably more correct to write the production rate of isoprene oxidation products by NO3 reaction is greater than for monoterpenes.

As suggested we modified this line to read: "At these relative concentrations, even if all of the monoterpene is oxidized, **the production rate of oxidation products** will be much larger for isoprene."

R2.6.  Figure 5: It would be helpful to the readers if in the legend, the symbol for ∆pRONO2 were black with a color scale next to the current legend (the red color of the symbol is confusing with some of the points being red). A separate legend for marker/bubble size would also be helpful.

Thank you, done, and color bar legend added (see new version of figure above in response to R2.2.

**Anonymous Referee #3**
Fry et al use airborne observations from the SENEX campaign to infer SOA yields for the reaction of isoprene with NO3 radicals. Specifically they show that night time transects through power plant plumes capture conditions in which the loss of NO3 is dominated by the reaction with isoprene. Comparisons of out of plume isoprene and particle phase nitrate measurements with values observed in the seconds to minutes long in-plume parts of the flight, are used to calculate SOA molar and mass yields. While the approach of using field data to evaluate SOA yields in "wall free" environments is interesting, the data analysis is based on highly speculative assumptions and the SOA yields can therefore not be taken as reliable real world reference. The paper needs major modifications before it can be published.

Major Points

R3.1. The particulate organic nitrate mass concentration is evaluated according to an established method using AMS observed NO2+/NO+ ion ratios. While this method has been used before for high resolution data sets, the authors have to apply corrections for unknown organic interferences to their C-TOF-AMS dataset, subtracting 55% and 33% of the total measured signal on m/z 30 and 46, respectively. As shown in Figure S2 e (lower panel), the thus derived UMR corrected NO2+/NO+ ratio agrees relatively well with the HR ratio, except for periods in which the total nitrate signal is low. The authors should have a look into this feature

and derive from it a threshold total nitrate mass concentration below which no reliable analysis of organic nitrate is possible.

Detection limits (DL) for nitrate using HR-ToF-AMS (HR data shown in Fig. S2) are ~10 ng/m3. As either $NO_x^+$ ion approaches their DL (and zero), the uncertainty in the $NO_x^+$ ratio determinations will blow up. This effect is clearly visible in both the HR and UMR -derived $NO_x^+$ ratios in Fig. S2. The DLs for $NO_2^+$ and $NO^+$ are similar to the nitrate DL. Depending on the instrument-specific response and the proportions of inorganic/organic nitrate, the $NO_2^+/NO^+$ ratio can vary between ~1 and ~0.1. Therefore, the $NO_x^+$ ratio detection limit is typically dominated by the $NO_2^+$ ion DL, especially when the nitrate is dominated by pRONO$_2$. So for HR-ToF AMS that would be equivalent to a total nitrate concentration of ~50 ng/m3. Importantly, when the $NO_x^+$ ratio is below DL, discarding pRONO$_2$ and ammonium nitrate concentration data is not necessarily warranted or desired, since despite that apportionment may be indeterminate, the concentration of both are still constrained to the nitrate concentration (if above the total nitrate DL) or the nitrate DL (if below the total nitrate DL) which is often valuable information and places quantitative constraints on concentrations. Given these considerations, we do not think there is a "threshold total nitrate mass concentration below which no reliable analysis of organic nitrate is possible".

For this study, the nitrate DLs (3-sigma) reported here for the native 10-second CToF data were 50 ng/m3 (L305) which for the upper limit of pure pRONO$_2$, where only ~20% of the $NO_x^+$ ions are $NO_2^+$, would correspond to a nitrate DL of 250 ng/m$^3$ for the $NO_x^+$ ratio. As seen in Fig. 6, all the plume pRONO$_2$ concentrations were between 200-600 ng/m3 (and total nitrate was similar or higher) and additionally the AMS plume averages typically consisted of several points (1-8) for which the combined DL should scale down as 1/sqrt(n). Therefore, for the plume analysis used in this manuscript, the pRONO$_2$ concentration determination should be near or well above expected 3-sigma DLs.

Note that the values for R ammonium nitrate and R organic nitrate indicated in Figure S2 do not match with the values of 0.49 and 0.175 reported in the paper and in Figure 3.

That is correct, the $NO_x^+$ ratios for ammonium nitrate in Fig. S2 are from calibrations conducted during that campaign (SEAC$^4$RS) while those in Fig. 3 are from the campaign investigated in this manuscript (SENEX). For both cases, the pRONO$_2$ ratio was estimated as 2.8 times lower $NO_2^+/NO^+$ ratio than measured for ammonium nitrate (see details in manuscript and below).

The use of a value R=0.175 of NO2+/NO+ for organic nitrates is justified with reference to Day et al 2017, a paper in preparation. As the R-value directly affects the calculated mass concentration of organic nitrates, basing its justification on unpublished work is not acceptable. In a more conservative approach the authors should instead use the organic nitrate R-value of 0.1, which will lead to a lower estimate of organic nitrate mass concentration. Implementing this value for the data set in Table 2 would lead to a reduction of organic nitrate mass concentration by ~ 25%, directly reducing the SOA molar and mass yields by the same percentage.

Noteworthy, the use of R=0.1 for organic nitrates would also increase slightly the mass concentration of ammonium nitrate. As for many plumes the authors calculate negative ammonium nitrate mass concentration, this negative bias for the ammonium nitrate would be overcome, further supporting the use of R=0.1 instead of R=0.175. As mentioned above, the use of R=0.1 would reduce organic nitrate mass concentration and therefore the SOA mass yield would be reduced to ~20% instead of the current 27%. Accounting for the 2/3 organic mass the SOA mass yield presented here would translate into an organic mass yield of 13%, well comparable to the literature data cited by the authors.

We agree that the $pRONO_2$ $NO_x^+$ ratio affects $pRONO_2$ quantification. However, we disagree with the proposed value, which is not consistent with the average of the published literature.

As for the "negative ammonium nitrate mass concentration" when calculating the plume enhancements, these cannot be simply prescribed to a bias in the $pRONO_2$ $NO_x^+$ ratio. Note that those values are differences between in/out of plume. As shown in Fig. 4, they are statistically zero when considering the uncertainties derived from the variability associated with in/out plume subtraction and measurement uncertainties (as clearly shown in the error bars on that plot). Therefore this does not provide any evidence that a $pRONO_2$ $NO_x^+$ ratio of 0.1 is more appropriate.

We have replaced the text in question describing the $pRONO_2$ ratio used with the following text, and removed all references to Day et al. from the manuscript:

**This factor was determined as the average of several literature studies (Fry et al., 2009; Rollins et al., 2009; Farmer et al., 2010; Sato et al., 2010; Fry et al., 2011; Boyd et al., 2015) and applied according to the "ratio of ratios" method (Fry et al., 2013).**

R3.2. The discussion on urban plumes, although acknowledging uncertainties, is far too speculative and should be removed from the manuscript.

We agree that the urban plume cases are more difficult to analyze due to variability in the background that is on the same scale as the enhancements. Therefore, we have moved this figure and discussion to the supplement to make this observation available, with only a qualitative analysis that organic nitrate aerosol is also enhanced in these urban plumes but is superimposed on an apparently large background variability.

Other points (in order of appearance in the manuscript)
R3.3. Page 5, line 172, 174: "0.7µgm-3. . .a factor of three lower than . . .. 1.7µgm-3" the numbers don't match up, check for consistency.

Edited to: "Xu et al. predict only 0.7 µg m$^{-3}$ of SOA would be produced, **substantially lower than** the measured nighttime LO-OOA production of 1.7 µg m$^{-3}$."

R3.4. Page 13, line 469: the nitrate radical production rate that was used to identify in-plume parts of the flight needs justification

Added text to explain: "**This threshold was chosen to be above background noise and large enough to isolate only true plumes (see Fig. 1a). The value is thus subjectively chosen, but was consistently applied across the dataset.**"

R3.5. Page 16, line 567 and following: To justify the statement, the authors need to show calibration data for deriving RIE of NH4 and show the precision of ion balance in the calibration aerosol.

The values for the relative ionization efficiencies for ammonium are mentioned in the experimental section on page 9 in lines 357-358: "Note that the relative ionization efficiency for ammonium was 3.91 and 3.87 for the two bracketing calibrations and an average value of 3.9 was used for the flight analyzed here."

The ion balance for the plume enhancements is now plotted with the ion balance for the ammonium nitrate calibration data along with uncertainty bands and error bars (new Figure S5b shown below & added to manuscript).

[Figure]

Figure 5. … **(b) Measured vs. calculated (ion balanced) NH$_4$ for calibration data and plume enhancements. This also shows that plumes are acidic than ammonium sulfate, ruling out the possibility of inorganic nitrate formation.**

Added at line 647 to describe the ion balance precision:

**"The ion balance for the ammonium nitrate calibration particles and the plume enhancements are shown in Fig. S5b. Complete neutralization of the calibration aerosols is nearly always within the gray 10% uncertainty band for the relative ionization efficiency of ammonium (Bahreini et al., 2009). In contrast, many of the plume enhancements are near the 1:2 line (as primarily ammonium bisulfate) within the combined 10% ammonium and 15% sulfate uncertainty error bars or without ammonium (sulfuric acid)."**

R3.6. Although the authors cite, that NO3 loss is dominated by reaction with isoprene, they could use the calculated potential for inorganic nitrate formation from N2O5 uptake to support the interpretation of most in-plume particulate nitrate formation having organic sources.

The contribution of $N_2O_5$ uptake to overall $NO_3$ losses was considered in detail in Edwards, et al. (2017). The results reported in Figure S4 show $N_2O_5$ heterogeneous uptake contributing negligibly, with the exception of 2 brief periods, which do not correspond to plumes analyzed in this work. The authors further argue that even this small contribution of $N_2O_5$ heterogeneous uptake is likely overestimated.

Added this line to the text at line 561 to clarify this:

**"Inorganic nitrate can also be produced by the heterogeneous uptake of $N_2O_5$ onto aqueous aerosol; Edwards et al. (2017) demonstrated that this process is negligible relative to $NO_3$ + BVOC for the July 2 SENEX night flight considered here."**

References for these responses:

S.S. Brown and J. Stutz, "Nighttime radical observations and chemistry," *Chem. Soc. Rev.*, 2012, 41, 6405-6447.

Brown, S.S., W.P. Dubé, P. Karamchandari, G. Yarwood, J. Peischl, T.B. Ryerson, J.A. Neuman, J.B. Nowak, J.S. Holloway, R.A. Washenfelder, C.A. Brock, G.J. Frost, M. Trainer, D.D. Parrish, F.C. Fehsenfeld, and A.R. Ravishankara, The effects of NOx control and plume mixing on nighttime chemical processing of plumes from coal-fired power plants. *J. Geophys. Res.*, 2012. 117: p. D07304.

---

## Referee Report (RR1)

**Summary:**

I am overall satisfied by the author's responses and changes to the manuscript. Together, the revisions make for a substantially improved manuscript, and do not change the conclusions. To my knowledge, there is no other study like this, and I appreciate the depth to which the authors have addressed the limitations of their study. In particular, new Fig. S9 clearly demonstrates the importance of $NO_3$ in the consumption of isoprene in the plume at night, and the error bars in Fig. 5 indicate clearly the variability in the derived yields. This change should be regarded not as skepticism of the yields, but rather a clear statement of the limitations of their measurement, which is not grounds for manuscript rejection.

While, in hindsight, there are some improvements to the experiments I'm sure the authors would like to have made to ensure the organic nitrates being detected in the plume were from isoprene+$NO_3$ (described below), this study represents a first approach and should be regarded as foundation for similar studies in the future. One such improvement to the experiment the authors might consider in the future is to use chemical ionization mass spectrometry (CIMS), or another complementary tool to the AMS, for qualitative or quantitative detection of isoprene+$NO_3$ oxidation products. Recent studies by Slade et al. (2017) and Lee et al. (2016) highlight this capability for monoterpene- and isoprene-derived organic nitrates.

My only criticism of the current manuscript is that the reported yields need hard uncertainty numbers. Given the error bars shown in Fig. 5, instead of using the standard deviation of the slope coefficient as the uncertainty, the authors should show how the slope varies accounting for the error in $\Delta$isoprene and $\Delta pRONO_2$. If you consider the max $\Delta pRONO_2 \sim 100$ ppt at $\Delta$isoprene $\sim 400$ ppt, the upper limit to the yield is $\sim 25\%$. In contrast, it appears the lower limit of the yield ($\Delta pRONO_2 \sim 20$ ppt at $\Delta$isoprene $\sim 700$ ppt) is $\sim 3\%$. Therefore, a more accurate organic nitrate yield with uncertainty might be 9(+14/-6)%.

**References**

Lee, B. H., Mohr, C., Lopez-Hilfiker, F. D., Lutz, A., Hallquist, M., Lee, L., Romer, P., Cohen, R. C., Lyer, S., Kurten, T., Hu, W., Day, D. A., Campuzano-Jost, P., Jimenez, J. L., Xu, L., Ng, N. L., Guo, H., Weber, R. J., Wilde, R. J., Brown, S. S., Koss, A., de Gouw, J., Olson, K., Goldstein, A. H., Seco, R., Kim, S., McAvey, K. M., Shepson, P. B., Starn, T. K., Baumann, K., Edgerton, E. S., Liu, J., Shilling, J. E., Miller, D. O., Brune, W., Schobesberger, S., D'Ambro, E. L., and Thornton, J. A.: Highly functionalized organic nitrates in the southeast United States: contribution to secondary organic aerosol and reactive nitrogen budgets, Proc. Nat. Acad. Sci., 113, 1516-1521, 10.1073/pnas.1508108113, 2016.

Slade, J. H., de Perre, C., Lee, L., and Shepson, P. B.: Nitrate radical oxidation of γ-terpinene: hydroxy nitrate, total organic nitrate, and secondary organic aerosol yields, Atmos. Chem. Phys., 10.5194/acp-2017-249, 2017.

---

## Author Response (AR2)

**Response to 2nd round reviews for the paper "Secondary Organic Aerosol (SOA) yields from NO3 radical + isoprene based on nighttime aircraft power plant plume transects" by J.L. Fry et al.**

We thank the reviewers for their careful reading of and thoughtful comments on our paper. To guide the review process we have copied the reviewer comments in black text. Our responses are in regular blue font. We have responded to all the referee comments and made alterations to our paper (**in bold text**).

Responses to 2nd round reviews:

**Anonymous Referee #1**

I am overall satisfied by the author's responses and changes to the manuscript. Together, the revisions make for a substantially improved manuscript, and do not change the conclusions. To my knowledge, there is no other study like this, and I appreciate the depth to which the authors have addressed the limitations of their study. In particular, new Fig. S9 clearly demonstrates the importance of NO3 in the consumption of isoprene in the plume at night, and the error bars in Fig. 5 indicate clearly the variability in the derived yields. This change should be regarded not as skepticism of the yields, but rather a clear statement of the limitations of their measurement, which is not grounds for manuscript rejection.

While, in hindsight, there are some improvements to the experiments I'm sure the authors would like to have made to ensure the organic nitrates being detected in the plume were from isoprene+NO3 (described below), this study represents a first approach and should be regarded as foundation for similar studies in the future. One such improvement to the experiment the authors might consider in the future is to use chemical ionization mass spectrometry (CIMS), or another complementary tool to the AMS, for qualitative or quantitative detection of isoprene+NO3 oxidation products. Recent studies by Slade et al. (2017) and Lee et al. (2016) highlight this capability for monoterpene- and isoprene-derived organic nitrates.

My only criticism of the current manuscript is that the reported yields need hard uncertainty numbers. Given the error bars shown in Fig. 5, instead of using the standard deviation of the slope coefficient as the uncertainty, the authors should show how the slope varies accounting for the error in Δisoprene and ΔpRONO2. If you consider the max ΔpRONO2~100 ppt at Δisoprene ~400 ppt, the upper limit to the yield is ~25%. In contrast, it appears the lower limit of the yield (ΔpRONO2~20 ppt at Δisoprene~700 ppt) is ~3%. Therefore, a more accurate organic nitrate yield with uncertainty might be 9(+14/-6)%.

**References**
Lee, B. H., Mohr, C., Lopez-Hilfiker, F. D., Lutz, A., Hallquist, M., Lee, L., Romer, P., Cohen, R. C., Lyer, S., Kurten, T., Hu, W., Day, D. A., Campuzano-Jost, P., Jimenez, J. L., Xu, L., Ng, N. L., Guo, H., Weber, R. J., Wilde, R. J., Brown, S. S., Koss, A., de Gouw, J., Olson, K., Goldstein, A. H., Seco, R., Kim, S., McAvey, K. M., Shepson, P. B., Starn, T. K., Baumann, K., Edgerton, E. S., Liu, J., Shilling, J. E., Miller, D. O., Brune, W., Schobesberger, S., D'Ambro, E. L., and Thornton, J. A.: Highly functionalized organic nitrates in the southeast United States: contribution to secondary organic aerosol and reactive nitrogen budgets, Proc. Nat. Acad. Sci., 113, 1516-1521, 10.1073/pnas.1508108113, 2016.

Slade, J. H., de Perre, C., Lee, L., and Shepson, P. B.: Nitrate radical oxidation of γ-terpinene: hydroxy nitrate, total organic nitrate, and secondary organic aerosol yields, Atmos. Chem. Phys., 10.5194/acp-2017-249, 2017.

Thank you for your comments. We have added a line referring to the CIMS methods you suggest in the last paragraph, at line 977:

"**Future similar field studies would benefit from the co-deployment of the complementary tool of a Chemical Ionization Mass Spectrometer (CIMS) to detect $NO_3$ + isoprene products such as organic nitrates (Lee et al., 2016, Slade et al., 2017).**"

Following your suggestion, we have added to Figure 5 lines showing the "outside" maximum uncertainty based on combined instrumental uncertainties, and have added text to the caption explaining these lines:

[Figure]

"

**Figure 5.** SOA molar yield can be determined as the slope of $\Delta pRONO_2$ vs. $\Delta$isoprene, both in mixing ratio units. The linear fit is weighted by square root of number of points used to determine each in-plume $pRONO_2$, with intercept held at zero. The slope coefficient ± one standard deviation is 0.0930 ± 0.0011. **Larger "outside" high and low limits of the slope (shown as dashed red and blue lines) are obtained by adding and subtracting from this slope the combined instrumental uncertainties, based on adding in quadrature the**

**PTR-MS uncertainty of 5% and the AMS uncertainty of 50%. This gives an overall uncertainty of 50.2%, resulting in upper and lower limit slopes of 0.140 and 0.046, respectively.** Points are colored by plume age, and size scaled by square root of number of points (the point weight used in linear fit). This plot and fit includes the nine plumes listed in Tables 1 and 2, as well as the 03:14 "unreacted" plume (at $\Delta$isoprene = -84 ppt). Error bars on isoprene are the propagated standard deviations of the (in plume - out plume) differences, for plumes in which multi-point averages were possible. Error bars on $pRONO_2$ are the same as in Figure 4. The points without error bars are single-point plumes."

And we have updated the text discussion and yields reported in the abstract to show these ranges: at line 706: "**This slope error gives a rather narrow uncertainty range for the slope (0.0930 +/- 0.0011); to obtain an upper limit in the uncertainty of this molar yield we apply the combined instrumental uncertainties, based on adding in quadrature the PTR-MS uncertainty of 5% and the AMS uncertainty of 50%. This gives an overall uncertainty of 50.2%, resulting in upper and lower limit slopes of 0.140 and 0.046, respectively; we use this maximum uncertainty estimate to report the average molar yield as 9% (+/- 5%)**."

"for which the average over 9 plumes is 9% **(+/- 5%)**. Corresponding mass yields depend on the assumed molecular formula for isoprene-$NO_3$-SOA, but the average over 9 plumes is 27% **(+/-14%), on average** larger than those previously measured in chamber studies (12 – 14% **mass yield as $\Delta$OA/$\Delta$VOC** after oxidation of both double bonds). "

**Anonymous Referee #2**
Minor comments:
Regarding the answers marked R1.2. L: 651:
Schwantes et al., 2015, show that autoxidation is not very important in HO2 regimes.
Rollins et al., 2009 likely operated in the RO2 regime and autoxidation could have contributed to that yield already. The observation time here was several hours, thus NO3 exposure was comparable to that in the plumes. So reaction time alone, cannot be an issue.

Thank you. We agree that reaction time alone will not determine whether initially formed RO2 undergo autoxidation or bimolecular reactions, and that instead the peroxy radical lifetime, driven by e.g. the concentration of HO2, will determine RO2 fate. We hope that future studies with more speciated measurements are able to better elucidate RO2 fate. We note however also that Rollins (2009) had NO2 concentration of up to 40 ppb, O3 up to 60 ppb, and two isoprene injections of 10 ppb each, meaning that radical concentrations are likely significantly higher, which would suppress autoxidation by competition of faster bimolecular rates.

**Anonymous Referee #3**
Review for revised manuscript „Secondary Organic Aerosol (SOA) yields from NO3 radical + isoprene based nighttime aircraft power plant plume transects" by J. L. Fry et al.

Fry et al. considerably improved the manuscript in their revised version. Yet there remains one major point, which needs modification before the manuscript can be published.

This reviewer requested, to show the uncertainty induced by not knowing the ratio R of NO2+/NO+ from the organic nitrates present in the aerosol. One such approach – as suggested in the review – would be a discussion on the impact of lower R on derived SOA yield. As shown in figure 3 a significant fraction of data points (eyeballing > 30%) shows R values below 0.175. As mentioned in the review, applying an R value of 0.1 (the lower envelope of the data), would reduce SOA yields to 13%, values comparable to the literature cited by the authors. The authors do not explain why they consider this comparably large number of data points as invalid, or within errors the same as R = 0.175.

To fully investigate the possibility of a lower value for R (0.1), we propagated it for this data all the way to the yield calculations below. Looking at Fig. 3, the fraction that get apportioned to pRONO2 is roughly the relative proportion that the NOx ratio data lie between the pRONO2 and NH4NO3 ratio lines. So, as an example, for data that the NOx ratio is on the current green line of 0.175, moving the green line down to 0.1 would decrease the amount of pRONO2 from ~100% to ~80% of the total nitrate (so ~20% decrease). Any data with NOx ratios already below the green line would change less:

[Figure]

However, in order to be absolutely certain, we have re-calculated the pRONO$_2$ enhancements for each plume using R=0.1 to determine how these would affect the yields. Here is a correlation plot of the R=0.1 "Lower limit" pRONO$_2$ plume enhancement values against those calculated with our best estimate for R (ratio-of-ratios=2.8, resulting in RON=0.175). Linear fit is using ODR=2:

[Figure]

It is already clear that this will not affect the yields by a factor of two as suggested by this reviewer's comment. Nevertheless, we use these new "lower limit" $pRONO_2$ plume enhancement numbers to determine a new molar yield estimate:

[Figure]

This lower limit would shift the average molar yield from 9.3% to 8.1%, and the average mass yield from 27% to 24%. As described above, however, we now report a wider error range based shown in the plot above, to be even more conservative in our conclusions.

In sum, we still think that applying a ratio-of-ratios based on literature averages and using the stated accuracy is the most robust and unbiased method for computing $pRONO_2$ concentrations. In the SI, we already stated the uncertainty in $pRONO_2$ from uncertainty of the ratio-of-ratios as +/-20% (and overall $pRONO_2$ concentration uncertainty as +/-50%).

The response of the authors implies that they assume an average of literature R values to be relevant to the conditions they encounter in a regime where SOA is formed from the nighttime reaction of isoprene with NO3. This is not understandable, when only one reference (Rollins et al., 2009) explored organic nitrates formed from controlled chamber studies exploring isoprene + NO3 reactions. That reference indicates an R value of 0.1. Why would other systems, with organic nitrates formed from e.g. aromatics (Sato et al.) or selected according to availability (Farmer et al.), be considered of the same relevance in this case?

We are not aware of any compelling evidence that the organic nitrate NOx ratios have a composition dependence, nor reason that one would expect there to be. Regarding the Rollins 2009 study, they reported $NO_2^+/NO^+$ ratios were 0.156 for $pRONO_2$ and 0.35 for ammonium nitrate. Thus, it is not clear why the referee states that study supports a $pRONO_2$ ratio of 0.1. Moreover, in our methods, we use the ratio-of-ratio method, as stated in the manuscript, rather than a fixed ratio for $pRONO_2$ based on studies conducted on different instruments as others have done. For the Rollins 2009 study, the ratio-of-ratio would be 0.35/0.156 = 2.24. If that value was used instead here (rather than the 2.8 from averaging multiple studies), $pRONO_2$ concentrations (and overall SOA yields) calculated in this study would be higher (not lower as this referee seems to be arguing for). I.e. the green line in Fig. 3 would move up and a larger portion of the NOx ratio data would fall below that line.

The major finding in the study, namely a larger SOA yield in the plumes than in previous chamber studies, directly depends on the R value used to calculate organic nitrate mass. Therefore a discussion of the uncertainty imposed by applying an unknown R value is mandatory to inform the reader on the certainty with which organic nitrate mass concentrations can be derived from AMS data sets. One such way is the application of an R-value as low as justifiable by the data, as this implies the analysis of data with respect to lower limits of organic nitrate mass and SOA yields.

We don't believe that including the lower limit R-value shown above in our manuscript is appropriate, because it introduces a bias in one direction. It also assumes that the choice of R is the largest uncertainty in this analysis, and we know that there are several other sources of uncertainty, so we don't wish to place sole emphasis on the choice of R. However, we think this reviewer may also be uncomfortable with the lack of uncertainty or range reported on the yield numbers reported in the abstract, which using R = 0.1 as an alternate value would provide. We have added to Figure 5 upper- and lower-limit yield slopes based on the outer bounds of the fully error-propagated x and y error bars, which provides a larger range of potential yields. The new, larger molar yield range of 9% (+/-5%) and corresponding mass yield range of 27% (+/-14%) does not unequivocally exclude the Rollins 2009 SOA (delOA/delVOC) mass yield report of 12-14%, but it does suggest that yields may be larger. We report this range in the hope that this will spur further work.

[revised manuscript text omitted]

Juliane Fry 6/1/2018 3:35 PM

Juliane Fry 6/1/2018 3:35 PM

Juliane Fry 5/26/2018 5:28 PM
**Formatted Table**

Juliane Fry 5/26/2018 5:18 PM

Juliane Fry 5/26/2018 5:14 PM

Juliane Fry 5/26/2018 5:14 PM

Juliane Fry 6/1/2018 3:25 PM

Juliane Fry 5/26/2018 5:14 PM

Juliane Fry 5/26/2018 5:14 PM

Juliane Fry 6/1/2018 3:25 PM

Juliane Fry 5/26/2018 5:14 PM

Juliane Fry 5/26/2018 5:14 PM

Juliane Fry 5/26/2018 5:14 PM

Juliane Fry 6/1/2018 3:25 PM

Juliane Fry 5/26/2018 5:14 PM

Juliane Fry 6/1/2018 3:25 PM

Juliane Fry 5/26/2018 5:14 PM

Juliane Fry 6/1/2018 3:27 PM

Juliane Fry 5/26/2018 5:14 PM

Juliane Fry 6/1/2018 3:27 PM

mass is the mass only of the –$ONO_2$ portion of the organonitrate aerosol). At standard
conditions of 273 K and 1 atm (all aerosol data are reported with this STP definition), 1000 ppt
$NO_3$ = 2.77 µg m$^{-3}$, so each $\Delta M_{pRONO2}$ is multiplied by 361 ppt (µg m$^{-3}$)$^{-1}$ to determine this molar
yield:
$$Y_{SOA,molar} = \frac{(pRONO2_{plume} \pm SD_{pRONO2plume}) - (pRONO2_{bkg} \pm SD_{pRONO2bkg})}{-[(isop_{plume} \pm SD_{isopplume}) - (isop_{bkg} \pm SD_{isopbkg})]} \times \frac{361 \; ppt \; NO_3}{µg \; m^{-3}}$$ (3)
The SOA molar yields resulting from this calculation are shown in Table 2, spanning a range of
5-28%, with uncertainties indicated based on the SDs in measured AMS and isoprene
concentrations. In addition to this uncertainty based on measurement precision and ambient
variability, there is an uncertainty of 50% in the AMS derived-organic nitrate mass loadings (see
SI) and 25% in the PTR-MS isoprene concentrations (Warneke et al., 2016). The average molar
$pRONO_2$ yield across all plumes, with each point weighed by the inverse of its standard
deviation, is 9%. (As noted below, the yield appears to increase with plume age, so this average
obscures that trend.) An alternate graphical analysis of molar SOA yield from all nine plumes
plus one 'null' plume (03:14, in which no isoprene had yet reacted and thus not included in
Tables 1 and 2) obtains the same average molar yield of 9% (Fig. 5). Here, the molar yield is
the slope of a plot of plume change in $pRONO_2$ vs plume change in isoprene. The slope is
determined by a linear fit with points weighted by the square root of the number of AMS data
points used to determine in-plume $pRONO_2$ in each case. This slope error gives a rather narrow
uncertainty range for the slope (0.0930 +/- 0.0011); to obtain an upper limit in the uncertainty of
this molar yield we apply the combined instrumental uncertainties, based on adding in
quadrature the PTR-MS uncertainty of 5% and the AMS uncertainty of 50%. This gives an
overall uncertainty of 50.2%, resulting in upper and lower limit slopes of 0.140 and 0.046,
respectively; we use this maximum uncertainty estimate to report the average molar yield as 9%
(+/- 5%). We have not corrected the calculated yields for the possibility of $NO_3$ heterogeneous
uptake, which could add a nitrate functionality to existing aerosol. Such a process could be rapid
if the uptake coefficient for $NO_3$ were 0.1, a value characteristics of unsaturated substrates (Ng
et al., 2017), but would not contribute measurably at more conventional $NO_3$ uptake coefficients
of 0.001 (Brown and Stutz 2012).

[Figure]

[Figure]

Juliane Fry 5/26/2018 9:05 PM

**Figure 5.** SOA molar yield can be determined as the slope of $\Delta pRONO_2$ vs. $\Delta$isoprene, both in mixing ratio units. The linear fit is weighted by square root of number of points used to determine each in-plume $pRONO_2$, with intercept held at zero. The slope coefficient ± one standard deviation is 0.0930 ± 0.0011. Larger "outside" high and low limits of the slope (shown as dashed red and blue lines) are obtained by adding and subtracting from this slope the combined instrumental uncertainties, based on adding in quadrature the PTR-MS uncertainty of

5% and the AMS uncertainty of 50%. This gives an overall uncertainty of 50.2%, resulting in upper and lower limit slopes of 0.140 and 0.046, respectively. Points are colored by plume age, and size scaled by square root of number of points (the point weight used in linear fit). This plot and fit includes the nine plumes listed in Tables 1 and 2, as well as the 03:14 "unreacted" plume (at $\Delta$isoprene = -84 ppt). Error bars on isoprene are the propagated standard deviations of the (in plume - out plume) differences, for plumes in which multi-point averages were possible. Error bars on pRONO2 are the same as in Figure 4, converted to ppt. The points without error bars are single-point plumes.

Juliane Fry 5/26/2018 9:08 PM
Juliane Fry 5/15/2018 7:09 PM
Juliane Fry 5/15/2018 7:09 PM
Juliane Fry 6/3/2018 5:53 PM
Juliane Fry 5/26/2018 9:07 PM
Juliane Fry 5/26/2018 5:28 PM
Juliane Fry 5/26/2018 9:07 PM
Juliane Fry 5/26/2018 9:07 PM
Juliane Fry 5/26/2018 9:07 PM
Juliane Fry 5/26/2018 9:07 PM
Juliane Fry 5/26/2018 9:08 PM
Font color: Auto

To estimate SOA **mass** yields, we need to make some assumption about the mass of the
organic molecules containing the nitrate groups that lead to the observed nitrate aerosol mass
increase. The observed changes in organic aerosol are too variable to be simply interpreted as
the organic portion of the aerosol organic nitrate molecules. We conservatively assume the
organic mass to be approximately double the nitrate mass (62 g mol$^{-1}$), based on an "average"
molecular structure of an isoprene nitrate with 3 additional oxygens: e.g. a tri-hydroxynitrate
(with organic portion of formula $C_5H_{11}O_3$, 119 g mol$^{-1}$), consistent with 2$^{nd}$-generation oxidation
product structures suggested in Schwantes, et al. (Schwantes et al., 2015). Based on this
assumed organic to nitrate ratio, all plumes' expected organic mass increases would be less
than the typical variability in organic of 0.75 μg m$^{-3}$. This assumed structure is consistent with
oxidation of both double bonds, which appears to be necessary for substantial condensation of
isoprene products, and which structures would have calculated vapor pressures sufficiently low
to partition to the aerosol phase (Rollins et al., 2009). Another possible route to low vapor
pressure products is intramolecular H rearragement reactions, discussed below in Section 4.3,
which would not require oxidant reactions at both double bonds. In the case of oxidant reactions
at both double bonds, it is difficult to understand how the second double bond would be oxidized
unless by another nitrate radical, which would halve these assumed organic to nitrate ratios
(assuming the nitrate is retained in the molecules). On the other hand, any organic nitrate
aerosol may lose NO$_3$ moieties, increasing the organic to nitrate ratio. Given these uncertainties
in both directions, we use the assumed "average" structure above to guess an associated
organic mass of double the nitrate mass. Thus, to estimate SOA mass yield, we multiply the
increase in organic nitrate aerosol mass concentration by three (i.e., $2 \times \Delta M_{pRONO2} + \Delta M_{pRONO2}$),
and divide by the observed decrease in isoprene, converted to μg m$^{-3}$ by multiplying by 329 ppt
(μg m$^{-3}$)$^{-1}$, the conversion factor based on isoprene's molecular weight of 68.12 g mol$^{-1}$.
$$Y_{SOA,mass} = \frac{(pRONO2_{plume} \pm SD_{pRONO2plume}) - (pRONO2_{bkg} \pm SD_{pRONO2bkg})}{-[(isop_{plume} \pm SD_{isopplume}) - (isop_{bkg} \pm SD_{isopbkg})]} \times 3 \times \frac{329ppt}{\mu g\ m^{-3}}$$    (4)
Note that the SOA mass yield reported here is based on the (assumed) mass of organic aerosol
plus the (organo)nitrate aerosol formed in each plume. If instead the yield were calculated using
only the assumed increase in **organic** mass (i.e., 2x $\Delta M_{pRONO2}$ instead of 3x $\Delta M_{pRONO2}$), which
would be consistent with the method used in Rollins, et al. (Rollins et al., 2009) and Brown et al.
(Brown et al., 2009), the mass yields would be 2/3 the values reported here. However, since
SOA mass yield is typically defined based on the total increase in aerosol mass, we use the
definition with the sum of the organic and nitrate mass here. This results in an average SOA
mass yield of 27%, with propagated instrumental errors (see caption to Fig. 5) giving a range of
27% +/- 14%.
We note also that correlation of in-plume increases in OA with pRONO$_2$ (Fig. 6) point to a
substantially larger 5:1 organic-to-nitrate ratio; if this were interpreted as indicating that the
average molecular formula of the condensing organic nitrate has 5 times the organic mass as
nitrate, this would increase the SOA mass yields reported here. However, due to the
aforementioned possibility of additional sources of co-condensing organic aerosol, which led us
to avoid using ΔOA in determining SOA yields, we do not consider this to be a direct indication

Juliane Fry 5/15/2018 8:20 PM

Juliane Fry 5/15/2018 8:17 PM
Juliane Fry 5/15/2018 8:17 PM

of the molecular formula of the condensing organic nitrate. Including OA in the SOA yield
determination, based on this 5:1 slope rather than the assumed 2:1 OA:pRONO$_2$, would give
2.5 times larger SOA mass yields than reported here.

[Figure]

**Figure 6**. Correlation of organic aerosol mass concentration with pRONO$_2$ mass concentration
for the full 2 July flight (grey points and red fit line, fitted slope and thus average OA/pRONO$_2$
mass ratio of ~30) and for the points during the selected plumes (colored points, colored by
plume age, average OA/pRONO$_2$ mass ratio of ~ 5).

**Table 2.** SOA Yields for each plume observation, estimated plume age, and likely origin. See
text for description of uncertainty estimates. For the mass yields, the calculated SOA mass
increase includes both the organic and (organo)nitrate aerosol mass; the measurements for OA
increases shown in Figure 6 do not include the nitrate mass.

| plume number | plume time (UTC) | SOA molar yield (fraction) [± SD] | SOA mass yield (fraction) [± SD] | plume age from $O_3$/ $NO_2$ clock assuming S=1 (hours) | Likely NOx origin & altitude (m) |
|---|---|---|---|---|---|
| 1 | 7/2/13 2:18 | 0.09 [0.06] | 0.25 [0.17] | 2.5 | Greene County @ 540 m |
| 2 | 7/2/13 2:20 | 0.07 | 0.21 | 1.5 | *ibid* |
| 3 | 7/2/13 2:21 | 0.12 [0.10] | 0.32 [0.25] | 1.5 | *ibid* |
| 4 | 7/2/13 3:03 | 0.13 | 0.36 | 1.5 | Gaston @ 720 m |
| 5 | 7/2/13 3:55 | 0.06 [0.07] | 0.17 [0.20] | 1.4 | Miller / Gorgas @ 690 m |
| 6 | 7/2/13 4:34 | 0.05 [0.03] | 0.15 [0.09] | 2 | *ibid* |
| 7 | 7/2/13 4:37 | 0.10 [0.09] | 0.26 [0.24] | 5.5 | *ibid* |
| 8 | 7/2/13 4:39 | 0.16 [0.10] | 0.45 [0.28] | 5.8 | Miller / Gorgas @ 1120 m |
| 9 | 7/2/13 5:04 | 0.28 [0.19] | 0.77 [0.52] | 6.3 | Gaston @ 1280 m |

Juliane Fry 6/1/2018 3:38 PM

Juliane Fry 6/1/2018 3:38 PM

Juliane Fry 6/1/2018 3:38 PM

Juliane Fry 6/1/2018 3:38 PM

Juliane Fry 6/1/2018 3:39 PM

Juliane Fry 6/1/2018 3:39 PM

Juliane Fry 6/1/2018 3:39 PM

Juliane Fry 6/1/2018 3:39 PM

Juliane Fry 6/1/2018 3:39 PM

Juliane Fry 6/1/2018 3:39 PM

[revised manuscript text omitted]

Juliane Fry 6/1/2018 8:26 AM
**Formatted** ... [10]
Juliane Fry 5/26/2018 8:10 PM
**Formatted** ... [11]
Juliane Fry 5/26/2018 8:10 PM
**Formatted** ... [12]
Juliane Fry 5/26/2018 8:10 PM
**Formatted** ... [13]
Juliane Fry 6/1/2018 8:25 AM
**Formatted** ... [14]
Juliane Fry 6/1/2018 8:25 AM
**Formatted** ... [15]
Juliane Fry 6/1/2018 8:26 AM
**Formatted** ... [16]
Juliane Fry 5/26/2018 6:54 PM
**Formatted Table** ... [17]
Juliane Fry 5/26/2018 8:07 PM
**Formatted** ... [19]
Juliane Fry 5/26/2018 7:59 PM
**Formatted** ... [18]
Juliane Fry 5/26/2018 8:07 PM
**Formatted** ... [21]
Juliane Fry 5/26/2018 7:59 PM
**Formatted** ... [20]
Juliane Fry 5/26/2018 8:07 PM
**Formatted** ... [23]
Juliane Fry 5/26/2018 7:59 PM
**Formatted** ... [22]
Juliane Fry 5/26/2018 8:07 PM
**Formatted** ... [25]
Juliane Fry 5/26/2018 7:59 PM
**Formatted** ... [24]
Juliane Fry 5/26/2018 8:07 PM
**Formatted** ... [27]
Juliane Fry 5/26/2018 7:59 PM
**Formatted** ... [26]
Juliane Fry 5/26/2018 8:07 PM
**Formatted** ... [29]
Juliane Fry 5/26/2018 7:59 PM
**Formatted** ... [28]
Juliane Fry 5/26/2018 8:07 PM
**Formatted** ... [31]
Juliane Fry 5/26/2018 7:59 PM
**Formatted** ... [30]
Juliane Fry 5/26/2018 8:07 PM
**Formatted** ... [33]
Juliane Fry 5/26/2018 7:59 PM
**Formatted** ... [32]
Juliane Fry 5/26/2018 8:07 PM
**Formatted** ... [35]
Juliane Fry 5/26/2018 7:59 PM
**Formatted** ... [34]

[revised manuscript text omitted]

ΔISOP (ppt)
[± SD]

| | | |
|---|---|---|
| **Page 19: [2] Deleted** | **Juliane Fry** | **5/26/18 5:14 PM** |

-335
[128]

| | | |
|---|---|---|
| **Page 19: [3] Deleted** | **Juliane Fry** | **5/26/18 5:14 PM** |

-228
[121]

| | | |
|---|---|---|
| **Page 19: [4] Deleted** | **Juliane Fry** | **5/26/18 5:14 PM** |

-255
[251]

| | | |
|---|---|---|
| **Page 19: [5] Deleted** | **Juliane Fry** | **5/26/18 5:14 PM** |

-713
[219]

| | | |
|---|---|---|
| **Page 19: [6] Deleted** | **Juliane Fry** | **5/26/18 5:14 PM** |

-298
[197]

| | | |
|---|---|---|
| **Page 19: [7] Deleted** | **Juliane Fry** | **5/26/18 5:14 PM** |

-443
[75]

| | | |
|---|---|---|
| **Page 19: [8] Deleted** | **Juliane Fry** | **5/26/18 5:14 PM** |

-293
[131]

| | | |
|---|---|---|
| **Page 28: [9] Deleted** | **Juliane Fry** | **5/26/18 10:23 PM** |

**4.4 Two urban plume case studies**

In addition to the nine power plant plumes analyzed above to determine the $NO_3$ + isoprene SOA molar yield, towards the end of the July 2 flight, the Birmingham urban plume was intercepted twice (around 5:36 am and 6:37 am UTC, Fig. 8). These downwind urban plumes are among the most aged plumes (estimated at 5.2 and 5.8 hours, respectively), but are also substantially more diffuse than the narrow power plant plume intercepts and have lower peak $P(NO_3)$. Nevertheless, we note that these two plumes contain periods of apparent anti-correlation of isoprene and organic nitrate aerosol time series and high apparent SOA molar yields (23%, 19%) and mass yields (62%, 51%), if calculated by the same method as above and omitting the period of vertical profiling in the second plume. Potentially complicating these urban SOA yield determinations is the fact that the inorganic fraction of nitrate was much larger than in the power plant plumes (see Fig. 8). The background isoprene is also somewhat lower in these urban plumes, potentially shifting the $NO_3/N_2O_5$ fate to reactions other than $NO_3$ + isoprene (see Fig. S4 in Edwards et al. (Edwards et al., 2017)). The aerosol surface area is not noticeably higher in these urban plumes, which one might expect to lead to a larger contribution of $N_2O_5$ uptake and hydrolysis. In the more complex mix of gases characteristic of an urban plume, we hesitate to attribute these apparent yields exclusively to the $NO_3$ + isoprene reaction.

[Figure]

**Figure 8. Flight map and time series of two urban plume intercepts, showing anticorrelation of organic nitrate and isoprene. These more diffuse plumes, with lower $P$(NO₃) and larger inorganic nitrate contribution, make yield determination more uncertain, so we do not include them in the overall yield determination. However, using the same methodology as for the power plant plumes would give similarly high yields for these very aged plumes.**

| Page 45: [10] Formatted | Juliane Fry | 6/1/18 8:26 AM |
|---|---|---|
| Normal | | |

| Page 45: [11] Formatted | Juliane Fry | 5/26/18 8:10 PM |
|---|---|---|
| Font:11 pt, Not Bold | | |

| Page 45: [12] Formatted | Juliane Fry | 5/26/18 8:10 PM |
|---|---|---|
| Font:11 pt, Not Bold | | |

| Page 45: [13] Formatted | Juliane Fry | 5/26/18 8:10 PM |
|---|---|---|
| Font:11 pt, Not Bold | | |

| Page 45: [14] Formatted | Juliane Fry | 6/1/18 8:25 AM |
|---|---|---|
| Font color: Auto | | |

| Page 45: [15] Formatted | Juliane Fry | 6/1/18 8:25 AM |
|---|---|---|
| Font:Not Bold, Font color: Auto | | |

| Page 45: [16] Formatted | Juliane Fry | 6/1/18 8:26 AM |
|---|---|---|
| Font:(Default) Times, 10 pt, Not Bold, Font color: Auto | | |

| Page 45: [17] Formatted Table | Juliane Fry | 5/26/18 6:54 PM |
|---|---|---|

Formatted Table

| Page 45: [18] Formatted | Juliane Fry | 5/26/18 7:59 PM |
|---|---|---|

Right

| Page 45: [19] Formatted | Juliane Fry | 5/26/18 8:07 PM |
|---|---|---|

Font:(Default) Arial, 10 pt

| Page 45: [20] Formatted | Juliane Fry | 5/26/18 7:59 PM |
|---|---|---|

Right

| Page 45: [21] Formatted | Juliane Fry | 5/26/18 8:07 PM |
|---|---|---|

Font:(Default) Arial, 10 pt

| Page 45: [22] Formatted | Juliane Fry | 5/26/18 7:59 PM |
|---|---|---|

Right

| Page 45: [23] Formatted | Juliane Fry | 5/26/18 8:07 PM |
|---|---|---|

Font:(Default) Arial, 10 pt

| Page 45: [24] Formatted | Juliane Fry | 5/26/18 7:59 PM |
|---|---|---|

Right

| Page 45: [25] Formatted | Juliane Fry | 5/26/18 8:07 PM |
|---|---|---|

Font:(Default) Arial, 10 pt

| Page 45: [26] Formatted | Juliane Fry | 5/26/18 7:59 PM |
|---|---|---|

Right

| Page 45: [27] Formatted | Juliane Fry | 5/26/18 8:07 PM |
|---|---|---|

Font:(Default) Arial, 10 pt

| Page 45: [28] Formatted | Juliane Fry | 5/26/18 7:59 PM |
|---|---|---|

Right

| Page 45: [29] Formatted | Juliane Fry | 5/26/18 8:07 PM |
|---|---|---|

Font:(Default) Arial, 10 pt

| Page 45: [30] Formatted | Juliane Fry | 5/26/18 7:59 PM |
|---|---|---|

Right

| Page 45: [31] Formatted | Juliane Fry | 5/26/18 8:07 PM |
|---|---|---|

Font:(Default) Arial, 10 pt

| Page 45: [32] Formatted | Juliane Fry | 5/26/18 7:59 PM |
|---|---|---|

Right

| Page 45: [33] Formatted | Juliane Fry | 5/26/18 8:07 PM |
|---|---|---|

Font:(Default) Arial, 10 pt

| Page 45: [34] Formatted | Juliane Fry | 5/26/18 7:59 PM |
|---|---|---|

Right

**Page 45: [35] Formatted**         **Juliane Fry**              **5/26/18 8:07 PM**

Font:(Default) Arial, 10 pt